# The interaction of $CO_2$ concentrations and water stress in semi-arid areas causes diverging response in instantaneous water use efficiency and carbon isotope composition

**Na Zhao**[1], **Ping Meng**[2], **Yabing He**[1], **Xinxiao Yu**[1]*

[1] College of soil and water conservation, Beijing Forestry University, Beijing 100083, P.R. China
[2] Research Institute of Forestry, Chinese Academy of Forestry 100091, Beijing, P.R. China

**Abstract.** In the context of global warming attributable to the increasing levels of $CO_2$, severe drought can be anticipated in areas with chronic water shortages (semi-arid areas), which necessitates research on the interaction between elevated atmospheric concentrations of $CO_2$ and drought on plant photosynthetic discrimination. It is commonly surveyed that the $^{13}C$ fractionation derived from the $CO_2$ diffusion occurred from ambient air to sub-stomatal cavity, and little investigate the $^{13}C$ fractionation generated from the site of carboxylation to cytoplasm before sugars transportation outward the leaf, which may respond to the environmental conditions (i. e. $CO_2$ concentration and water stress) and their interactions. Therefore, saplings of typical species to a semi-arid area of Northern China that have similar growth status—*Platycladus orientalis* and *Quercus variabilis*—were selected and cultivated in growth chambers with orthogonal treatments (four $CO_2$ concentrations [$CO_2$] × five soil volumetric water contents (SWC)). The $\delta^{13}C$ of water-soluble compounds extracted from leaves of saplings was measured to determine the instantaneous water use efficiency ($WUE_{cp}$) after cultivation. Instantaneous water use efficiency derived from gas exchange ($WUE_{ge}$) was integrated to estimate differences in $\delta^{13}C$ signal variation before leaf-exported translocation of primary assimilates. The $WUE_{ge}$ of the two species both decreased with increased soil moisture, and increased with elevated [$CO_2$] at 35%–80% of field capacity (FC) by strengthening photosynthetic capacity and reducing transpiration. Differences in instantaneous water use efficiency (iWUE) according to distinct environmental changes differed between species. The $WUE_{ge}$ of *P. orientalis* was significantly greater than that of *Q. variabilis*, while the opposite results were obtained in a comparison of $WUE_{cp}$ in two species. Total $^{13}C$ fractionation from the site of carboxylation to cytoplasm before sugars transportation (total $^{13}C$ fractionation) was clearly species-specific, as demonstrated in the interaction of [$CO_2$] and SWC. Rising [$CO_2$] coupled with moistened soil generated increasing disparities of $\delta^{13}C$ between the water soluble compounds ($\delta^{13}C_{WSC}$) and estimated by gas-exchange observation ($\delta^{13}C_{obs}$) in *P. orientalis* with amplitude of 0.0328‰–0.0472‰. Furthermore, differences between $\delta^{13}C_{WSC}$ and $\delta^{13}C_{obs}$ of *Q. variabilis* increased as $CO_2$ concentration and SWC increased (0.0384‰–0.0466‰). The $^{13}C$ fractionations from mesophyll conductance and post-carboxylation both contributed to the total $^{13}C$ fractionation determined by two measurements (1.06%-24.94% and 75.30%-98.9% of total $^{13}C$ fractionation, respectively). Total $^{13}C$ fractionations were linearly dependent on $g_s$, indicating post-carboxylation fractionation was attributed to environmental variation. Thus, clear description of magnitude and environmental dependence of apparent post-carboxylation fractionation is worth our attention in photosynthetic fractionation.

**Key words:** Post-carboxylation fractionation; Carbon isotope fractionation; Elevated $CO_2$

concentration; Soil volumetric water content; Instantaneous water use efficiency

**1 Introduction**

Since the onset of the industrial revolution, atmospheric $CO_2$ concentration has increased at an
annual rate of 0.4%, and is expected to increase further to 700 µmol·mol$^{-1}$, together with more frequent
periods of low water availability (IPCC, 2014). Increasing atmospheric $CO_2$ concentrations that trigger
an ongoing greenhouse effect will not only lead to fluctuations in global patterns of precipitation, but
will amplify drought in arid regions, and lead to more frequent occurrences of extreme drought events
in humid regions (Lobell et al., 2014). Accompanying the increasing concentration of $CO_2$, the mean
$\delta^{13}C$ of atmospheric $CO_2$ is depleted by 0.02‰–0.03‰ year$^{-1}$ (data available from the
CU-INSTAAR/NOAACMDL network for atmospheric $CO_2$; http://www.esrl.noaa.gov/gmd/).
The carbon isotopic composition determined recently could respond more subtly to environmental
changes and their influences on diffusion via plant physiological and metabolic processes (Gessler et
al., 2014; Streit et al., 2013). While the depletion of $\delta^{13}C_{CO_2}$ has been shown in the atmosphere,
variations in $CO_2$ concentration itself might also affect the $\delta^{13}C$ of plant organs that, in turn, respond
physiologically to climatic change (Gessler et al., 2014). The carbon discrimination ($^{13}\Delta$) of leaves
could also provide timely feedback about the availability of soil moisture and the atmospheric vapor
pressure deficit (Cernusak et al., 2012). Discrimination against $^{13}C$ in leaves relies mainly on
environmental factors that affect the ratio of intercellular to ambient $CO_2$ concentration ($C_i/C_a$) and
Rubisco activities, even the mesophyll conductance derived from the difference of $CO_2$ concentrations
between intercellular site and chloroplast (Farquhar et al., 1982; Cano et al., 2014). As changes in
environmental conditions affect photosynthetic discrimination, they are expected to be recorded
differentially in the $\delta^{13}C$ of water-soluble compounds ($\delta^{13}C_{WSC}$) of the different plant organs.
Meanwhile, several processes during photosynthesis alter the $\delta^{13}C$ of carbon transported within plants.
Carbon-fractionation during photosynthetic $CO_2$ fixation has been described and reviewed elsewhere
(Farquhar et al., 1982; Farquhar and Sharkey, 1982).
Post-photosynthetic fractionation is derived from equilibrium and kinetic isotopic effects, which
determines isotopic differences between metabolites and intramolecular reaction positions, defined as
"post-photosynthetic" or "post-carboxylation" fractionation (Jäggi et al., 2002; Badeck et al., 2005;
Gessler et al., 2008). Post-carboxylation fractionation in plants includes the carbon discriminations that
follow carboxylation of ribulose-1, 5-bisphosphate, and internal diffusion (RuBP, 27‰), as well as
related transitory starch metabolism (Gessler et al., 2008; Gessler et al., 2014), fractionation in leaves,
fractionation-associated phloem transport, remobilization or storage of soluble carbohydrates, and
starch metabolism fractionation in sink tissue (tree rings). In the synthesis of soluble sugars,
$^{13}C$-depletions of triose phosphates occur during exportation from the cytoplasm, and during
production of fructose-1, as does 6-bisphosphate by aldolase in transitory starch synthesis (Rossmann
et al., 1991; Gleixner and Schmidt, 1997). Synthesis of sugars before transportation to the twig is
associated with the post-carboxylation fractionation generated in leaves. Although these are likely to
play a role, what should also be considered is the $CO_2$ concentration in the chloroplast ($C_c$), not in the
intercellular space, as used in the simplified equation of the Farquar's model (Evans et al., 1986;
Farquhar et al., 1989) is actually defined as carbon isotope discrimination ($\delta^{13}C$). Indeed, difference
between gas-exchange derived values and online measurements of $\delta^{13}C$ has been widely used to
estimate $C_i$-$C_c$ and mesophyll conductance for $CO_2$ (Le Roux et al., 2001; Warren and Adams, 2006;
Flexas et al., 2006; Evans et al., 2009; Flexas et al., 2012; Evans and von Caemmerer 2013). In this

regard, changes in mesophyll conductance could be partly responsible for the differences from two measurements, as it generally increases in the short term in response to elevated $CO_2$ (Flexas et al., 2014), whereas it tends to decrease under drought (Hommel et al., 2014; Théroux-Rancourt et al., 2014). Therefore, it is necessary to avoid confusion of carbon isotope discrimination derived from synthesis of soluble sugars or/and mesophyll conductance, and furthermore, whether and what magnitude of these carbon fractionations are related to environmental variation have not yet been investigated.

The simultaneous isotopic analysis of leaves is a recent refinement in isotopic studies that allows us to determine the temporal variation in isotopic fractionation (Rinne et al., 2016), which may help decipher environmental conditions more reliably. Newly assimilated carbohydrates can be extracted, and are defined as the water-soluble compounds (WSCs) in leaves (Brandes et al., 2006; Gessler et al., 2009), which can also be associated with an assimilation-weighted average of $C_i/C_a$ (and $C_c/C_a$) photosynthesized over a period ranging from a few hours to 1–2 d (Pons et al., 2009). However, there is a dispute whether the fractionation stemmed from post-carboxylation or/and mesophyll resistance may alter the stable signatures of leaf carbon and thence influence instantaneous water use efficiency (iWUE). In addition, the way in which iWUE derived from theses isotopic fractionations responds to different environmental factors, such as elevated $[CO_2]$ and/or soil water gradients, has yet to be observed.

Consequently, we investigated the $\delta^{13}C$ of fast-turnover carbohydrate pool in leaves from saplings of two typical species to semi-arid areas of China—*Platycladus orientalis* and *Quercus variabilis*—together with simultaneous gas exchange measurements in control-environment of growth chambers (FH-230). Our goals are to differentiate the $^{13}C$ fractionation from the site of carboxylation to cytoplasm before sugars transportation (total $^{13}C$ fractionation) of *P. orientalis* and *Q. variabilis*, which were determined from the $\delta^{13}C$ of water-soluble compounds and gas-exchange measurements, and then to discuss the potential causes for the observed divergence, estimate the contributions of post-photosynthetic and mesophyll resistance on these differences, and describe how these carbon isotopic fractionations respond to the interactive effects of elevated $[CO_2]$ and water stress.

## 2 Material and Methods

### 2.1 Study site and design

Saplings of *P. orientalis* and *Quercus variabilis* were selected as experimental material from the Capital Circle forest ecosystem station, a part of Chinese Forest Ecosystem Research Network (CFERN, 40°03'45"N, 116°5'45"E) in Beijing, China. This region is populated by trees of *Platycladus orientalis* (L.) Franco and *Quercus variabilis* Bl. Saplings of two species that have similar ground diameters, heights, and growth statuses were selected. One sapling from two species was placed in one pot (22 cm in diameter and 22 cm in height). Undisturbed soil samples were collected from the field, sieved (with all particles >10 mm removed), and placed into the pots. The soil bulk density in each pot was maintained at 1.337–1.447 g cm$^{-3}$. After the rejuvenation for one month, potted-saplings were placed into chambers for orthogonal cultivation.

The controlled experimental treatments were conducted in growth chambers (FH-230, Taiwan Hipoint Corporation, Kaohsiung City, Taiwan). To imitate the meteorological factors of growth seasons in the research region, the daytime temperature in chambers was set to $25 \pm 0.5°C$ from 07:00 to 17:00, and the night-time temperature was $18 \pm 0.5°C$ from 17:00 to 07:00. Relative humidity was

maintained at 60% and 80% during the daytime and night, respectively. The light system was activated in the daytime and shut down at night. The average daytime light intensity was maintained at 200−240 $\mu$mol m$^{-2}$ s$^{-1}$. The central controlling system of the chambers (FH-230) can timely monitor and control the $CO_2$ concentration. Two growth chambers (A and B) were used in our study. Chamber A was switched in turn to maintain the $CO_2$ concentration of 400 ppm ($C_{400}$) and 500 ppm ($C_{500}$). The other one was adjusted to maintain the $CO_2$ concentration of 600 ppm ($C_{600}$) and 800 ppm ($C_{600}$). The target concentrations of $CO_2$ in the chambers were permitted the standard deviation of ± 50 ppm during cultivation. Thus, the gradient of four $CO_2$ concentrations in our study was formed. Detectors inside the chambers monitored and maintained the target concentrations of $CO_2$.

We designed a device to irrigate the potted saplings automatically and avoid heterogeneity caused by interruptions in watering process (Fig. 1). It consisted of a water storage tank, holder, controller, soil moisture sensors, and drip irrigation components. Prior to use, the water tank was filled with water, and the soil moisture sensor was inserted to a uniform depth in the soil. After connecting the controller to an AC power supply, target soil volumetric water content (SWC) could be set and monitored by soil moisture sensors. Since timely SWC could be sensed by the sensors, the automatic irrigation device can be regulated to water or stop watering the plants. One drip irrigation device was installed per chamber. Based on the average field capacity (FC) of potted soil determined (30.70%), five levels of SWC were maintained before the orthogonal cultivations, as follows: 100% FC (or CK) (SWC approximately 27.63%−30.70%), 70%−80% of FC (SWC approximately 21.49%−24.56%), 60%−70% of FC (SWC approximately 18.42%−21.49%), 50%−60% of FC (SWC approximately 15.35%−18.42%), and 35%−45% of FC (SWC approximately 10.74%−13.81%).

While undergoing 20 groups of orthogonal treatments for [$CO_2$] × SWC, the saplings were ready for sampling. Due to one chamber only containing five plant-pots (per species) and one pot one SWC level under one $CO_2$ concentration, two saplings per specie in one orthogonal treatment were replicated for two periods, respectively. Each period per orthogonal treatment continued for 7 days. Pots were rearranged periodically to minimize non-uniform illumination. All orthogonal tests were formed as: elevated $CO_2$ concentration gradient for $C_{400}$ (during June 2−9, June 12−19, June 21−28, and July 2−9, 2015, $C_{400}$), $C_{500}$ (during July 11−18, July 22−29, August 4−11, and August 15−22, 2015, $C_{500}$), $C_{600}$ (during June 2−9, June 12−19, June 21−28, and July 2−9, 2015, $C_{600}$), and $C_{800}$ (during July 11−18, July 22−29, August 4−11, and August 15−22, 2015, $C_{800}$), combined with a soil-water gradient for 35%−45% of FC, 50%−60% of FC, 60%−70% of FC, and 70%−80% of FC and 100% FC (CK).

**2.2 Foliar gas exchange measurement**

Fully expanded primary annual leaves of the saplings were measured with a portable infrared gas photosynthesis system (LI-6400, Li-Cor, Lincoln, US) before and after the 7-day cultivation. Two saplings per specie were replicated per treatment (SWC × [$CO_2$]). For each sapling, four leaves were chosen and then four measurements were conducted on each leaf. The main photosynthetic parameters, such as net photosynthetic rate ($P_n$) and transpiration rate ($T_r$), were measured. Based on the theories proposed by Von Caemmerer and Farquhar (1981), stomatal conductance ($g_s$) and intercellular $CO_2$ concentration ($C_i$) were calculated by the Li-Cor software. Instantaneous water use efficiency via gas exchange (WUE$_{ge}$) was calculated as the ratio of $P_n$ to $T_r$.

**2.3 Plant material collection and leaf water soluble compounds extraction**

Recently-expanded, eight sun leaves per sapling were selected and homogenized in liquid nitrogen since the gas-exchange measurements accomplished. For the extraction of the water-soluble

compounds (WSCs) from the leaves (Gessler et al., 2004), 50 mg of ground leaves and 100 mg of PVPP (polyvinylpolypyrrolidone) were mixed and incubated in 1 mL double demineralized water for 60 min at 5℃ in a centrifuge tube. Each leaf was replicated two times. Two saplings per specie were chosen for each orthogonal treatment. The tubes containing above mixture were heated in 100℃ water for 3 min. Waiting for cooling to the room temperature, the supernatant of the mixture was centrifuged (12000 ×$g$ for 5 min, $g$ represents one gravity) and transferred 10 μL supernatant into tin capsule to be dried at 70℃. Folded capsules were then ready for $\delta^{13}C$ analysis of WSCs. The samples of WSCs from leaves were combusted in an elemental analyzer (EuroEA, HEKAtech GmbH, Wegberg, Germany) and analyzed with a mass spectrometer (DELTA$^{plus}$XP, ThernoFinnigan).

Carbon isotope signatures are expressed in δ-notation in parts per thousand, relative to the international Pee Dee Belemnite (PDB):

$$\delta^{13}C = \left(\frac{R_{sample}}{R_{standard}} - 1\right) \times 1000 \tag{1}$$

where $\delta^{13}C$ is the heavy isotope and $R_{sample}$ and $R_{sample}$ refer to the isotope ratio between the particular substance and the corresponding standard, respectively. The precision of the repeated measurements was 0.1 ‰.

**2.4 Isotopic calculation**

2.4.1 $^{13}C$ fractionation from the site of carboxylation to cytoplasm before sugars transportation

Based on the linear model developed by Farquhar and Sharkey (1982), the isotope discrimination, $\Delta$, is calculated as:

$$\Delta = \left(\delta^{13}C_a - \delta^{13}C_{WSC}\right)/\left(1 + \delta^{13}C_{WSC}\right) \tag{2}$$

where $\delta^{13}C_a$ is the isotope signature of ambient [$CO_2$] in chambers; $\delta^{13}C_{WSC}$ is the carbon isotopic composition of water soluble compounds extracted from leaves. The $C_i:C_a$ is determined by:

$$C_i:C_a = (\Delta - a)/(b - a) \tag{3}$$

where $C_i$ is the intercellular $CO_2$ concentration, and $C_a$ is the ambient $CO_2$ concentration in chambers; $a$ is the fractionation occurring $CO_2$ diffusion in still air (4‰) and $b$ refers to the discrimination during $CO_2$ fixation by ribulose 1,5- bisphosphate carboxylase/oxygenase (Rubisco) and internal diffusion (30‰). Instantaneous water use efficiency by gas-exchange measurements (WUE$_{ge}$) is calculated as:

$$WUE_{ge} = P_n:T_r = (C_a - C_i)/1.6\Delta e \tag{4}$$

where 1.6 is the diffusion ratio of stomatal conductance to water vapor to $CO_2$ in chambers and $\Delta e$ is the difference between $e_{lf}$ and $e_{atm}$ that represent the extra- and intra-cellular water vapor pressure, respectively:

$$\Delta e = e_{lf} - e_{atm} = 0.611 \times e^{17.502T/(240.97+T)} \times (1 - RH) \tag{5}$$

where $T$ and RH are the temperature and relative humidity on leaf surface, respectively. Combining Eqns. (2, 3 and 4), the instantaneous water use efficiency could be determined by the $\delta^{13}C_{WSC}$ of leaves, defined as WUE$_{cp}$:

$$WUE_{cp} = \frac{P_n}{T_r} = (1 - \varphi)(C_a - C_i)/1.6\Delta e = C_a(1 - \varphi)\left[\frac{b - \delta^{13}C_a + (b+1)\delta^{13}C_{WSC}}{(b-a)(1+\delta^{13}C_{WSC})}\right]/1.6\Delta e \tag{6}$$

where $\varphi$ is the respiratory ratio of leaf carbohydrates to other organs at night (0.3).

Then the [13]C fractionation from the site of carboxylation to cytoplasm before sugars transportation (total [13]C fractionation) can be estimated by the observed $\delta^{13}C$ of water soluble compounds from leaves ($\delta^{13}C_{WSC}$) and the modeled $\delta^{13}C$ calculated from gas-exchange ($\delta^{13}C_{model}$). The $\delta^{13}C_{model}$ is calculated from $\Delta_{model}$ from Eqn. (2). The $\Delta_{model}$ can be determined by Eqns. (3 and 4) as:

$$\Delta_{model} = (b - a)\left(1 - \frac{1.6\Delta eWUE_{ge}}{c_a}\right) + a \tag{7}$$

$$\delta^{13}C_{model} = \frac{c_a - \Delta_{model}}{1 + \Delta_{model}} \tag{8}$$

$$\text{Total } ^{13}C \text{ fractionation} = \delta^{13}C_{WSC} - \delta^{13}C_{model} \tag{9}$$

2.4.2 Methodology of calculating mesophyll conductance and estimating contribution of post-carboxylation fractionation

Actually, the carbon isotope discrimination is generated from the relative contribution of diffusion and carboxylation, reflected by the ratio of $CO_2$ concentration at the site of carboxylation ($C_c$) to that in the ambient environment surrounding plants ($C_a$). The carbon isotopic discrimination ($\Delta$) could be presented as (Farquhar et al. 1982):

$$\Delta = a_b\frac{c_a - c_s}{c_a} + a\frac{c_s - c_i}{c_a} + (e_s + a_l)\frac{c_i - c_c}{c_a} + b\frac{c_c}{c_a} - \frac{\frac{eR_D}{k} + f\Gamma_*}{c_a} \tag{10}$$

Where $C_a$, $C_s$, $C_i$, and $C_c$ indicate the $CO_2$ concentrations in the ambient environment, at the boundary layer of leaf, in the intercellular air spaces before entrancing into solution, and at the sites of carboxylation, respectively; $a_b$ is the fractionation for the $CO_2$ diffusion at the boundary layer (2.9‰); $e_s$ is the discrimination of $CO_2$ diffusion when $CO_2$ enters in solution (1.1‰, at 25 ℃); $a_l$ is the fractionation derived from diffusion in the liquid phase (0.7‰); $e$ and $f$ are carbon discrimination derived in dark respiration ($R_D$) and photorespiration, respectively; $k$ is the carboxylation efficiency, and $\Gamma^*$ is the $CO_2$ compensation point in the absence of dark respiration (Brooks and Farquhar,1985).

When the gas in the cuvette could be well stirred during measurements of carbon isotopic discrimination and gas exchange, the diffusion in the boundary layer could be neglected and Equation 10 could be shown:

$$\Delta = a\frac{c_a - c_i}{c_a} + (e_s + a_l)\frac{c_i - c_c}{c_a} + b\frac{c_c}{c_a} - \frac{\frac{eR_D}{k} + f\Gamma_*}{c_a} \tag{11}$$

There was no agreement about the value of $e$, although recent measurements estimated it as 0-4‰. Value of $f$ has been estimated ranging at 8-12‰ (Gillon and Griffiths, 1997; Igamberdiev et al., 2004; Lanigan et al., 2008). As the most direct factor, the value of $b$ would influence the calculation for $g_m$, had been thought to be close to 30‰ in higher plants (Guy et al., 1993).

The difference of $CO_2$ concentration between the substomatal cavities and the chloroplast is omitted while diffusion discrimination related with dark-respiration and photorespiration is negligible, Equation 11 could be simplified as:

$$\Delta_i = a + (b - a)\frac{c_i}{c_a} \tag{12}$$

Equation 12 presents the linear relationship between carbon discrimination and $C_i/C_a$ that is used normally in carbon isotopic fractionation. That underlines the subsequent comparison between the

expected $\Delta$ (originated from gas-exchange, $\Delta_i$, and those actually measured $\Delta_{obs}$), that is the $^{13}C$
fractionation from mesophyll conductance, could evaluate the differences of $CO_2$ concentration
between the intercellular air and the sites of carboxylation that generated by mesophyll resistance.
Consequently, $g_m$ can be estimated by performing the $\Delta_{obs}$ by isotope ratio mass spectrometry and
expected $\Delta_i$ from $C_i/C_a$ by gas exchange measurements.
Then the $^{13}C$ fractionation from mesophyll conductance is calculated by subtracting $\Delta_{obs}$ of
Equation 11 from $\Delta_i$ (Equation 12):
$$\Delta_i - \Delta_{obs} = (b - e_s - a_l)\frac{c_i - c_c}{c_a} + \frac{\frac{eR_D + f\Gamma^*}{k}}{c_a} \tag{13}$$
and the $P_n$ from the first Fick's law is presented by:
$$P_n = g_m(C_i - C_c) \tag{14}$$
Substitute Equation 14 into Equation 13 we obtain:
$$\Delta_i - \Delta_{obs} = (b - e_s - a_l)\frac{P_n}{g_m c_a} + \frac{\frac{eR_D + f\Gamma^*}{k}}{c_a} \tag{15}$$
$$g_m = \frac{(b - e_s - a_l)\frac{P_n}{c_a}}{(\Delta_i - \Delta_{obs}) - \frac{eR_D/k + f\Gamma^*}{c_a}} \tag{16}$$
In calculation of $g_m$, the respiratory and photorespiratory terms could be ignored or be given the
specific constant values. Here, $e$ and $f$ are assumed to be zero or be cancelled out in the calculation of
$g_m$.
Then Equation 16 can be transformed into:
$$g_m = \frac{(b - e_s - a_l)\frac{P_n}{c_a}}{\Delta_i - \Delta_{obs}} \tag{17}$$
Therefore, the contribution of post-carboxylation fractionation could be estimated by:
$Contribution\ of\ post - $carboxylation $fractionation =$
$\frac{(\text{Total } ^{13}C \text{ fractionation} - \text{fractionation from mesophll conductance})}{\text{Total } ^{13}C \text{ fractionation}} \times 100\%$ $\tag{18}$
**3 Results**
**3.1 Foliar gas exchange measurements**
Saplings of *P. orientalis* and *Q. variabilis* were exposed to the orthogonal treatments. When SWC
increased, $P_n$, $g_s$ and $T_r$ in *P. orientalis* and *Q. variabilis* peaked at 70%−80% of FC or/and 100% FC
(Fig. 2). The $C_i$ in *P. orientalis* rose as SWC increased, while it peaked at 60%−70% of FC and
declined thereafter with increased SWC in *Q. variabilis*. The capacity of carbon uptake and $C_i$ were
improved significantly by elevated [$CO_2$] at any given SWC for two species ($p < 0.5$). Furthermore,
greater increments of $P_n$ in *P. orientalis* were found at 50%−70% of FC from $C_{400}$ to $C_{800}$, which was at
35%−45% of FC in *Q. variabilis*. As the water stress was alleviated (at 70%−80% of FC and 100% FC),
the reduction of $g_s$ in *P. orientalis* was more pronounced with elevated [$CO_2$] at a given SWC ($p < 0.01$).
Nevertheless, $g_s$ of *Q. variabilis* in $C_{400}$, $C_{500,}$ and $C_{600}$ was significantly higher than that in $C_{800}$ at
50%−80% of FC ($p < 0.01$). Coordinated with $g_s$, $T_r$ of two species in $C_{400}$ and $C_{500}$ was significantly

higher than that in $C_{600}$ and $C_{800}$ except for 35%−60% of FC ($p<0.01$, Figs. 2g and 2h). Larger $P_n$, $g_s$, $C_i$ and $T_r$ of $Q.$ $variabilis$ was significantly presented than that of $P.$ $orientalis$ ($p<0.01$, Fig. 2).

**3.2 $\delta^{13}C$ of water-soluble compounds in leaves**

After the observations of the photosynthetic traits in two species, the same leaf was frozen immediately and the water-soluble compounds (WSCs) were extracted for all orthogonal treatments. The carbon isotope composition of WSCs ($\delta^{13}C_{WSC}$) of two species both increased as soil moistened (Figs. 3a and 3b, $p<0.01$). The average ($\pm$ SD) $\delta^{13}C_{WSC}$ of $P.$ $orientalis$ and $Q.$ $variabilis$ ranged from -27.44 $\pm$ 0.155‰ to -26.71 $\pm$ 0.133‰, and from -27.96 $\pm$ 0.129‰ to -26.49 $\pm$ 0.236‰, respectively. Similarly with the photosynthetic capacity varying with increased SWC, average $\delta^{13}C_{WSC}$ of two species reached their maxima at 70%−80% of FC. Together with the gradual enrichment of [$CO_2$], average $\delta^{13}C_{WSC}$ in two species declined while [$CO_2$] exceeded 600 ppm ($p<0.01$). Except for $C_{400}$ at 50%−100% of FC, $\delta^{13}C_{WSC}$ of $P.$ $orientalis$ was significantly larger than that of $Q.$ $variabilis$ in any [$CO_2$] $\times$ SWC treatment ($p<0.01$, Fig. 3).

**3.3 Estimations of $WUE_{ge}$ and $WUE_{cp}$**

Figure 4a showed that increments of $WUE_{ge}$ in $P.$ $orientalis$ under severe drought (i.e., 35%−45% of FC) were highest at any given [$CO_2$], ranging from 90.70% to 564.65%. The $WUE_{ge}$ in $P.$ $orientalis$ decreased as SWC increased, while they increased as [$CO_2$] increased. Differing from variation in $WUE_{ge}$ of $P.$ $orientalis$ with soil moistened, $WUE_{ge}$ in $Q.$ $variabilis$ were improved slightly at 100% FC in $C_{600}$ or $C_{800}$ (Fig. 4b). The maximum of $WUE_{ge}$ thus occurred at 35%−45% of FC in $C_{800}$ among all orthogonal treatments for $P.$ $orientalis$; this was also observed in $Q.$ $variabilis$. Furthermore, elevated [$CO_2$] enhanced the $WUE_{ge}$ of $Q.$ $variabilis$ clearly at any SWC except that at 60%−80% of FC. Thirty-two saplings of $P.$ $orientalis$ had greater $WUE_{ge}$ than did $Q.$ $variabilis$ between the same [$CO_2$] $\times$ SWC treatments ($p<0.5$).

The instantaneous water use efficiency could be determined from Eqn. (6) by the $\delta^{13}C_{WSC}$ of leaves of two species, defined as $WUE_{cp}$. As illustrated in Fig. 5a, $WUE_{cp}$ of $P.$ $orientalis$ in $C_{600}$ or $C_{800}$ climbed up as water stress alleviated beyond 50%−60% of FC, as well as that in $C_{400}$ or $C_{500}$ while SWC exceeding 60%−70% of FC. $Q.$ $variabilis$ exhibited no uniform trend of $WUE_{cp}$ with soil wetting (Fig. 5b). Except for $C_{400}$, $WUE_{cp}$ of $Q.$ $variabilis$ decreased abruptly at 50%−60% of FC, and then rose as soil moisture improved in $C_{500}$, $C_{600}$, and $C_{800}$. In contrast to the results of $WUE_{ge}$ in two species, $WUE_{cp}$ of $Q.$ $variabilis$ was more pronounced than that of $P.$ $orientalis$ among all orthogonal treatments.

**3.4 $^{13}C$ fractionation from the site of carboxylation to cytoplasm before sugars transportation**

We evaluated the total $^{13}C$ fractionation from the site of carboxylation to cytoplasm by gas exchange measurements and $\delta^{13}C$ of water-soluble compounds from leaf (Table 1), which can retrace $^{13}C$ fractionation before carboxylation transport to the twig. Comparing $\delta^{13}C_{WSC}$ with $\delta^{13}C_{model}$ from Eqns. (4, 7–9), total $^{13}C$ fractionation of $P.$ $orientalis$ ranged from 0.0328‰ to 0.0472‰, which was smaller than that of $Q.$ $variabilis$ (0.0384‰ to 0.0466‰). The total fractionations of $P.$ $orientalis$ were magnified with soil wetting especially that reached 35%−80% of FC from $C_{400}$ to $C_{800}$ (increased by 21.30%−42.04%). The total fractionation under $C_{400}$ and $C_{500}$ were amplified as SWC increased until 50%−60% of FC in $Q.$ $variabilis$, whereas it was increased at 50%−80% of FC and decreased at 100% FC under $C_{600}$ and $C_{800}$. Elevated [$CO_2$] enhanced the average total fractionation of $P.$ $orientalis$, while those of $Q.$ $variabilis$ declined sharply from $C_{600}$ to $C_{800}$. Total $^{13}C$ fractionation in $P.$ $orientalis$ increased faster than did those of $Q.$ $variabilis$ with increased soil moisture.

### 3.5 $g_m$ imposed on the interaction of $CO_2$ concentration and water stress

According to comparison between online leaf $\delta^{13}C_{WSC}$ and the values of gas exchange measurements, $g_m$ over all treatments was presented in Fig. 6 (Eqns. 10–17). Significant increment trend of $g_m$ was observed with water stress alleviated in *P. orientalis*, ranging from 0.0091−0.0690 mol $CO_2$ $m^{-2}$ $s^{-1}$ ($p<0.5$), which reached the maximum at 100% FC under a given [$CO_2$]. Yet increases in $g_m$ of *Q. variabilis* with increasing SWC become unremarkable except that under $C_{400}$. With $CO_2$ concentration elevated, $g_m$ of two species was increased in different degrees. Comparing with *P. orientalis* under $C_{400}$, $g_m$ was increased gradiently and reached its maximum under $C_{800}$ at 35%−60% of FC and 100% FC ($p<0.5$), however, that was maximized under $C_{600}$ ($p<0.5$) and slipped down under $C_{800}$ at 60%−80% of FC. The maximum increment of $g_m$ (8.2%−58.4%) occurred at $C_{800}$ at any given SWC in *Q. variabilis*. It is evidently shown that $g_m$ of *Q. variabilis* was larger than that of *P. orientalis* in the same treatment.

### 3.6 The contribution of post-carboxylation fractionation

Here, the difference between $\Delta_i$ and $\Delta_{obs}$ presented the $^{13}C$ fractionation derived from mesophyll conductance. So the post-photosynthetic fractionation after carboxylation can be calculated by subtracting the fractionation derived from mesophyll conductance from the total $^{13}C$ fractionation that is generated from the site of carboxylation to cytoplasm before sugars transportation (Table 1). The fractionation from $g_m$ had less contribution on total $^{13}C$ fractionation than that from synthesis of sugars belonging to post-carboxylation fractionation in any given treatment (Table 1). The contributions of fractionation from $g_m$ in two species were illustrated different variations with soil water increasing, which declined at 50%−80% of FC and rose up at 100% FC in *P. orientalis*, yet it was shown increasing with water stress alleviated at 50%−80% of FC and then decreased at 100% FC in *Q. variabilis*. Nevertheless, the fractionations from synthesis of sugars in leaf and these contributions to total fractionation were all increased as soil moistened in two species. Considering the effects of enriched [$CO_2$] on $g_m$, fractionation from $g_m$ reached its average peak under $C_{600}$ in *P. orientalis*, which occurred under $C_{800}$ with *Q. variabilis*. Post-carboxylation fractionations were increased along with [$CO_2$] increased in *P. orientalis*, which reached those maxima under $C_{600}$ and then slipped down under $C_{800}$ differing in degrees.

### 3.7 Relationship between $g_s$, $g_m$ and total $^{13}C$ fractionation

Total $^{13}C$ fractionation after carboxylation may be correlated with the resistances derived from stomata and mesophyll cells. Here, we performed linear regressions between $g_s/g_m$ and total $^{13}C$ fractionation for *P. orientalis* and *Q. variabilis*, respectively (Fig. 7 and 8). It was apparent that total $^{13}C$ fractionation was linearly dependent on the $g_s$ ($p<0.01$) that controls the exchange of $CO_2$ and $H_2O$, and responds to environmental variation. Subsequently, the linear relationships between $g_m$ and total $^{13}C$ fractionation were shown ($p<0.01$), which reflected the variation of $CO_2$ concentration through the chloroplast was correlated with carbon discrimination happened after photosynthesis in the leaf.

## 4 Discussion

### 4.1 Photosynthetic traits

The exchange of $CO_2$ and water vapor via stomata is modulated in part by the soil/leaf water potential (Robredo et al., 2010). Saplings of *P. orientalis* reached their maxima of $P_n$ and $g_s$ at 70%−80% of FC irrespective of [$CO_2$] treatments. As SWC exceeded this water threshold, elevated $CO_2$ would cause a greater reduction in $g_s$, as has been reported for barley and wheat (Wall et al., 2011). The

decrease of $g_s$ responding to elevated $[CO_2]$ could be mitigated by the coupling effects of soil wetting. In addition, $C_i$ of *Q. variabilis* peaked at 60%–70% of FC and followed declines as soil moisture increased (Wall et al., 2006; Wall et al., 2011). This is interpreted as stomata having the tendency to maintain a constant $C_i$ or $C_i/C_a$ when ambient $[CO_2]$ increased, which would determine the $CO_2$ used directly in chloroplast (Yu et al., 2010). On the basis of theories (Farquhar and Sharkey, 1982) and common experimental technologies (Xu, 1997), this could be explained as the stomatal limitation. However, $C_i$ of *P. orientalis* was increased considerably while SWC exceeded 70%–80% of FC, as found by Mielke et al. (2000). One factor that can account for that is plants close their stomata to reduce the loss of water during the synthesis of organic matter, simultaneously decreasing the availability of $CO_2$ and generating respiration of organic matter (Robredo et al., 2007). Another explanation is the limited root volume in potted experiments may not be able to absorb sufficient water to support full growth of shoots (Leakey et al., 2009; Wall et al., 2011). In our study, the coupling of increasing $[CO_2]$ may cause nonstomatal limitation as SWC exceeding the threshold (70%–80% of FC), i.e., accumulation of nonstructural carbohydrates in leaf tissue that induces mesophyll-based and/or biochemical-based transient inhibition of photosynthetic capacity (Farquhar and Sharkey, 1982). Xu and Zhou (2011) developed a five-level SWC gradient to examine the effect of water on the physiological characteristics of perennial *Leymus chinensis*, demonstrating that there was a clear irrigation maximum of SWC below which the plant could manage itself to adjust changing environment. Miranda Apodaca et al. (2015) also concluded that, in suitable water conditions, elevated $CO_2$ augmented $CO_2$ assimilation in herbaceous plants.

The $P_n$ of two species increased with elevated $[CO_2]$ in our study, similarly with the results from $C_3$ woody plants (Kgope et al., 2010). Furthermore, increasing $[CO_2]$ alleviated severe drought and heavy irrigation, which suggests that photosynthetic inhibition produced by water stress or excess may be mediated by increased $[CO_2]$ (Robredo et al., 2007; Robredo et al., 2010) and meliorate the adverse effects of drought stress by decreasing plant transpiration (Kirkham, 2016; Kadam et al., 2014; Miranda Apodaca et al., 2015; Tausz Posch et al., 2013).

### 4.2 Differences between WUE$_{ge}$ and WUE$_{cp}$

The increments of WUE$_{ge}$ in *P. orientalis* and *Q. variabilis* that resulted from the combination of an increase in $P_n$ and decrease in $g_s$, followed by a reduction in $T_r$ (Figs. 2a, 2g, 2b and 2h), were also demonstrated by Ainsworth and McGrath (2010). Combining $P_n$ and $T_r$ of two species in the same treatment, lower WUE$_{ge}$ in *Q. variabilis* is obtained due to its physiological and morphological traits, such as larger leaf area, rapid growth, and higher stomatal conductance than that of *P. orientalis* (Adiredjo et al., 2014). Medlyn et al. (2001) reported that the stomatal conductance of broadleaved species is more sensitive to elevated $CO_2$ concentrations than in conifers. Moreover, there has been no consensus on the patterns of iWUE with related SWC at the leaf level, although some have discussed this topic (Yang et al., 2010). The WUE$_{ge}$ of *P. orientalis* and *Q. variabilis* was enhanced with soil drying, as presented by Parker and Pallardy (1991), DeLucia and Heckathorn (1989), Reich et al. (1989), and Leakey (2009).

Bögelein et al. (2012) confirmed that WUE$_{cp}$ was more consistent with daily mean WUE$_{ge}$ than WUE$_{phloem}$. The WUE$_{cp}$ of two species demonstrated similar variation to those $\delta^{13}C_{WSC}$, which differentiated with that of WUE$_{ge}$. Pons et al. (2009) reviewed that $\Delta$ of leaf soluble sugar is coupled with environmental dynamics over a period ranging from a few hours to 1–2 d. The WUE$_{cp}$ of our materials could respond to $[CO_2] \times$ SWC treatments over cultivated days, whereas WUE$_{ge}$ is characterized as the instantaneous physiology of plants to conditions. In addition, species-specific

$\delta^{13}C_{WSC}$ were observed in the same environmental treatment. Consequently, WUE$_{cp}$ and WUE$_{ge}$ have
different variable curves according to different treatments.

## 4.3 The influence of mesophyll conductance on the fractionation after carboxylation

The consensus has been reached that the routine of $CO_2$ diffusion into photosynthetic site includes
two main procedures, which are $CO_2$ moving from ambient air surrounding the leaf ($C_a$) to the
sub-stomatic cavities ($C_i$) through stomata, and from there to the site of carboxylation within the
chloroplast stroma ($C_c$) of leaf mesophyll. The latter procedure of diffusion is defined as mesophyll
conductance ($g_m$) (Flexas et al., 2008). Moreover, $g_m$ has been identified to coordinate with
environmental factors more faster than stomatal conductance (Galmés et al., 2007; Tazoe et al., 2011;
Flexas et al., 2007). During our 7-day cultivations of SWC $\times$ [$CO_2$], $g_m$ was increased and WUE$_{ge}$ was
decreased as soil moistened, which has been verified that $g_m$ as an important factor, could improve
WUE under drought pretreatment (Han et al., 2016). There has been a dispute how $g_m$ responds to the
fluctuation of $CO_2$ concentration. Terashima *et al.* (2006) have confirmed that $CO_2$ permeable
aquaporin, located in the plasma membrane and inner envelope of chloroplasts (Uehlein et al. 2008),
could regulate the change of $g_m$. In our study, $g_m$ is specific-special to the gradient of [$CO_2$]. The $g_m$ of
*P. orientalis* was significantly decreased by 9.08%-44.42% from C$_{600}$ to C$_{800}$ at 60%-80% of FC, being
similar to the results obtained by Flexas *et al.* (2007). Although larger $g_m$ of *Q. variabilis* under C$_{800}$
was observed, it made almost no difference.
Furthermore, $g_m$ contributed to total $^{13}C$ fractionation that followed the carboxylation while
photosynthate has not been transported to the twigs of sapling. The $^{13}C$ fractionation of $CO_2$ from the
air surrounding leaf to sub-stomatal cavity may be simply considered, whereas the fractionation
induced by mesophyll conductance from sub-stomatic cavities to the site of carboxylation in the
chloroplast cannot be neglected (Pons et al., 2009; Cano et al., 2014). As estimating the
post-carboxylation fractionation, carbon isotope fractionation derived from $g_m$ must be subtracted from
the total $^{13}C$ fractionation (the difference between $\delta^{13}C_{WSC}$ and $\delta^{13}C_{model}$), which was closely associated
with $g_m$ (Fig. 8, *p*=0.01 or *p*<0.01). Similar variations of $^{13}C$ fractionations derived from $g_m$ were
presented with that of $g_m$ under orthogonal treatments on Table 1.

## 4.4 Post-carboxylation fractionation generated before photosynthate leaving leaves

Photosynthesis, a biochemical and physiological process (Badeck et al., 2005), is characterized by
discrimination against $^{13}C$, which leaves an isotopic signature in the photosynthetic apparatus. There is
a classic review of the carbon-fractionation in leaves that covers the significant aspects of
photosynthetic carbon isotope discrimination (Farquhar et al., 1989). The
post-carboxylation/photosynthetic fractionation associated with the metabolic pathways of
non-structural carbohydrates (NSC; defined here as soluble sugars + starch) within leaves, and
fractionation during translocation, storage, and remobilization prior to tree ring formation remain
unclear (Epron et al., 2012; Gessler et al., 2014; Rinne et al., 2016). The synthetic processes of sucrose
and starch before transportation to the twig are within the domain of post-carboxylation fractionation
generated in leaves. Hence, we hypothesized that the $^{13}C$ fractionation might exist. When we finished
the leaf gas-exchange measurements, the leaf samples were collected immediately to determine the
$\delta^{13}C$ of water-soluble compounds ($\delta^{13}C_{WSC}$). Presumably, the $^{13}C$ fractionation generated in the
synthetic processes of sucrose and starch was approximately contained within the $^{13}C$ fractionation
from the site of carboxylation to cytoplasm before sugars transportation as total $^{13}C$ fractionation.
When comparing $\delta^{13}C_{WSC}$ with $\delta^{13}C_{obs}$, total $^{13}C$ fractionation of *P. orientalis* ranged from 0.0328‰ to

0.0472‰, less than that of *Q. variabilis* (from 0.0384‰ to 0.0466‰). The post-carboxylation fractionation contributed 75.30%-98.9% on total $^{13}$C fractionation, which was determined by subtracting the fractionation of mesophyll conductance from total $^{13}$C fractionation. Recently, Gessler et al. (2004) reviewed the environmental drivers of variation in photosynthetic carbon isotope discrimination in terrestrial plants. Total $^{13}$C fractionation of *P. orientalis* was enhanced by soil moistening, consistent with that of *Q. variabilis*, except at 100% FC. The $^{13}$C isotope signature of *P. orientalis* was dampened by elevated [$CO_2$]. Yet, $^{13}$C-depletion was weakened in *Q. variabilis* at $C_{600}$ and $C_{800}$. Linear regressions between $g_s$ and total $^{13}$C fractionation indicated that the post-carboxylation fractionation in leaves depended on the variation of $g_s$ and stomata aperture correlated with environmental change.

## 5 Conclusions

Through orthogonal treatments of four [$CO_2$]s × five SWCs, $WUE_{cp}$ calculated by $\delta^{13}$C of water-soluble compound and $WUE_{ge}$ derived from simultaneous leaf gas exchange were estimated to differentiate the $\delta^{13}$C signal variation before leaf-exported translocation of primary assimilates. The influence of mesophyll conductance on the difference of $^{13}$C fractionation between the sub-stomatic cavities and the ambient environment need to be considered, while testing the hypothesis that the post-carboxylation will contribute on the $^{13}$C fractionation from the site of carboxylation to cytoplasm before sugars transportation. In response to the interactive effects of [$CO_2$] and SWC, $WUE_{ge}$ of two species both decreased with soil moistening, and increased with elevated [$CO_2$] at 35%−80% of FC. We concluded that relative soil drying, coupled with elevated [$CO_2$], could improve $WUE_{ge}$ by strengthening photosynthetic capacity and reducing transpiration. $WUE_{ge}$ of *P. orientalis* was significantly greater than that of *Q. variabilis*, while the opposite was the case for $WUE_{cp}$ in two species. Mesophyll conductance and post-carboxylation were manifested both contributing on the $^{13}$C fractionation from the site of carboxylation to cytoplasm before sugars transportation determined by gas-exchange and carbon isotopic measurements. Rising [$CO_2$] and/or soil moistening generated increasing disparities between $\delta^{13}C_{WSC}$ and $\delta^{13}C_{model}$ in *P. orientalis*; nevertheless, the differences between $\delta^{13}C_{WSC}$ and $\delta^{13}C_{model}$ in *Q. variabilis* increased as [$CO_2$] being less than 600 ppm and/or water stress alleviated. Total $^{13}$C fractionation in leaf was linearly dependent on $g_s$. With respect to carbon isotope fractionation in post-carboxylation and transportation processes, we cannot neglect that the $^{13}$C fractionation derived from the synthesis of sucrose and starch were influenced inevitably by environmental changes. Thus, clear description of the magnitude and environmental dependence of apparent post-carboxylation fractionation are worth our attention in photosynthetic fractionation.

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

**Author contribution**

Na Zhao and Yabing He collected field samples, and performed the experiment. Na Zhao engaged in data analysis and writing this paper. Ping Meng proposed the suggestions on the theory and practice of experiment. Xinxiao Yu revised the paper and contributed to edit the manuscript.

*Acknowledgements*. We would like to thank Beibei Zhou and Yuanhai Lou for kind supports in the collection of materials and management of saplings. We are grateful to anonymous reviewers for constructive suggestions of this manuscript. Due to the limitation of space allowed, we cited a part of literatures involving this study area and apologized for authors whose work has not been cited. All authors acknowledges support of National Natural Science Foundation of China (grant No. 41430747).


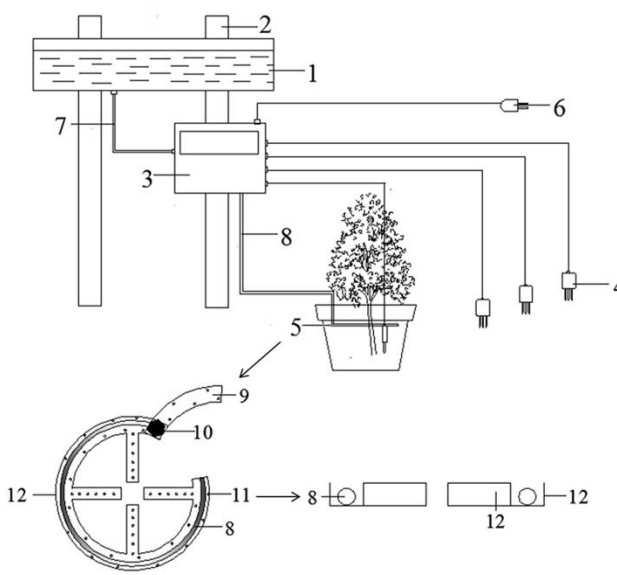

1. water storage tank
2. holder
3. controller
4. soil moisture sensors
5. drip irrigation component
6. AC power supply
7. main water pipe
8. distributed water pipe
9. movable annular body
10. spindle of the annular body
11. drainage holes of drip irrigation
12. steel supporting ring


**Figure 1.** Structural diagram of the device for automatic drip irrigation
Arabic numerals indicate the individual parts of the automatic drip irrigation device (No. 1–7). The
lower-left corner of this figure presents the detailed schematic for the drip irrigation components (No.
8–12). The lower-right corner of this figure shows the schematic for the drip irrigation component in
profile.








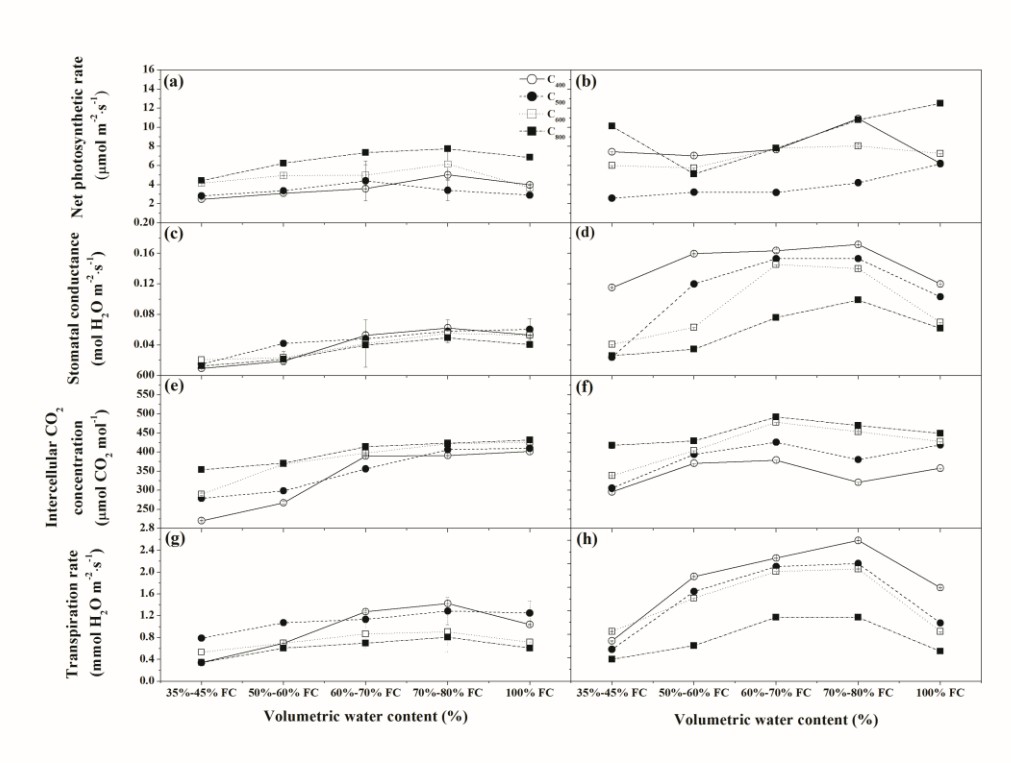

**Figure 2.** Net photosynthetic rates ($P_n$, μmol m$^{-2}$ s$^{-1}$, a and b), stomatal conductance ($g_s$, mol H$_2$O m$^{-2}$
s$^{-1}$, c and d), intercellular CO$_2$ concentration ($C_i$, μmol CO$_2$ mol$^{-1}$, e and f), and transpiration rates ($T_r$,
mmol H$_2$O m$^{-2}$ s$^{-1}$, g and h) of *P. orientalis* and *Q. variabilis* for four CO$_2$ concentrations × five soil
volumetric water contents. Means ±SDs, n = 32.

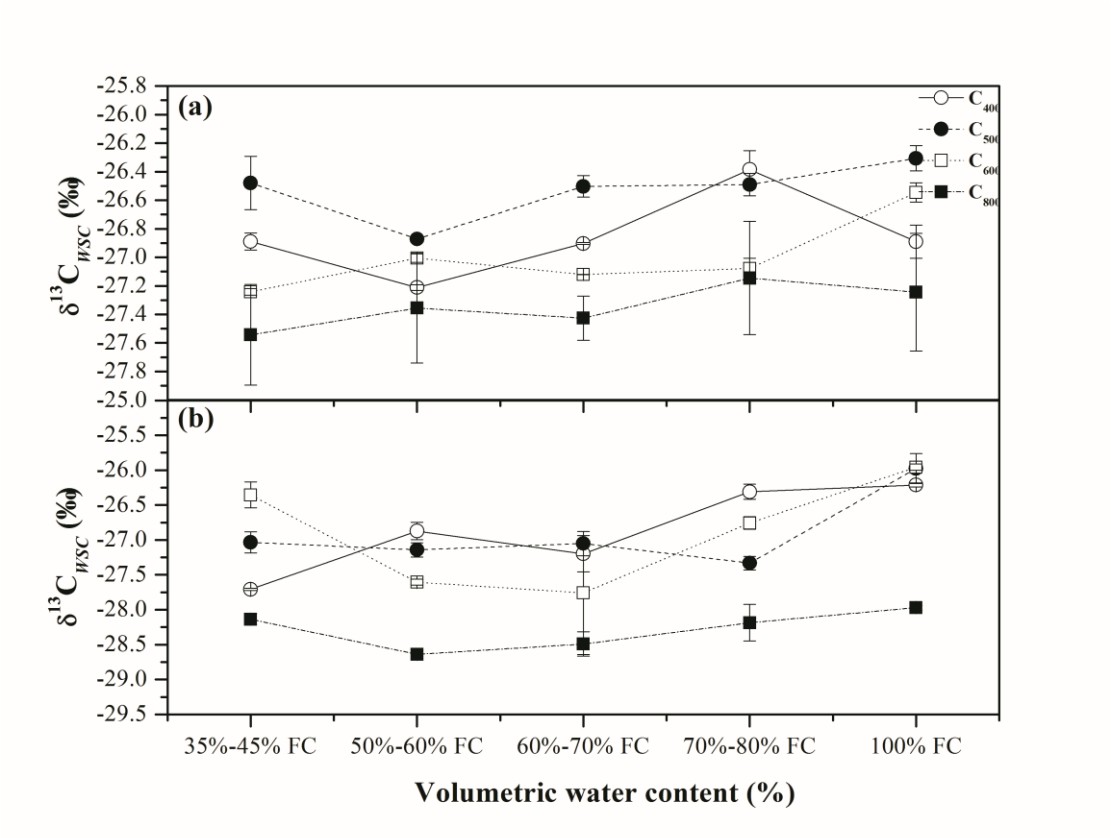

**Figure 3.** Carbon isotope composition of water-soluble compounds ($\delta^{13}C_{WSC}$) extracted from leaves of
*P. orientalis* (a) and *Q. variabilis* (b) for four $CO_2$ concentrations × five soil volumetric water contents.
Means ± SDs, n = 32.

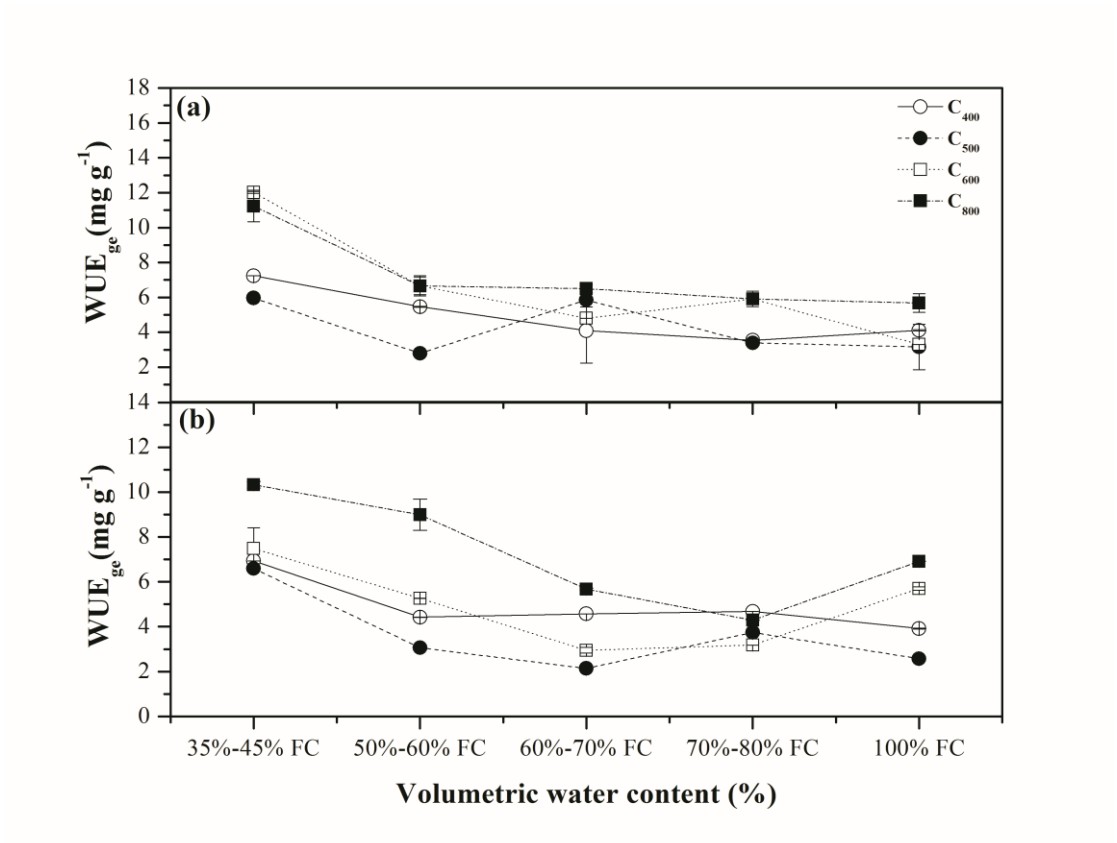

**Figure 4.** Instantaneous water use efficiency through gas exchange measurements (WUE$_{ge}$) for leaves
of *P. orientalis* (a) and *Q. variabilis* (b) for four $CO_2$ concentrations × five soil volumetric water
contents. Means ±SDs, n = 32.

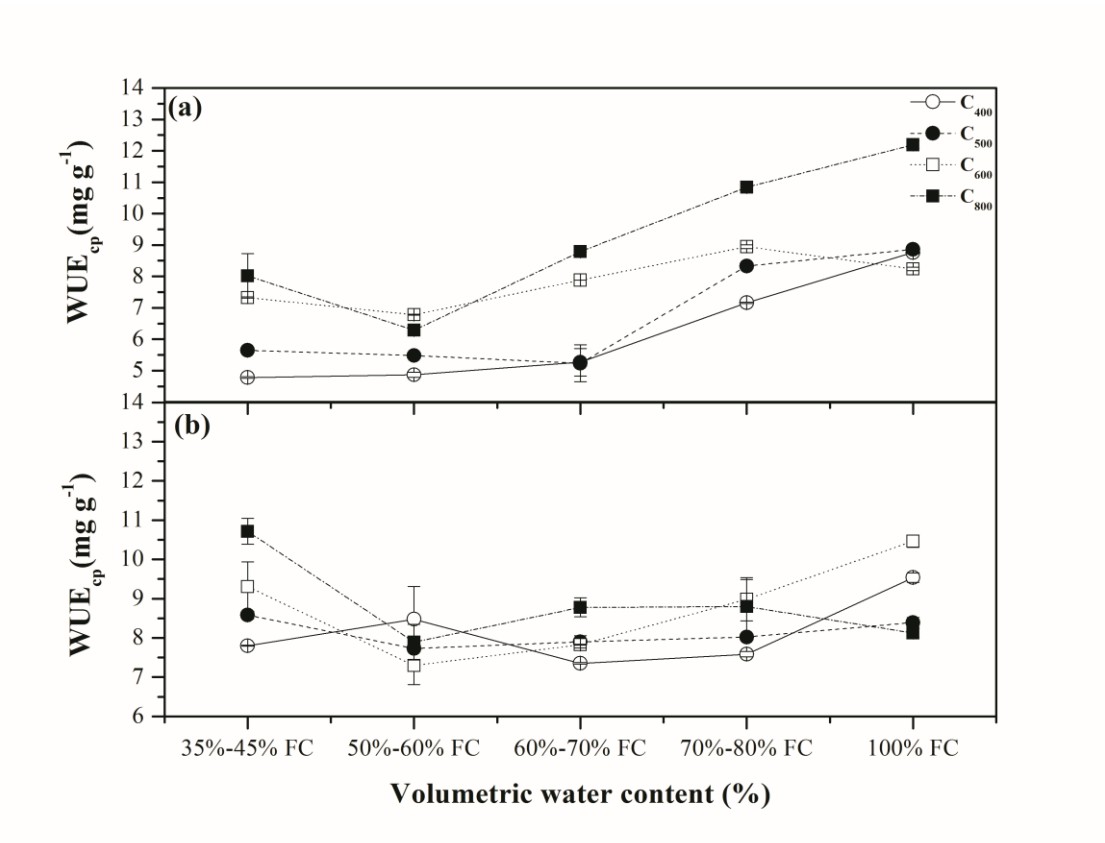

**Figure 5.** Instantaneous water use efficiency estimated by $\delta^{13}C$ of water-soluble compounds (WUE$_{cp}$)
from leaves of *P. orientalis* (a) and *Q. variabilis* (b) for four $CO_2$ concentrations ×five soil volumetric
water contents. Means ±SDs, n = 32.

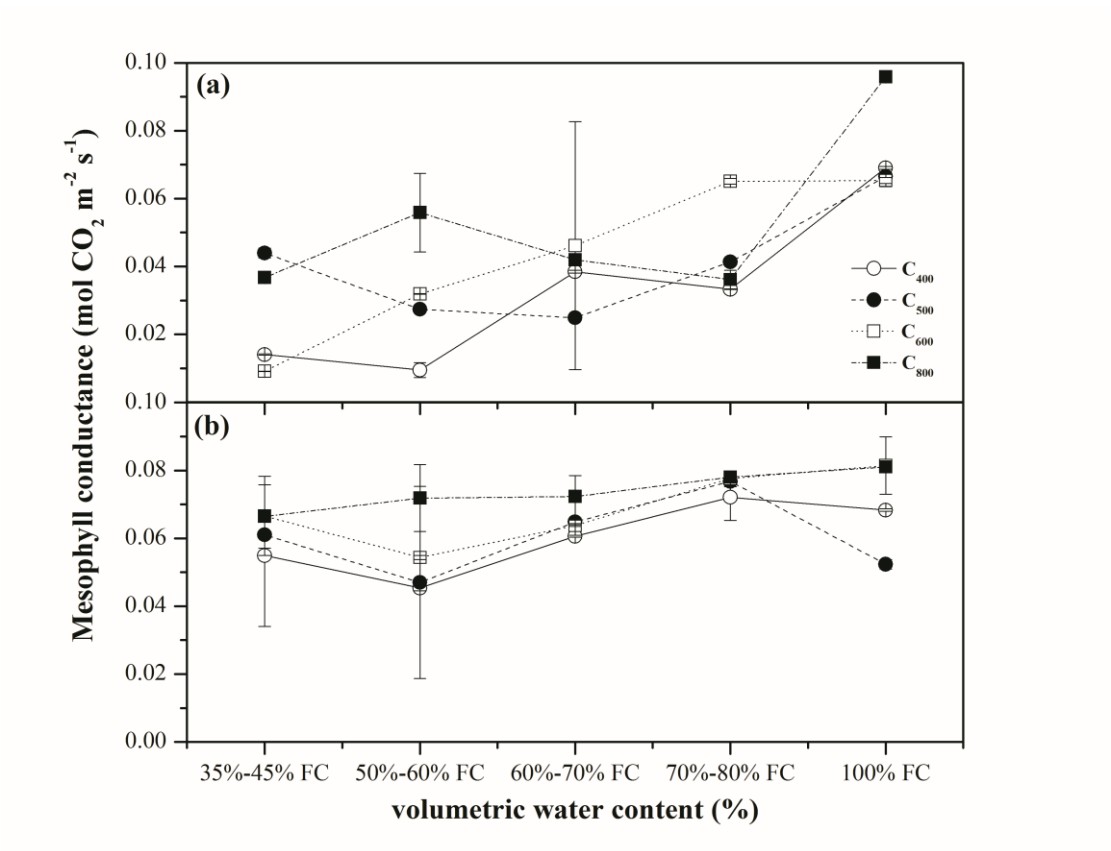

**Figure 6.** Mesophyll conductance of *P. orientalis* (a) and *Q. variabilis* (b) for four $CO_2$ concentrations
×five soil volumetric water contents. Means ±SDs, n = 32.

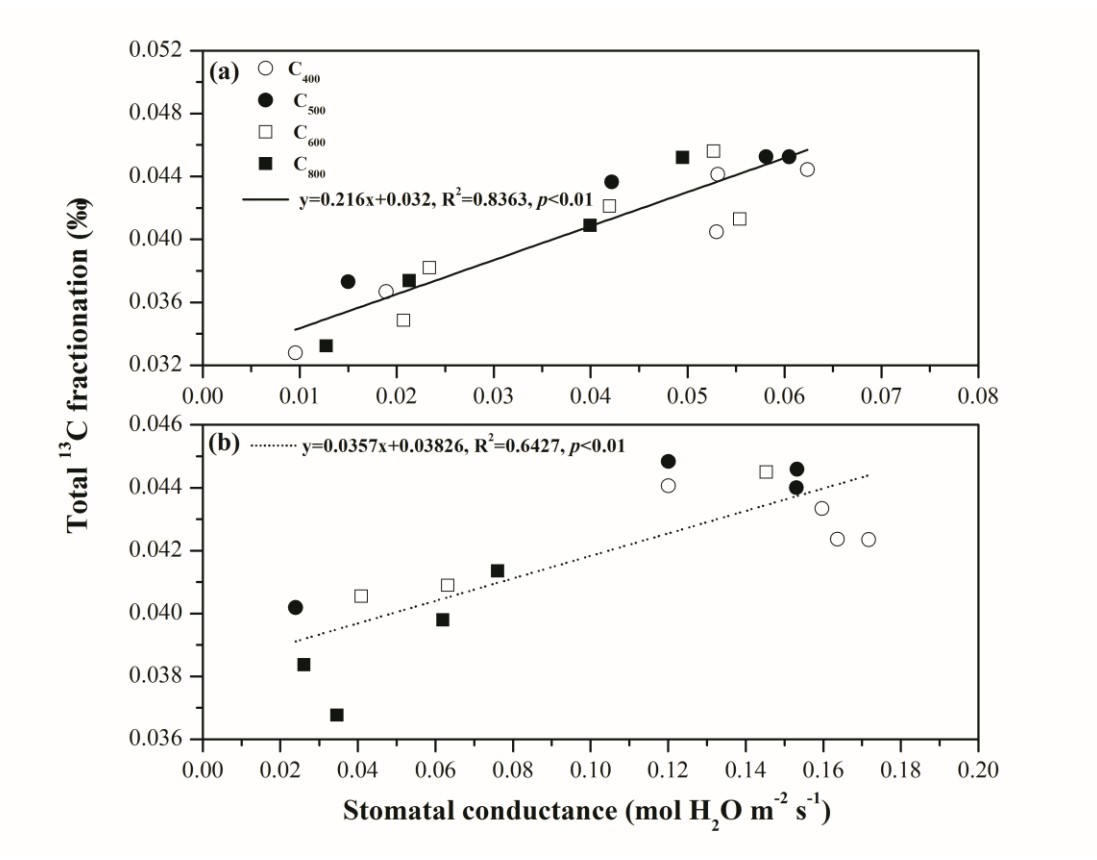


**Figure 7.** Regression between stomatal conductance and total [13]C fractionation of *P. orientalis* (a) and
*Q. variabilis* (b) for four $CO_2$ concentrations ×five soil volumetric water contents ($p$=0.01, n = 32).

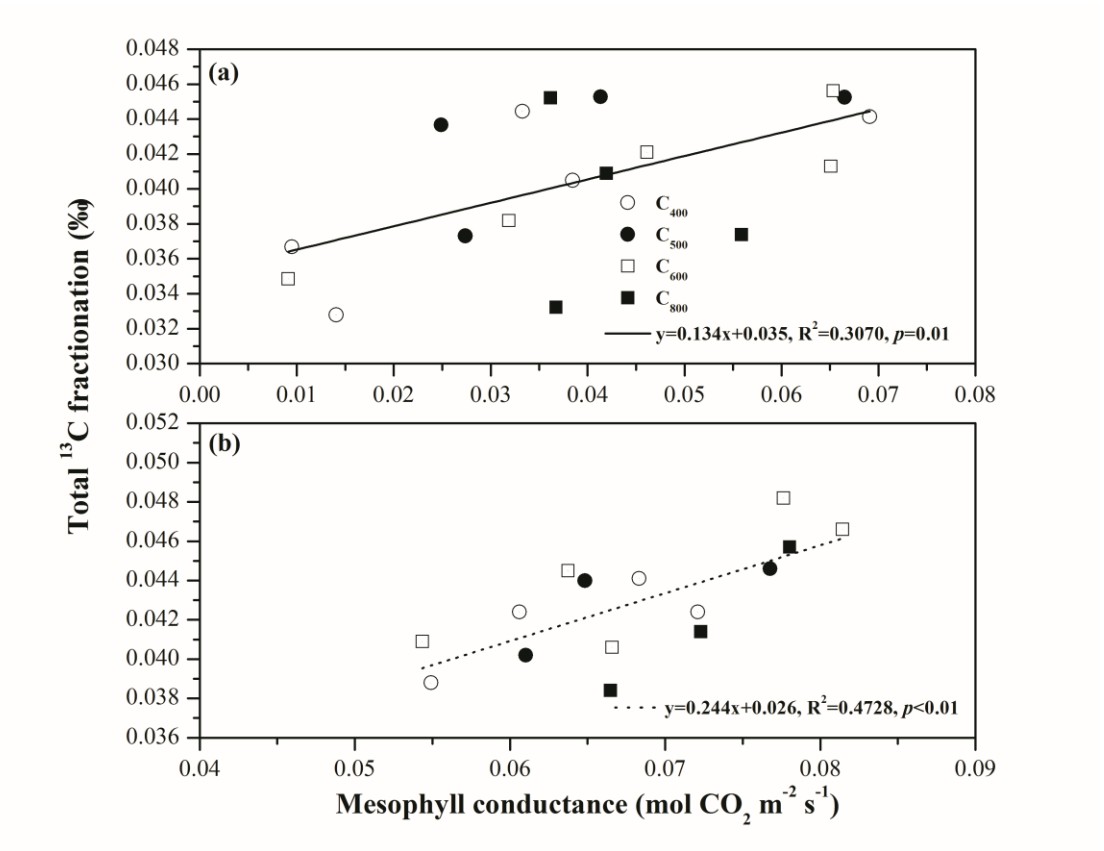


**Figure 8.** Regression between mesophyll conductance and total $^{13}$C fractionation of *P. orientalis* (a)
and *Q. variabilis* (b) for four $CO_2$ concentrations × five soil volumetric water contents ($p$=0.01, n =

746 32).

**Table 1.** Carbon-13 isotope fractionation of *P. orientalis* and *Q. variabilis* for four $CO_2$ concentrations $\times$ five soil volumetric water contents.

| Species | SWC (of FC) | Total $^{13}$C fractionation (‰) | 400 | 500 | 600 | 800 | $^{13}$C fractionation (‰) | 400 | 500 | 600 | 800 | $^{13}$C fractionation (‰) | 400 | 500 | 600 | 800 |
|---|---|---|---|---|---|---|---|---|---|---|---|---|---|---|---|---|
| *P. orientalis* | 35%–45% | | 0.0328 | 0.0373 | 0.0349 | 0.0332 | | 0.0081 | 0.0030 | 0.0034 | 0.0072 | | 0.0247 | 0.0343 | 0.0315 | 0.0260 |
| | 50%–60% | | 0.0367 | 0.0437 | 0.0382 | 0.0374 | | 0.0018 | 0.0058 | 0.0094 | 0.0004 | | 0.0349 | 0.0379 | 0.0288 | 0.0370 |
| | 60%–70% | | 0.0405 | 0.0366 | 0.0421 | 0.0409 | | 0.0018 | 0.0050 | 0.0026 | 0.0007 | | 0.0387 | 0.0316 | 0.0395 | 0.0402 |
| | 70%–80% | Total $^{13}$C fractionation (‰) | 0.0444 | 0.0453 | 0.0413 | 0.0452 | | 0.0044 | 0.0052 | 0.0103 | 0.0013 | | 0.0400 | 0.0401 | 0.0310 | 0.0439 |
| | 100% | | 0.0441 | 0.0453 | 0.0456 | 0.0472 | Mesophyll conductance | 0.0057 | 0.0040 | 0.0025 | 0.0039 | Post-photosynthesis | 0.0384 | 0.0413 | 0.0431 | 0.0433 |
| *Q. variabilis* | 35%–45% | | 0.0388 | 0.0402 | 0.0406 | 0.0384 | | 0.0007 | 0.0025 | 0.0006 | 0.0091 | | 0.0381 | 0.0377 | 0.0400 | 0.0293 |
| | 50%–60% | | 0.0433 | 0.0448 | 0.0409 | 0.0368 | | 0.0061 | 0.0084 | 0.0023 | 0.0018 | | 0.0372 | 0.0364 | 0.0386 | 0.0350 |
| | 60%–70% | | 0.0424 | 0.0440 | 0.0445 | 0.0414 | | 0.0066 | 0.0086 | 0.0078 | 0.0041 | | 0.0358 | 0.0354 | 0.0367 | 0.0373 |
| | 70%–80% | | 0.0424 | 0.0446 | 0.0482 | 0.0457 | | 0.0034 | 0.0016 | 0.0074 | 0.0028 | | 0.0390 | 0.0430 | 0.0408 | 0.0429 |
| | 100% | | 0.0441 | 0.0466 | 0.0466 | 0.0398 | | 0.0027 | 0.0076 | 0.0022 | 0.0125 | | 0.0414 | 0.0390 | 0.0444 | 0.0273 |


