# Peer review of "Interaction of CO2 concentrations and water stress in semi-arid plants causes diverging response in instantaneous water use efficiency and carbon isotope composition"

_Biogeosciences, 2016_

## Referee Comment (RC1) · J. P. Ferrio Diaz (Referee) · 23 Oct 2016

REFEREE COMMENT-DISCUSSION

Differences in instantaneous water use efficiency derived from post-carboxylation fractionation respond to the interaction of CO2 concentrations and water stress in semi-arid areas

by Na Zhao, Ping Meng, Yabing He, and Xinxiao Yu

doi:10.5194/bg-2016-372

[Figure]

General comments

In this work, Zhao et al. present an experimental study on the interactive effects of CO2 and water availability on instantaneous water-use efficiency (iWUE) and the carbon isotope composition (d13C) of leaf water-soluble organic matter (LWSOM). Although the study of the interaction between CO2 and drought and its effects on d13C and iWUE is not new (Picon, Ferhi, & Guehl 1997), there is no clear consensus on the interpretation of d13C changes in response to increasing CO2 (Schubert & Jahren 2012). In this context, the comprehensive dataset here presented may contribute to understand the limitations of d13C as a surrogate for iWUE, and to better predict the response of tree species to increasing CO2, particularly in drought-prone environments. This is particularly relevant for the proper interpretation of long-term trends in d13C in relation to changes in water use efficiency, particularly in drought-prone environments, e.g. based on tree-ring records (Duquesnay et al. 1998;Saurer, Siegwolf, & Schweingruber 2004;Voltas et al. 2013), or from herbarium and sub-fossil material (Peñuelas & Azcón-Bieto 1992;Beerling 1996;Köhler et al. 2010).

The experiment is well-designed and the data is generally well presented, although some details on the methodology are missing (see technical corrections). However, the manuscript requires some improvements, particularly on the interpretation of results.

Specific comments

My main concern about the manuscript is that it relies on the assumption that the only source of divergence between gas-exchange iWUE and d13C of recent assimilates could be post-photosynthetic fractionation. Although this is likely to play a role, the authors should consider that what actually defines carbon isotope discrimination (D13C) is the CO2 concentration in the chloroplast (Cc), not in the intercellular space, as used in the simplified equation of the Farquar's model (Evans et al. 1986;Farquhar, Ehleringer, & Hubick 1989). Indeed, the difference between gas-exchange derived values and online measurements of D13C has been widely used to estimate Ci-Cc and

mesophyll conductance for CO2 (Le Roux et al. 2001;Warren & Adams 2006;Flexas et al. 2006;Evans et al. 2009;Flexas et al. 2012;Evans & von Caemmerer 2013). In this regard, changes in mesophyll conductance could be partly responsible for the observed variations, as it generally increases in the short term in response to elevated CO2 (Flexas et al. 2007;Flexas et al. 2014), whereas it tends to decrease under drought (Flexas et al. 2004;Ferrio et al. 2012;Hommel et al. 2014;Théroux-Rancourt, Éthier, & Pepin 2014). Hence, the manuscript would be greatly improved by considering both post-photosynthetic fractionation and mesophyll conductance as potential sources of variation. With the data available, the authors may be able to estimate changes in mesophyll conductance, based on the Evans method, which can be adapted to recent assimilates (Pons et al. 2009). Even without alternative estimates for mesophyll conductance, this would provide an useful ground for a deeper discussion.

Technical corrections

In its present form, the title may suggest that instantaneous water use efficiency is changing because of post-carboxylation fractionation, which is clearly not the case. Besides, after considering the role of mesophyll conductance, post-carboxylation fractionation should not play such a major role in the title. An alternative might be "The interaction of CO2 concentrations and water stress in semi-arid areas causes diverging response in instantaneous water use efficiency and carbon isotope composition". This leaves open the possibility to discuss both post-photosynthetic fractionation and mesophyll conductance as potential causes for the observed divergence.

In the abstract, lines 11-14: it seems that several concepts are mixed together here, trying to summarize everything in one sentence, but the result is unclear. I would recommend to split the ideas in shorter lines, and to try to go step by step in the argumentation line of the abstract.

The number of replicates (saplings) per treatment is not given in the methods (however it is shown in the figures, n=32). Please add, and also specify the number of leaves

measured/sampled per tree, number of gas-exchange measurements per leaf, etc.

In line 263 an attempt to quantify the so-called 'post-carboxylation fractionation' is given, but the methodology used is not described. As it is written, the sentence "When comparing WUEge and WUEcp, the 13C-depletion" is misleading, since it is not WUE calculated by the two methods what is compared here, but observed and modelled d13C. I guess the value results from the difference between observed d13C and modelled d13C calculated from gas-exchange data, i.e. by reverting equations 3 and 4, however this is not explained in the methods.

Text in the legends of Figs. 2-5 could be larger. Since each panel is associated to one single species, they could be simplified by including the name of the species elsewhere in the figure, and using the symbols only for the CO2 levels. The symbols for a given CO2 level could be the same in all panels, regardless of the species (in this way, one legend would be enough for all the panels).

In Figure 6 I would use the symbols to indicate CO2 levels, as in the rest of figures. This would be useful to see whether the positive association between "fractionation" and gs is linked with CO2 or water availability.

References

Beerling D.J. (1996) 13C discrimination by fossil leaves during the late-glacial climate oscillation 12-10 ka BP: measurements and physiological controls. Oecologia 108, 29-37.

Duquesnay A., Breda N., Stievenard M., & Dupouey J.L. (1998) Changes of tree-ring d13C and water-use efficiency of beech (Fagus sylvatica L.) in north-eastern France during the past century. Plant, Cell & Environment 21, 565-572.

Evans J.R., Sharkey T.D., Berry J.A., & Farquhar G.D. (1986) Carbon Isotope Discrimination Measured Concurrently with Gas- Exchange to Investigate Co2 Diffusion in Leaves of Higher- Plants. Australian Journal of Plant Physiology 13, 281-292.

Evans J.R., Kaldenhoff R., Genty B., & Terashima I. (2009) Resistances along the CO2 diffusion pathway inside leaves. Journal of Experimental Botany 60, 2235-2248.

Evans J.R. & von Caemmerer S. (2013) Temperature response of carbon isotope discrimination and mesophyll conductance in tobacco. Plant, Cell & Environment 36, 745-756.

Farquhar G.D., Ehleringer J.R., & Hubick K.T. (1989) Carbon isotope discrimination and photosynthesis. Annual Review of Plant Physiology and Plant Molecular Biology 40, 503-537.

Ferrio J.P., Pou A., Florez-Sarasa I., Gessler A., Kodama N., Flexas J., & Ribas-Carbo M. (2012) The Péclet effect on leaf water enrichment correlates with leaf hydraulic conductance and mesophyll conductance for CO2. Plant, Cell and Environment 35, 611-625.

Flexas J., Bota J., Loreto F., Cornic G., & Sharkey T.D. (2004) Diffusive and metabolic limitations to photosynthesis under drought and salinity in C-3 plants. Plant Biology 6, 269-279.

Flexas J., Diaz-Espejo A., Galmes J., Kaldenhoff R., Medrano H., & Ribas-Carbo M. (2007) Rapid variations of mesophyll conductance in response to changes in CO2 concentration around leaves. Plant Cell and Environment 30, 1284-1298.

Flexas J., Ribas-Carbo M., Hanson D.T., Bota J., Otto B., Cifre J., McDowell N., Medrano H., & Kaldenhoff R. (2006) Tobacco aquaporin NtAQP1 is involved in mesophyll conductance to CO2 in vivo. The Plant Journal 48, 427-439.

Flexas J., Barbour M.M., Brendel O., Cabrera H.M., Carriquí M., Díaz-Espejo A., Douthe C., Dreyer E., Ferrio J.P., Gago J., Gallé A., Galmés J., Kodama N., Medrano H., Niinemets Ü., Peguero-Pina J.J., Pou A., Ribas-Carbó M., Tomás M., Tosens T., & Warren C.R. (2012) Mesophyll diffusion conductance to CO2: An unappreciated central player in photosynthesis. Plant Science 193-194, 70-84.

Flexas J., Carriquí M., Coopman R.E., Gago J., Galmés J., Martorell S., Morales F., & Diaz-Espejo A. (2014) Stomatal and mesophyll conductances to CO 2 in different plant groups: Underrated factors for predicting leaf photosynthesis responses to climate change? Plant Science 226, 41-48.

Hommel R., Siegwolf R., Saurer M., Farquhar G.D., Kayler Z., Ferrio J.P., & Gessler A. (2014) Drought response of mesophyll conductance in forest understory species - impacts on water-use efficiency and interactions with leaf water movement. Physiologia Plantarum 152, 98-114.

Köhler I.H., Poulton P.R., Auerswald K., & Schnyder H. (2010) Intrinsic water-use efficiency of temperate seminatural grassland has increased since 1857: an analysis of carbon isotope discrimination of herbage from the Park Grass Experiment. Global Change Biology 16, 1531-1541.

Le Roux X., Bariac T., Sinoquet H., Genty B., Piel C., Mariotti A., Girardin C., & Richard P. (2001) Spatial distribution of leaf water-use efficiency and carbon isotope discrimination within an isolated tree crown. Plant, Cell and Environment 24, 1021-1032.

Peñuelas J. & Azcón-Bieto J. (1992) Clanges in leaf D13C of herbarium plant species during the last 3 centuries of CO2 increase. Plant, Cell and Environment 5, 485-489.

Picon C., Ferhi A., & Guehl J.M. (1997) Concentration and d13C of leaf carbohydrates in relation to gas exchange in Quercus robur under elevated CO2 and drought. Journal of Experimental Botany 48, 1547-1556.

Pons T.L., Flexas J., Von-Caemmerer S., Evans J.R., Genty B., Ribas-Carbo M., & Brugnoli E. (2009) Estimating mesophyl conductance to CO2: methodology, potential errors and recomendations. Journal of Experimental Botany 60, 2217-2234.

Saurer M., Siegwolf R., & Schweingruber F.H. (2004) Carbon isotope discrimination indicates improving water-use efficiency of trees in northern Eurasia over the last 100 years. Global Change Biology 10, 2109-2120.

Schubert B.A. & Jahren A.H. (2012) The effect of atmospheric $CO_2$ concentration on carbon isotope fractionation in $C_3$ land plants. Geochimica et Cosmochimica Acta 96, 29-43.

Théroux-Rancourt G., Éthier G., & Pepin S. (2014) Threshold response of mesophyll $CO_2$ conductance to leaf hydraulics in highly transpiring hybrid poplar clones exposed to soil drying. Journal of Experimental Botany 65, 741-753.

Voltas J., Camarero J.J., Carulla D., Aguilera M., Ortiz A., & Ferrio J.P. (2013) A retrospective, dual-isotope approach reveals individual predispositions to winter-drought induced tree dieback in the southernmost distribution limit of Scots pine. Plant, Cell & Environment 36, 1435-1448.

Warren C.R. & Adams M.A. (2006) Internal conductance does not scale with photosynthetic capacity: implications for carbon isotope discrimination and the economics of water and nitrogen use in photosynthesis. Plant Cell and Environment 29, 192-201

---

## Editor Comment (EC1) · C. Bourque (Editor) · 4 Nov 2016

Thank you for your review of the manuscript. The authors should consider all of your comments in their revision of their manuscript.

Kind regards,

Charles Bourque

---

## Author Comment (AC1) · 21 Dec 2016

Response to referee's comments

We thank and greatly appreciate the thoughtful and constructive comments from Professor Ferrio Diaz. We have fully considered your comments in the revision and improved the manuscript (revised manuscript marked in red color).

General comments

In this work, Zhao et al. present an experimental study on the interactive effects of CO2

and water availability on instantaneous water-use efficiency (iWUE) and the carbon isotope composition (d13C) of leaf water-soluble organic matter (LWSOM). Although the study of the interaction between CO2 and drought and its effects on d13C and iWUE is not new (Picon, Ferhi, & Guehl 1997), there is no clear consensus on the interpretation of d13C changes in response to increasing CO2 (Schubert & Jahren 2012). In this context, the comprehensive dataset here presented may contribute to understand the limitations of d13C as a surrogate for iWUE, and to better predict the response of tree species to increasing CO2, particularly in drought-prone environments. This is particularly relevant for the proper interpretation of long-term trends in d13C in relation to changes in water use efficiency, particularly in drought-prone environments, e.g. based on tree-ring records (Duquesnay et al. 1998; Saurer, Siegwolf, & Schweingruber 2004; Voltas et al. 2013), or from herbarium and sub-fossil material (Peñuelas & Azcón-Bieto 1992;Beerling 1996;Köhler et al. 2010). The experiment is well-designed and the data is generally well presented, although some details on the methodology are missing (see technical corrections). However, the manuscript requires some improvements, particularly on the interpretation of results.

Response: Thank you for the careful review and constructive comments. According your helpful suggestions, revisions throughout the whole article have been made and the results have been improved and supplemented with the related contents.

Specific comments

My main concern about the manuscript is that it relies on the assumption that the only source of divergence between gas-exchange iWUE and d13C of recent assimilates could be post-photosynthetic fractionation. Although this is likely to play a role, the authors should consider that what actually defines carbon isotope discrimination (D13C) is the CO2 concentration in the chloroplast (Cc), not in the intercellular space, as used in the simplified equation of the Farquar's model (Evans et al. 1986; Farquhar, Ehleringer, & Hubick 1989). Indeed, the difference between gas-exchange derived values and online measurements of D13C has been widely used to estimate Ci-Cc and

mesophyll conductance for CO2 (Le Roux et al. 2001;Warren & Adams 2006;Flexas et al. 2006;Evans et al. 2009;Flexas et al. 2012;Evans & von Caemmerer 2013). In this regard, changes in mesophyll conductance could be partly responsible for the observed variations, as it generally increases in the short term in response to elevated CO2 (Flexas et al. 2007;Flexas et al. 2014), whereas it tends to decrease under drought (Flexas et al. 2004;Ferrio et al. 2012;Hommel et al. 2014;Théroux-Rancourt, Éthier, & Pepin 2014). Hence, the manuscript would be greatly improved by considering both post-photosynthetic fractionation and mesophyll conductance as potential sources of variation. With the data available, the authors may be able to estimate changes in mesophyll conductance, based on the Evans method, which can be adapted to recent assimilates (Pons et al. 2009). Even without alternative estimates for mesophyll conductance, this would provide a useful ground for a deeper discussion.

Response: Thanks for your relevant and helpful comments about our research. The consensus has been reached that the routine of CO2 diffusion into photosynthetic site in plant includes two main procedures, which are CO2 moving from ambient environment surrounding the leaf (Ca) to the sub-stomatic cavities (Ci) through stomata, and from there to the site of carboxylation within the chloroplast stroma (Cc) of leaf mesophyll. The latter diffusion is defined as mesophyll conductance (gm) (Flexas et al., 2008; Evans et al. 2009). Moreover, gm has been identified to coordinate with environmental variables at the faster rate than that of stomatal conductance (Galmés et al., 2007; Tazoe et al., 2011; Flexas et al., 2007). gm as the important factor that could improve water use efficiency under drought pretreatment (Han et al., 2016). There has been a dispute how gm responds to fluctuation of CO2 concentration. Terashima et al. (2006) have confirmed that CO2 permeable aquaporin, located in the plasma membrane and inner envelope of chloroplasts (Uehlein et al. 2008), could regulate the change of gm.

The 13C fractionation of CO2 from air surrounding leaf to sub-stomatal cavity may be simply considered (Eqn. 6), whereas the fractionation induced by mesophyll conductance from sub-stomatic cavities to the site of carboxylation in the chloroplast cannot be neglected (Pons et al., 2009; Cano et al., 2014). As estimating the post-photosynthetic fractionation in leaf, carbon discrimination generated by mesophyll conductance must be subtracted from 13C fractionation from the site of carboxylation to cytoplasm before sugars transportation, estimated from the difference between $\delta\hat{}13$ C_WSC ($\delta\hat{}13$ C of water soluble compounds by carbon isotopic method) and $\delta\hat{}13$ C_model ($\delta\hat{}13$ C modeled from gas exchange measurement), which was closely associated with gm. Consequently, considering your constructive suggestions, gm in our study was determined based on the Evans method, which can be adapted to recent assimilates (Pons et al. 2009). And then we can estimate the variation of gm under SWC $\times$ [CO2] treatments. Related methods, results, discussions and conclusion of gm have been added in the revised manuscript (see Page 6-7, lines 216-261, Page 8-9, lines 319-330 and 347-355, Page 10-11, lines 410-435 and Page 12, lines 473-475). Subsequently, it has been shown that mesophyll conductance and post-carboxylation fractionation both contribute to the 13C fractionation from the site of carboxylation to cytoplasm (the difference between $\delta\hat{}13$ C_WSC and $\delta\hat{}13$ C_obs), which is derived from 13C fractionation following the carboxylation while photosynthate having not been transported to the twigs of plant in our study.

Added citations:

Brooks, A. and Farquhar, G. D.: Effect of temperature on the CO2/O2 specificity of ribulose-1,5-bisphosphate carboxylase/oxygenase and the rate of respiration in the light, Planta, 165, 397–406, 1985.

Cano, F. J., López, R., and Warren, C. R.: Implications of the mesophyll conductance to CO2 for photosynthesis and water-use efficiency during long-term water stress and recovery in two contrasting Eucalyptus species, Plant Cell Environ., 37, 2470–2490, 2014.

Flexas, J., Diaz-Espejo, A., Galmés, J., Kaldenhoff, R., Medano, H., and Ribas-Carbo,

M.: Rapid variations of mesophyll conductance in response to changes in $CO_2$ concentration around leaves, Plant Cell Environ., 30, 1284–1298, 2007.

Flexas, J., Ribas-Carbó, M., Diaz-Espejo, A., Galmés, J., and Medrano, H.: Mesophyll conductance to $CO_2$: current knowledge and future prospects, Plant Cell Environ., 31, 602–621, 2008.

Galmés, J., Medrano, H., and Flexas, J.: Photosynthetic limitations in response to water stress and recovery in Mediterranean plants with different growth forms, New Phytol., 175, 81–93. 2007.

Gillon, J. S., Griffiths, H.: The influence of (photo)respiration on carbon isotope discrimination in plants. Plant Cell Environ., 20, 1217–1230, 1997.

Guy, R. D., Fogel, M. L., and Berry, J. A.: Photosynthetic fractionation of the stable isotopes of oxygen and carbon, Plant Physiol., 101, 37–47, 1993.

Han, J. M., Meng, H. F., Wang, S. Y., Jiang, C. D., Liu, F., Zhang, W. F., and Zhang, Y. L.: Variability of mesophyll conductance and its relationship with water use efficiency in cotton leaves under drought pretreatment, J. Plant Physiol., 194, 61–71, 2016.

Igamberdiev, A. U., Mikkelsen, T. N., Ambus, P., Bauwe, H., and Lea, P. J.: Photorespiration contributes to stomatal regulation and carbon isotope fractionation: a study with barley, potato and Arabidopsis plants deficient in glycine decarboxylase, Photosynth. Res., 81, 139–152, 2004.

Lanigan, G. J., Betson, N., Griffiths, H., and Seibt, U.: Carbon isotope fractionation during photorespiration and carboxylation in Senecio, Plant Physiol., 148, 2013–2020, 2008.

Pons, T. L., Flexas, J., von Caemmerer, S., Evans, J. R., Genty, B., Ribas-Carbo, M., and Brugnoli, E.: Estimating mesophyll conductance to $CO_2$: methodology, potential errors, and recommendations, J. Exp. Bot., 8, 1–18, 2009.

Tazoe, Y., von Caemmerer, S., Estavillo, G. M., and Evans, J. R.: Using tunable diode laser spectroscopy to measure carbon isotope discrimination and mesophyll conductance to CO2 diffusion dynamically at different CO2 concentrations, Plant Cell Environ., 34, 580–591, 2011.

Terashima, I., Hanba, Y.T., Tazoe, Y., Vyas, P., and Yano, S.: Irradiance and phenotype: comparative eco-development of sun and shade leaves in relation to photosynthetic CO2 diffusion, J. Exp. Bot., 57, 343–354, 2006.

Uehlein, N., Otto, B., Hanson, D. T., Fischer, M., McDowell, N., and Kaldenhoff, R.: Function of Nicotiana tabacum aquaporins as chloroplast gas pores challenges the concept of membrane CO2 permeability, Plant Cell, 20, 648–657, 2008.

Technical corrections

In its present form, the title may suggest that instantaneous water use efficiency is changing because of post-carboxylation fractionation, which is clearly not the case. Besides, after considering the role of mesophyll conductance, post-carboxylation fractionation should not play such a major role in the title. An alternative might be "The interaction of CO2 concentrations and water stress in semi-arid areas causes diverging response in instantaneous water use efficiency and carbon isotope composition". This leaves open the possibility to discuss both post-photosynthetic fractionation and mesophyll conductance as potential causes for the observed divergence.

Response: We thank referee and greatly appreciate the thoughtful and constructive comments. Following your suggestions, the title was changed as "The interaction of CO2 concentrations and water stress in semi-arid areas causes diverging response in instantaneous water use efficiency and carbon isotope composition" in the revised manuscript, which can more comprehensively discuss both post-photosynthetic fractionation and mesophyll conductance as potential causes for the observed divergence.

In the abstract, lines 11-14: it seems that several concepts are mixed together here,

trying to summarize everything in one sentence, but the result is unclear. I would recommend to split the ideas in shorter lines, and to try to go step by step in the argumentation line of the abstract.

Response: Based on your constructive recommendation, we rewrote this part as (starting on Lines 10-13 in the abstract):

"The 13C fractionation may be generated through the transformation from photosynthate to sugars before transporting them outward the leaf. The influence of environmental conditions (i. e. CO2 concentration and water stress) and their interactions on this fractionation have not yet been identified".

The number of replicates (saplings) per treatment is not given in the methods (however it is shown in the figures, n=32). Please add, and also specify the number of leaves measured/sampled per tree, number of gas-exchange measurements per leaf, etc.

Response: Considering your suggestions, we modified and specified the sampling and measuring process in gas-exchange measurements and the extractions of water soluble compound of leaves to read (starting on Page 4, Line 159-161 and on Page 5, Line 168-170, respectively):

"Four replicates were measured with each leaf and four leaves were chosen per tree in the gas-exchange measurement. There were two saplings ready for one orthogonal treatment ([CO2] × water stress)."

"After gas exchange measurements, recently-expanded, eight sun leaves were removed per tree of two species and two cultivated saplings per specie were replicated per treatment, and then were frozen immediately in liquid nitrogen."

In line 263 an attempt to quantify the so-called 'post-carboxylation fractionation' is given, but the methodology used is not described. As it is written, the sentence "When comparing WUEge and WUEcp, the 13C-depletion" is misleading, since it is not WUE calculated by the two methods what is compared here, but observed and modelled

d13C. I guess the value results from the difference between observed d13C and modelled d13C calculated from gas-exchange data, i.e. by reverting equations 3 and 4, however this is not explained in the methods.

Response: Thanks for your helpful comments. Consistent with your speculation and considering the effect of mesophyll conductance, the defined 'post-carboxylation' or 'post-photosynthesis' that can explain part of the 13C fractionation from the site of carboxylation to cytoplasm before sugars transportation that is the difference between observed $\delta$13C of water soluble compounds from leaves and the modeled $\delta$13C calculated from gas-exchange, which in unmodified manuscript was not been explained in the methods, misleading that with the difference between WUEge and WUEcp. Considering with your suggestions, we added the methodology of post-carboxylation in "2.4.1 The 13C fractionation between the sub-stomatic cavities and the ambient environment" that reads (starting on Page 6, Line 209-215):

"Then the 13C fractionation between the sub-stomatic cavities and the ambient environment (total 13C fractionation) can be estimated by the observed $\delta$13C of water soluble compounds from leaves ($\delta$13CWSC) and the modelled $\delta$13C calculated from gas-exchange ($\delta$13Cmodel). The $\delta$13Cmodel can be calculated from $\Delta$_model from Eqn. (2). The $\Delta$_model can be determined by Eqns. (3 and 4) as:

$\Delta$_model=(b-a)(1-(1.6$\Delta$eWUE_ge)/C_a )+a (7)

$\delta$ˆ13 C_model=(C_a-$\Delta$_model)/(1+$\Delta$_model ) (8)

Total (_ˆ13)C fractionation=$\delta$ˆ13 C_WSC-$\delta$ˆ13 C_model (9)."

"3.4 13C fractionation from the site of carboxylation to cytoplasm before sugars transportation" has been modified as (starting on Page 8, Line 307-318):

"We evaluated the total 13C fractionation from the site of carboxylation to cytoplasm by gas exchange and $\delta$13C of water-soluble compounds from leaf measurements (Table 1), which can retrace 13C fractionation before carboxylation transport to the twig. Comparing $\delta\hat{}13$ C_WSC with $\delta\hat{}13$ C_model from Eqns. (4, 7 and 8), total 13C fractionation of P. orientalis ranged from 0.0328‰ to 0.0472‰ which was smaller than that of Q. variabilis (0.0384‰ to 0.0466‰. The total fractionations of P. orientalis were magnified with soil wetting especially that was increased by 21.30%–42.04% at 35%–80% of FC from C400 to C800. Fractionation coefficients under C400 and C500 were amplified as SWC increased until 50%–60% of FC in Q. variabilis, whereas it was increased at 50%–80% of FC and decreased at FC under C600 and C800. Elevated [CO2] enhanced the average fractionation effect of P. orientalis, while those of Q. variabilis declined sharply from C600 to C800. Total 13C fractionation in P. orientalis increased faster than did those of Q. variabilis with increased soil moisture."

"4.4 Post-carboxylation fractionation generated before photosynthate leaving leaves" was been improved as (starting on Page 11, Line 450-453):

"When comparing $\delta\hat{}13$ C_WSC with $\delta\hat{}13$ C_obs, total fractionations of P. orientalis ranged from 0.0328‰ to 0.0472‰ less than that of Q. variabilis (from 0.0384‰ to 0.0466‰. Then total 13C fractionation subtracted by fractionation derived from mesophyll conductance, post-photosynthetic fractionation occupied 75.30%-98.9% of total 13C fractionation."

The conclusion of this manuscript need to be modified as (starting on Page 12, Line 465-468 and 475-479):

"The influence of mesophyll conductance on the difference of 13C fractionation between the sub-stomatic cavities and the ambient environment need to be considered, while testing the hypothesis that the post-carboxylation will contribute to the 13C fractionation from the site of carboxylation to cytoplasm before sugars transportation."

"Rising [CO2] and/or soil moistening generated increasing disparities between $\delta\hat{}13$ C_WSC and $\delta\hat{}13$ C_model in P. orientalis; nevertheless, the differences between $\delta\hat{}13$ C_WSC and $\delta\hat{}13$ C_model in Q. variabilis increased as [CO2] being less than 600 ppm and/or water stress was alleviated. Total 13C fractionation in leaf was linearly

dependent on gs."

Text in the legends of Figs. 2-5 could be larger. Since each panel is associated to one single species, they could be simplified by including the name of the species elsewhere in the figure, and using the symbols only for the CO2 levels. The symbols for a given CO2 level could be the same in all panels, regardless of the species (in this way, one legend would be enough for all the panels).

Response: Thanks for your constructive comments. Considering your suggestions, the legends of Figs. 2-5 were simplified in the revised manuscript. The symbols for CO2 concentration of 400 ppm, 500 ppm, 600 ppm and 800 ppm were uniformly presented as C400, C500, C600 and C800 in sequence. One legend was shown in all panels of one Figure shown in Figs. 2-5 of revised manuscript.

In Figure 6 I would use the symbols to indicate CO2 levels, as in the rest of figures. This would be useful to see whether the positive association between "fractionation" and gs is linked with CO2 or water availability.

Response: Thank you for suggestions about the graphic settings. According your consideration, we have redrawn the images of Figs. 7 and 8 in the revised manuscript, which could obviously illustrate the relationships between gs/gm and total 13C fractionation. The legends of Figs. 7 and 8 were simplified. The symbols for CO2 concentration of 400 ppm, 500 ppm, 600 ppm and 800 ppm were uniformly presented as C400, C500, C600 and C800 in sequence.

Please also note the supplement to this comment:
http://www.biogeosciences-discuss.net/bg-2016-372/bg-2016-372-AC1-supplement.zip

---

## Referee Comment (RC2) · Anonymous Referee #2 · 19 Jan 2017

Interactive comment on "Differences in instantaneous water use efficiency derived from post-carboxylation fractionation respond to the interaction of CO2 concentrations and water stress in semi-arid areas"

General comments

In the context of global warming derived from the rising CO2 levels, severe drought conditions can be anticipated and are poised to change rapidly. Simultaneously, elevated CO2 concentrations ([CO2]) and more frequent droughts may also have interactive effects on physiological indexes and processes in plant. The carbon discrimination (13Δ) assimilated recently could more subtly provide timely feedback to environmental changes and their influences on diffusion via plant physiology and metabolic process within plants. Post-photosynthetic fractionation at the biochemical level is a well-documented phenomenon, which is caused by the difference in signatures between metabolites and intramolecullar position isotopic effects. Further, there is no clear consensus on the interpretation of $\delta$13C changes in response to the interaction of increasing CO2 and soil-water stresses. This paper distinctly presents the interaction of CO2 concentrations and water stress on the instantaneous water use efficiency and carbon isotope composition. The post-photosynthesis fractionation can explained the differences of the instantaneous water use efficiency measured by the gas-exchange method and the carbon isotopic composition from water-soluble compounds of leaves. The results of this study suggested that rising [CO2] coupled with moistened soil generated increasing disparities of $\delta$13C between the water soluble compounds ($\delta$13Cwsc) and estimated by gas-exchange observation ($\delta$13Cobs) in two species. Thus, cautious descriptions of the magnitude and environmental dependence of apparent post-carboxylation fractionation are worth our attention in photosynthetic fractionation. The experiment is well-designed and the data is generally well presented. This manuscript is suitable and has a merit for publication in this journal, although some details on the methodology and statement on results require some improvements (in special comments).

Special comments

In abstract, the author tried to state the carbon fractionation was generated from the carbon assimilation in the chloroplast to the sugars synthesized in the cytoplasm before photosynthetic products transportation outward the leaf. The vague concepts on Line 11-14 are stated. Separation of the long sentence into the shorter ones would be more beneficial for the readers to understand.

The replications of the measurements of gas-exchange and extractions of water-soluble compounds of leaves could not be found in the part of the materials and methods. Please specify the replications of leaves and trees measured in the gas-exchange and the number of leaves extracted the water-soluble compounds.

There are the 13C fractionation coefficients of two species involved in Tab. 1, which has not been defined in the introductions of methods. Please add and detail the definition of the 13C fractionation coefficients in the materials and methods.

In Line 202-232, the results of photosynthetic parameters were described one by one in detail. I would recommend stating the parameters with the same or similar trends all together. The physiological response of plants to the interactions of rising CO2 and water stresses could be better presented.
* * *

---

## Author Comment (AC2) · 23 Jan 2017

Response to referee's comments

Thanks for your thoughtful and constructive comments that provide scientific guidance for our writing and future research. We have fully considered your suggestions in the revised manuscript (marked in red color).

General comments

In the context of global warming derived from the rising CO2 levels, severe drought

conditions can be anticipated and are poised to change rapidly. Simultaneously, elevated CO2 concentrations ([CO2]) and more frequent droughts may also have interactive effects on physiological indexes and processes in plant. The carbon discrimination (13$\Delta$) assimilated recently could more subtly provide timely feedback to environmental changes and their influences on diffusion via plant physiology and metabolic process within plants. Post-photosynthetic fractionation at the biochemical level is a well-documented phenomenon, which is caused by the difference in signatures between metabolites and intramolecullar position isotopic effects. Further, there is no clear consensus on the interpretation of $\delta$13C changes in response to the interaction of increasing CO2 and soil-water stresses. This paper distinctly presents the interaction of CO2 concentrations and water stress on the instantaneous water use efficiency and carbon isotope composition. The post-photosynthesis fractionation can explained the differences of the instantaneous water use efficiency measured by the gas-exchange method and the carbon isotopic composition from water-soluble compounds of leaves. The results of this study suggested that rising [CO2] coupled with moistened soil generated increasing disparities of $\delta$13C between the water soluble compounds ($\delta$13Cwsc) and estimated by gas-exchange observation ($\delta$13Cobs) in two species. Thus, cautious descriptions of the magnitude and environmental dependence of apparent post-carboxylation fractionation are worth our attention in photosynthetic fractionation. The experiment is well-designed and the data is generally well presented. This manuscript is suitable and has a merit for publication in this journal, although some details on the methodology and statement on results require some improvements (in special comments).

Response: We thank and greatly appreciate the thoughtful and constructive comments. According your helpful suggestions, revisions for methodology and results have been made and the specific descriptions have been supplemented with the related contents.

Special comments

In abstract, the author tried to state the carbon fractionation was generated from the

carbon assimilation in the chloroplast to the sugars synthesized in the cytoplasm before photosynthetic products transportation outward the leaf. The vague concepts on Line 11-14 are stated. Separation of the long sentence into the shorter ones would be more beneficial for the readers to understand.

Response: We accept the referee's constructive suggestions and have rewritten the descriptions as (starting on Lines 10-14 in the abstract):

"It is commonly surveyed that the 13C fractionation derived from the CO2 diffusion occurred from ambient air to stomatal sub-cavity, and little investigate the 13C fractionation generated from the site of carboxylation to cytoplasm before sugars transportation outward the leaf, which may respond to the environmental conditions (i. e. CO2 concentration and water stress) and their interactive effects".

The replications of the measurements of gas-exchange and extractions of water-soluble compounds of leaves could not be found in the part of the materials and methods. Please specify the replications of leaves and trees measured in the gas-exchange and the number of leaves extracted the water-soluble compounds.

Response: As the referee's comments pointed out, we specified the sampling process in gas-exchange measurements and the extracted number for water soluble compound of leaves (starting on Page 4, Line 158-159 and on Page 4-5, Line 165-167, respectively):

"Two saplings per specie were replicated per treatment ([CO2] $\times$ water stress). For each sapling, four leaves were chosen and then four measurements were conducted on each leaf" on Page 4, Line 161-162.

"Recently-expanded, eight sun leaves per sapling were selected and frozen immediately in liquid nitrogen since the gas-exchange measurements accomplished. Two saplings per specie were chosen for each treatment" on Page 5, Line 168-170.

There are the 13C fractionation coefficients of two species involved in Tab. 1, which has

not been defined in the introductions of methods. Please add and detail the definition of the 13C fractionation coefficients in the materials and methods.

Response: Considering your advices combined with the first comments posted by the Professor Ferrio Diaz, we have redefined the '13C fractionation coefficients' as the 'total 13C fractionation' that represented the 13C fractionation from the site of carboxylation to cytoplasm before sugars transportation outward leaves. The 'total 13C fractionation' can be estimated by the observed $\delta$13C of water soluble compounds from leaves ($\delta$13CWSC) and the modeled $\delta$13C calculated from gas-exchange ($\delta$13Cmodel). Further, the calculation of mesophyll conductance and its contribution to the total 13C fractionation have been determined in the results and discussions (starting from Line 183 on Page 5 to Line 258 on Page 7):

"2.4.1 13C fractionation from the site of carboxylation to cytoplasm before sugars transportation

Based on the linear model developed by Farquhar and Sharkey (1982), the isotope discrimination factor, $\Delta$, was calculated as:

$\Delta=(((\_^{13})C\_a- (\_^{13})C\_P ))/((1+(\_^{13})C\_P ) )$ (2)

where $(\_^{13})C\_a$ is the isotope signature of ambient [CO2] in the chamber; $(\_^{13})C\_P$ is the $(\_^{13})C$âĹ̋ $(\_^{12})C$ of the water-soluble compounds extracted from foliage. The Ci:Ca is determined by:

$C\_i:C\_a=((\Delta-a))/((b-a) )$ (3)

where Ci is the intercellular CO2 concentration, and Ca is the ambient CO2 concentration in the chamber; a is the discrimination dependent on a fraction factor (4‰. b is the discrimination during CO2 fixation by ribulose 1,5- bisphosphate carboxylase/oxygenase (Rubisco) and internal diffusion (30‰. Instantaneous water use efficiency by gas-exchange measurements (WUE_ge) is calculated as: $WUE\_ge=P\_n:T\_r=((C\_a-C\_i ))/1.6\Delta e$ (4) whereãĂŰ PãĂŮ\_n is the net carbon assimilation, $T\_r$ is the molar rate of transpiration, and 1.6 is the diffusion ratio of stomatal conductance to water vapor to CO2 in the chamber. $\triangle e$ is the difference in water vapor pressure between the intracellular in leaves and ambient air, which may be calculated as:

$$\triangle e = e\_lf - e\_atm = 0.611 \times e^{\hat{}}(17.502T/((240.97+T)\,)) \times (1-RH) \quad (5)$$

where elf and eatm represent the extra- and intra-cellular water vapor pressure, respectively. T and RH is temperature and relative humidity on leaf surface. The instantaneous water use efficiency could be determined by the $\delta$13CWSC of leaves of two species, defined as WUEcp:

$$WUE\_cp = (\text{ãĂÚ } P\text{ãĂÙ}\_n/T\_r = (1-\varphi)\text{ãĂÚ}((C\_a-C\_i\,))/1.6\triangle e = C\text{ãĂÙ}\_a\,\,(1-\varphi)[(b-\delta^{\hat{}}13 C\_a + (b+1)\,\delta^{\hat{}}13 C\_WSC)/(b-a)(1+\delta^{\hat{}}13 C\_WSC\,)])/1.6\triangle e \quad (6)$$

$\varphi$ is the ratio between carbohydrates consumed during respiration of the leaves and that of other organs at night (0.3). $\delta^{\hat{}}13 C\_WSC$ is the carbon isotopic composition of water soluble compounds extracted from leaves.

Then the 13C fractionation from the site of carboxylation to cytoplasm before sugars transportation (total 13C fractionation) can be estimated by the observed $\delta$13C of water soluble compounds from leaves ($\delta$13CWSC) and the modeled $\delta$13C calculated from gas-exchange ($\delta$13Cmodel). The $\delta$13Cmodel can be calculated from $\triangle\_model$ from Eqn. (2). The $\triangle\_model$ can be determined by Eqns. (3 and 4) as:

$$\triangle\_model = (b-a)(1-(1.6\triangle eWUE\_ge)/C\_a\,)+a \quad (7)$$

$$\delta^{\hat{}}13 C\_model = (C\_a - \triangle\_model)/(1+\triangle\_model\,) \quad (8)$$

$$\text{Total } (\_^{\hat{}}13)C \text{ fractionation} = \delta^{\hat{}}13 C\_WSC - \delta^{\hat{}}13 C\_model \quad (9)$$

2.4.2 Methodology of calculating mesophyll conductance

Actually, the carbon isotope discrimination is generated from the relative contribution of diffusion and carboxylation, reflected by the ratio of CO2 concentration at the site

of carboxylation (Cc) to that in the ambient environment surrounding plants (Ca). The carbon isotopic discrimination ($\Delta$) could be presented as (Farquhar et al. 1982):

$$\Delta = a\_b\ (C\_a - C\_s)/C\_a + a\ (C\_s - C\_i)/C\_a + (e\_s + a\_l)\ (C\_i - C\_c)/C\_a + b\ C\_c/C\_a - ((eR\_D)/k + f\Gamma\_*)/C\_a\ (10)$$

where $C\_a, C\_s, C\_i$, and $C\_c$ indicate the $CO_2$ concentrations in the ambient environment, at the boundary layer of leaf, in the intercellular air spaces before entrancing into solution, and at the sites of carboxylation, respectively; $a\_b$ is the fractionation for the $CO_2$ diffusion at the boundary layer (2.9‰; a is the fractionation occurring $CO_2$ diffusion in still air (4‰; $e\_s$ is the discrimination of $CO_2$ diffusion when $CO_2$ enters in solution (1.1‰ at 25 °C); $a\_l$ is the fractionation derived from diffusion in the liquid phase (0.7‰; b is the carboxylation discrimination in C3 plants (27‰; e and f are carbon discrimination derived in dark respiration (RD) and photorespiration, respectively. k is the carboxylation efficiency, and $\Gamma^*$ is the $CO_2$ compensation point in the absence of dark respiration (Brooks and Farquhar,1985).

When the gas in the cuvette could be well stirred during measurements of carbon isotopic discrimination and gas exchange, the diffusion in the boundary layer could be neglected and Equation 7 could be shown:

$$\Delta = a\ (C\_a - C\_i)/C\_a + (e\_s + a\_l)\ (C\_i - C\_c)/C\_a + b\ C\_c/C\_a - ((eR\_D)/k + f\Gamma\_*)/C\_a\ (11)$$

There is no agreement about the value of e, although recent measurements estimated it as 0-4‰. Value of f has been estimated ranging at 8-12‰ (Gillon and Griffiths, 1997; Igamberdiev et al., 2004; Lanigan et al., 2008). As the most direct factor, the value of b will influence the calculation for gm, has been thought to be close to 30‰ in higher plants (Guy et al., 1993).

The difference of $CO_2$ concentration between the substomatal cavities and the chloroplast is omitted while diffusion discrimination related with dark-respiration and photorespiration is also negligible, the Equation 8 could be simplified as:
[Figure]

$\Delta_i = a + (b-a)\ C_i/C_a$ (12)

Equation 12 presents the linear relationship between carbon discrimination and Ci/Ca that used normally in carbon isotopic fractionation. That underlined the subsequent comparison between the expected $\Delta$ (originated from gas-exchange,ãĂŰ $\Delta$ãĂŮ_i) and those actually measuredãĂŰ ( $\Delta$ãĂŮ_obs), which could evaluate the magnitude of differences of CO2 concentration between the intercellular air and the sites of carboxylation that generated by mesophyll resistance. Consequently, gm can be estimated by performing the $\Delta$_obs by isotope ratio mass spectrometry and expected $\Delta$_i from Ci/Ca by gas exchange measurements.

Then subtract $\Delta$_obs of Equation 11 from $\Delta$_i calculated by Equation 12:

$\Delta_i - \Delta_{obs} = (b - e_s - a_l)\ (C_i - C_c)/C_a + ((eR\_D)/k + f\Gamma^*)/C_a$ (13)

and the net assimilation rate (An) from the first Fick's law is presented by:

$A_n = g_m\ (C_i - C_c)$ (14)

Substitute Equation 14 into Equation 13 we obtain:

$\Delta_i - \Delta_{obs} = (b - e_s - a_l)\ A_n/(g_m\ C_a) + ((eR\_D)/k + f\Gamma^*)/C_a$ (15)

$g_m = ((b - e_s - a_l)\ A_n/C_a )/((\Delta_i - \Delta_{obs}) - (ãĂŰeRãĂŮ\_D/k + f\Gamma^*)/C_a)$ (16)

In calculation of gm, the respiratory and photorespiratory terms could be ignored or be given the specific constant values. Here, e and f are assumed to be zero or be cancelled out in the calculation of gm.

Then Equation 16 can be transformed into:

$g_m = ((b - e_s - a_l)\ A_n/C_a )/(\Delta_i - \Delta_{obs})$ (17)".

In Line 202-232, the results of photosynthetic parameters were described one by one in detail. I would recommend stating the parameters with the same or similar trends all together. The physiological response of plants to the interactions of rising CO2 and

water stresses could be better presented.

Response: Thanks for your constructive comments. We have restated the photosynthetic parameters with the similar trends of $CO_2$ concentrations coupling the water stress (on Page 7, Lines 261-272):

"P. orientalis and Q. variabilis saplings were exposed to the orthogonal treatments. When SWC increased, Pn, gs and Tr in P. orientalis and Q. variabilis peaked at 70%–80% of FC or/and FC (Fig. 2). The Ci in P. orientalis rose as SWC increased, while it peaked at 60%–70% of FC and declined thereafter with increased SWC in Q. variabilis. The capacity of carbon uptake and Ci were elevated significantly by elevated [CO2] at any given SWC for two species ($p<0.05$). Further, greater increasing magnitudes of Pn in P. orientalis were found at 50%–70% of FC from C400 to C800, which was at 35%–45% of FC in Q. variabilis. As the water stress was alleviated (at 70%–80% of FC and FC), the reduction of gs in P. orientalis was more pronounced with elevated [CO2] at a given SWC ($p<0.01$). Nevertheless, gs of Q. variabilis in C400, C500, and C600 was significantly higher than that in C800 at 50%–80% of FC ($p<0.01$). Coordinated with gs, Tr of two species in C400 and C500 was significantly higher than that in C600 and C800 except for 35%–60% of FC ($p<0.01$, Figs. 2g and 2h). Larger Pn, gs, Ci and Tr of Q. variabilis was significantly presented than that of P. orientalis ($p<0.01$, Fig. 2)".

Please also note the supplement to this comment:
http://www.biogeosciences-discuss.net/bg-2016-372/bg-2016-372-AC2-supplement.zip

---

## Author Response (AR1)

Response list to the reviewers' comments

Ref: doi:10.5194/bg-2016-372

Title: **Differences in instantaneous water use efficiency derived from post-carboxylation fractionation respond to the interaction of $CO_2$ concentrations and water stress in semi-arid areas**

Authors: Na Zhao, Ping Meng, Yabing He, Xinxiao Yu*

Dear Editor,

Thanks for your thoughtful and constructive comments that provide scientific guidance for our writing and future research. We have been carefully considering your suggestions and revising the manuscript in the revised manuscript (marked in red color) accordingly. The following below in blue are our point-to-point responses for the referees' questions and your comments.

We are looking forward to your further comments and a possible publication in the BG special issue (Ecosystem processes and functioning across current and future dryness gradients in arid and semi-arid lands).

Kind regards,

Xinxiao Yu

■■■■■■■■■■■■■■■■■■■■■■■■■■■■■■■■■■■■■■■■■■■■■■■■■■■■■■■■■■■

**Referee #1**

General comments

*In this work, Zhao et al. present an experimental study on the interactive effects of CO2 and water availability on instantaneous water-use efficiency (iWUE) and the carbon isotope composition (d13C) of leaf water-soluble organic matter (LWSOM). Although the study of the interaction between CO2 and drought and its effects on d13C and iWUE is not new (Picon, Ferhi, & Guehl 1997), there is no clear consensus on the interpretation of d13C changes in response to increasing CO2 (Schubert & Jahren 2012). In this context, the comprehensive dataset here presented may contribute to understand the limitations of d13C as a surrogate for iWUE, and to better predict the response of tree species to increasing CO2, particularly in drought-prone environments. This is particularly relevant for the proper interpretation of long-term trends in d13C in relation to changes in water use efficiency, particularly in drought-prone environments, e.g. based on tree-ring records (Duquesnay et al. 1998; Saurer, Siegwolf, & Schweingruber 2004; Voltas et al. 2013), or from herbarium and sub-fossil material (Peñuelas & Azcón-Bieto 1992;Beerling 1996;Köhler et al. 2010). The experiment is well-designed and the data is generally well presented, although some details on the methodology are missing (see technical corrections). However, the manuscript requires some improvements, particularly on the interpretation of results.

Response: Thank you for the careful review and constructive comments. According your helpful suggestions, revisions throughout the whole article have been made and the results have been improved and supplemented with the related contents.

Specific comments

*My main concern about the manuscript is that it relies on the assumption that the only source of divergence between gas-exchange iWUE and d13C of recent assimilates could be post-photosynthetic fractionation. Although this is likely to play a role, the authors should

consider that what actually defines carbon isotope discrimination (D13C) is the CO2 concentration in the chloroplast (Cc), not in the intercellular space, as used in the simplified equation of the Farquar's model (Evans et al. 1986; Farquhar, Ehleringer, & Hubick 1989). Indeed, the difference between gas-exchange derived values and online measurements of D13C has been widely used to estimate Ci-Cc and mesophyll conductance for CO2 (Le Roux et al. 2001;Warren & Adams 2006;Flexas et al. 2006;Evans et al. 2009;Flexas et al. 2012;Evans & von Caemmerer 2013). In this regard, changes in mesophyll conductance could be partly responsible for the observed variations, as it generally increases in the short term in response to elevated CO2 (Flexas et al. 2007;Flexas et al. 2014), whereas it tends to decrease under drought (Flexas et al. 2004;Ferrio et al. 2012;Hommel et al. 2014;Théroux-Rancourt, Éthier, & Pepin 2014). Hence, the manuscript would be greatly improved by considering both post-photosynthetic fractionation and mesophyll conductance as potential sources of variation. With the data available, the authors may be able to estimate changes in mesophyll conductance, based on the Evans method, which can be adapted to recent assimilates (Pons et al. 2009). Even without alternative estimates for mesophyll conductance, this would provide a useful ground for a deeper discussion.

Response: Thanks for your helpful comments about our research. The consensus has been reached that the routine of $CO_2$ diffusion into photosynthetic site in plant includes two main procedures, which are $CO_2$ moving from ambient environment surrounding the leaf ($C_a$) to the sub-stomatic cavities ($C_i$) through stomata, and from there to the site of carboxylation within the chloroplast stroma ($C_c$) of leaf mesophyll. The latter diffusion is defined as mesophyll conductance ($g_m$) (Flexas et al., 2008; Evans et al. 2009). Moreover, $g_m$ has been identified to coordinate with environmental variables at the faster rate than that of stomatal conductance (Galmés et al., 2007; Tazoe et al., 2011; Flexas et al., 2007). $g_m$ as the important factor that could improve water use efficiency under drought pretreatment (Han et al., 2016). There has been a dispute how $g_m$ responds to fluctuation of $CO_2$ concentration. Terashima *et al*. (2006) have confirmed that $CO_2$ permeable aquaporin, located in the plasma membrane and inner envelope of chloroplasts (Uehlein et al. 2008), could regulate the change of $g_m$.

The [13]C fractionation of $CO_2$ from air surrounding leaf to sub-stomatal cavity may be simply considered (Eqn. 6), whereas the fractionation induced by mesophyll conductance from sub-stomatal cavities to the site of carboxylation in the chloroplast cannot be neglected (Pons et al., 2009; Cano et al., 2014). As estimating the post-photosynthetic fractionation in leaf, carbon discrimination generated by mesophyll conductance must be subtracted from [13]C fractionation from the site of carboxylation to cytoplasm before sugars transportation, estimated from the difference between $\delta^{13}C_{WSC}$ ($\delta^{13}C$ of water soluble compounds by carbon isotopic method) and $\delta^{13}C_{WSC}$ ($\delta^{13}C$ modeled from gas exchange measurement), which was closely associated with $g_m$. Consequently, considering your constructive suggestions, $g_m$ in our study was determined based on the Evans method, which can be adapted to recent assimilates (Pons et al. 2009). And then we can estimate the variation of $g_m$ under SWC $\times$ [CO2] treatments. Related data in Figures, methods, results, discussions and conclusion of $g_m$ have been added in the revised manuscript (see Figure 6 and 8, Pages 6-7, Lines 210-258, Pages 8-9, Lines 302-348, Pages 11-12, Lines 401-451 and Page 12, lines 455-459 and 464-469 in the revised manuscript). Subsequently, it has been shown that mesophyll conductance and post-carboxylation fractionation both contribute to the [13]C fractionation from the site of carboxylation to cytoplasm (the difference between $\delta^{13}C_{WSC}$ and $\delta^{13}C_{obs}$), which is derived from [13]C fractionation following the carboxylation while photosynthate

having not been transported to the twigs of plant in our study.

Added citations:

Brooks, A. and Farquhar, G. D.: Effect of temperature on the $CO_2/O_2$ specificity of ribulose-1,5-bisphosphate carboxylase/oxygenase and the rate of respiration in the light, Planta, 165, 397–406, 1985.

Cano, F. J., López, R., and Warren, C. R.: Implications of the mesophyll conductance to $CO_2$ for photosynthesis and water-use efficiency during long-term water stress and recovery in two contrasting Eucalyptus species, Plant Cell Environ., 37, 2470–2490, 2014.

Flexas, J., Diaz-Espejo, A., Galmés, J., Kaldenhoff, R., Medano, H., and Ribas-Carbo, M.: Rapid variations of mesophyll conductance in response to changes in $CO_2$ concentration around leaves, Plant Cell Environ., 30, 1284–1298, 2007.

Flexas, J., Ribas-Carbó, M., Diaz-Espejo, A., Galmés, J., and Medrano, H.: Mesophyll conductance to $CO_2$: current knowledge and future prospects, Plant Cell Environ., 31, 602–621, 2008.

Galmés, J., Medrano, H., and Flexas, J.: Photosynthetic limitations in response to water stress and recovery in Mediterranean plants with different growth forms, New Phytol., 175, 81–93. 2007.

Gillon, J. S., Griffiths, H.: The influence of (photo)respiration on carbon isotope discrimination in plants. Plant Cell Environ., 20, 1217–1230, 1997.

Guy, R. D., Fogel, M. L., and Berry, J. A.: Photosynthetic fractionation of the stable isotopes of oxygen and carbon, Plant Physiol., 101, 37–47, 1993.

Han, J. M., Meng, H. F., Wang, S. Y., Jiang, C. D., Liu, F., Zhang, W. F., and Zhang, Y. L.: Variability of mesophyll conductance and its relationship with water use efficiency in cotton leaves under drought pretreatment, J. Plant Physiol., 194, 61–71, 2016.

Igamberdiev, A. U., Mikkelsen, T. N., Ambus, P., Bauwe, H., and Lea, P. J.: Photorespiration contributes to stomatal regulation and carbon isotope fractionation: a study with barley, potato and Arabidopsis plants deficient in glycine decarboxylase, Photosynth. Res., 81, 139–152, 2004.

Lanigan, G. J., Betson, N., Griffiths, H., and Seibt, U.: Carbon isotope fractionation during photorespiration and carboxylation in Senecio, Plant Physiol., 148, 2013–2020, 2008.

Pons, T. L., Flexas, J., von Caemmerer, S., Evans, J. R., Genty, B., Ribas-Carbo, M., and Brugnoli, E.: Estimating mesophyll conductance to $CO_2$: methodology, potential errors, and recommendations, J. Exp. Bot., 8, 1–18, 2009.

Tazoe, Y., von Caemmerer, S., Estavillo, G. M., and Evans, J. R.: Using tunable diode laser spectroscopy to measure carbon isotope discrimination and mesophyll conductance to $CO_2$ diffusion dynamically at different $CO_2$ concentrations, Plant Cell Environ., 34, 580–591, 2011.

Terashima, I., Hanba, Y.T., Tazoe, Y., Vyas, P., and Yano, S.: Irradiance and phenotype: comparative eco-development of sun and shade leaves in relation to photosynthetic $CO_2$ diffusion, J. Exp. Bot., 57, 343–354, 2006.

Uehlein, N., Otto, B., Hanson, D. T., Fischer, M., McDowell, N., and Kaldenhoff, R.: Function of Nicotiana tabacum aquaporins as chloroplast gas pores challenges the concept of membrane $CO_2$ permeability, Plant Cell, 20, 648–657, 2008.

Technical corrections

*In its present form, the title may suggest that instantaneous water use efficiency is changing because of post-carboxylation fractionation, which is clearly not the case. Besides, after considering the role of mesophyll conductance, post-carboxylation fractionation should not play such a major role in the title. An alternative might be "The interaction of $CO_2$ concentrations and

water stress in semi-arid areas causes diverging response in instantaneous water use efficiency and carbon isotope composition". This leaves open the possibility to discuss both post-photosynthetic fractionation and mesophyll conductance as potential causes for the observed divergence.

Response: We thank referee and greatly appreciate the thoughtful and constructive comments. Following your suggestions, the title was changed as "The interaction of $CO_2$ concentrations and water stress in semi-arid areas causes diverging response in instantaneous water use efficiency and carbon isotope composition" in the revised manuscript, which can more comprehensively discuss both post-carboxylation fractionation and mesophyll conductance as potential causes for the observed divergence.

*In the abstract, lines 11-14: it seems that several concepts are mixed together here, trying to summarize everything in one sentence, but the result is unclear. I would recommend to split the ideas in shorter lines, and to try to go step by step in the argumentation line of the abstract.

Response: Based on your constructive recommendation, we rewrote this part as (starting on Lines 10-14 in the abstract of revised manuscript):

"It is commonly surveyed that the $^{13}C$ fractionation derived from the $CO_2$ diffusion occurred from ambient air to sub-stomatal cavity, and little investigate the $^{13}C$ fractionation generated from the site of carboxylation to cytoplasm before sugars transportation outward the leaf, which may respond to the environmental conditions (i. e. $CO_2$ concentration and water stress) and their interactions".

*The number of replicates (saplings) per treatment is not given in the methods (however it is shown in the figures, n=32). Please add, and also specify the number of leaves measured/sampled per tree, number of gas-exchange measurements per leaf, etc.

Response: Considering your suggestions, we modified and specified the sampling and measuring process in gas-exchange measurements and the extractions of water soluble compound of leaves to read (starting on Page 3, Lines 114-116, Page 4, Lines 156-158 and Pages 4-5, Line 164-169, respectively):

"Saplings of two species that have similar ground diameters, heights, and growth statuses were selected. One sapling from two species was placed in one pot (22 cm in diameter and 22 cm in height)".

"Two saplings per specie were replicated per treatment (SWC× [$CO_2$]). For each sapling, four leaves were chosen and then four measurements were conducted on each leaf".

"Recently-expanded, eight sun leaves per sapling were selected and homogenized in liquid nitrogen since the gas-exchange measurements accomplished. For the extraction of the water-soluble compounds (WSCs) from the leaves (Gessler et al., 2004), 50 mg of ground leaves and 100 mg of PVPP (polyvinylpolypyrrolidone) were mixed and incubated in 1mL double demineralized water for 60 min at 5℃ in a centrifuge tube. Each leaf was replicated two times. Two saplings per specie were chosen for each orthogonal treatment".

*In line 263 an attempt to quantify the so-called 'post-carboxylation fractionation' is given, but the methodology used is not described. As it is written, the sentence "When comparing WUEge and WUEcp, the 13C-depletion" is misleading, since it is not WUE calculated by the two methods what is compared here, but observed and modelled d13C. I guess the value results from the difference between observed d13C and modelled d13C calculated from gas-exchange data, i.e. by reverting equations 3 and 4, however this is not explained in the methods.

Response: Thanks for your helpful comments. Consistent with your speculation and considering

the effect of mesophyll conductance, the defined 'post-carboxylation' or 'post-photosynthesis' that can explain part of the [13]C fractionation from the site of carboxylation to cytoplasm before sugars transportation that is the difference between observed $\delta^{13}C$ of water soluble compounds from leaves and the modeled $\delta^{13}C$ calculated from gas-exchange, which in unmodified manuscript was not been explained in the methods, misleading that with the difference between $WUE_{ge}$ and $WUE_{cp}$. Considering with your suggestions, we added the methodology of post-carboxylation in the revised manuscript.

"2.4.1 [13]C fractionation from the site of carboxylation to cytoplasm before sugars transportation" that reads (starting on Page 6, Lines 203-209):

"Then the [13]C fractionation from the site of carboxylation to cytoplasm before sugars transportation (total [13]C fractionation) can be estimated by the observed $\delta^{13}C$ of water soluble compounds from leaves ($\delta^{13}C_{WSC}$) and the modeled $\delta^{13}C$ calculated from gas-exchange ($\delta^{13}C_{model}$). The $\delta^{13}C_{model}$ is calculated from $\Delta_{model}$ from Eqn. (2). The $\Delta_{model}$ can be determined by Eqns. (3 and 4) as:

$$\Delta_{model} = (b-a)\left(1 - \frac{1.6\Delta e WUE_{ge}}{C_a}\right) + a \tag{7}$$

$$\delta^{13}C_{model} = \frac{C_a - \Delta_{model}}{1 + \Delta_{model}} \tag{8}$$

$$\text{Total } ^{13}\text{C fractionation} = \delta^{13}C_{WSC} - \delta^{13}C_{model} \tag{9}$$".

"3.4 [13]C fractionation from the site of carboxylation to cytoplasm before sugars transportation" has been modified as (starting on Pages 8, Lines 303-313):

"We evaluated the total [13]C fractionation from the site of carboxylation to cytoplasm by gas exchange and $\delta^{13}C$ of water-soluble compounds from leaf (Table 1), which can retrace [13]C fractionation before carboxylation transport to the twig. Comparing $\delta^{13}C_{WSC}$ with $\delta^{13}C_{model}$ from Eqns. (4, 7 and 8), total [13]C fractionation of *P. orientalis* ranged from 0.0328‰ to 0.0472‰, which was smaller than that of *Q. variabilis* (0.0384‰ to 0.0466‰). The total fractionations of *P. orientalis* were magnified with soil wetting especially that reached 35%−80% of FC from $C_{400}$ to $C_{800}$ (increased by 21.30%−42.04%). The total fractionation under $C_{400}$ and $C_{500}$ were amplified as SWC increased until 50%−60% of FC in *Q. variabilis*, whereas it was increased at 50%−80% of FC and decreased at 100% FC under $C_{600}$ and $C_{800}$. Elevated [CO_2] enhanced the average total fractionation effect of *P. orientalis*, while those of *Q. variabilis* declined sharply from $C_{600}$ to $C_{800}$. Total [13]C fractionation in *P. orientalis* increased faster than did those of *Q. variabilis* with increased soil moisture".

"4.4 Post-carboxylation fractionation generated before photosynthate leaving leaves" was been improved as (starting on Page 11-12, Lines 441-444):

"When comparing $\delta^{13}C_{WSC}$ with $\delta^{13}C_{obs}$, total [13]C fractionation of *P. orientalis* ranged from 0.0328‰ to 0.0472‰, less than that of *Q. variabilis* (from 0.0384‰ to 0.0466‰). The post-carboxylation fractionation contributed 75.30%-98.9% of total [13]C fractionation, which was determined by subtracting the fractionation of mesophyll conductance from total [13]C fractionation".

The conclusion of this manuscript need to be modified as (starting on Page 12, Lines 455-459 and 464-469):

"The influence of mesophyll conductance on the difference of [13]C fractionation between the sub-stomatic cavities and the ambient environment need to be considered, while testing the hypothesis that the post-carboxylation will contribute to the [13]C fractionation from the site of

carboxylation to cytoplasm before sugars transportation".

"Mesophyll conductance and post-photosynthesis were manifested both contributing to the $^{13}$C fractionation from the site of carboxylation to cytoplasm before sugars transportation determined by gas exchange and carbon isotopic measurements. Rising [$CO_2$] and/or soil moistening generated increasing disparities between $\delta^{13}C_{WSC}$ and in $\delta^{13}C_{model}$ P. orientalis; nevertheless, the differences between $\delta^{13}C_{WSC}$ and $\delta^{13}C_{model}$ in Q. variabilis increased as [$CO_2$] being less than 600 ppm and/or water stress was alleviated. Total $^{13}$C fractionation in leaf was linearly dependent on $g_s$".

*Text in the legends of Figs. 2-5 could be larger. Since each panel is associated to one single species, they could be simplified by including the name of the species elsewhere in the figure, and using the symbols only for the CO2 levels. The symbols for a given CO2 level could be the same in all panels, regardless of the species (in this way, one legend would be enough for all the panels).

Response: Thanks for your constructive comments. Considering your suggestions, the legends of Figs. 2-5 were simplified in the revised manuscript. The symbols for $CO_2$ concentration of 400 ppm, 500 ppm, 600 ppm and 800 ppm were uniformly presented as $C_{400}$, $C_{500}$, $C_{600}$ and $C_{800}$ in sequence. One legend was shown in all panels of one Figure shown in Figs. 2-5 of revised manuscript. Furthermore, we have revised the captions of Figs. 2-5 and 7 shown in the revised manuscript, and then numbered and named the individual panels within a composite figure with a lower-case letter in the upper left hand corner of the graph and cite in the simplified caption.

*In Figure 6 I would use the symbols to indicate CO2 levels, as in the rest of figures. This would be useful to see whether the positive association between "fractionation" and gs is linked with CO2 or water availability.

Response: Thank you for suggestions about the graphic settings. According your consideration, we have redrawn the images of Figs. 7 and 8 in the revised manuscript, which could obviously illustrate the relationships between $g_s/g_m$ and total $^{13}$C fractionation. The legends of Figs. 7 and 8 were simplified. The symbols for $CO_2$ concentration of 400 ppm, 500 ppm, 600 ppm and 800 ppm were uniformly presented as $C_{400}$, $C_{500}$, $C_{600}$ and $C_{800}$ in sequence. Furthermore, the captions have been simplified to number the panels of the composite figure with a lower-case letter in the upper left hand corner of the graph and cite in the simplified caption.

**Referee #2**

General comments

*In the context of global warming derived from the rising $CO_2$ levels, severe drought conditions can be anticipated and are poised to change rapidly. Simultaneously, elevated $CO_2$ concentrations ([$CO_2$]) and more frequent droughts may also have interactive effects on physiological indexes and processes in plant. The carbon discrimination ($^{13}\Delta$) assimilated recently could more subtly provide timely feedback to environmental changes and their influences on diffusion via plant physiology and metabolic process within plants. Post-photosynthetic fractionation at the biochemical level is a well-documented phenomenon, which is caused by the difference in signatures between metabolites and intramolecullar position isotopic effects. Further, there is no clear consensus on the interpretation of $\delta^{13}C$ changes in response to the interaction of increasing $CO_2$ and soil-water stresses. This paper distinctly presents the interaction of $CO_2$ concentrations and water stress on the instantaneous water use efficiency and carbon isotope composition. The

post-photosynthesis fractionation can explained the differences of the instantaneous water use efficiency measured by the gas-exchange method and the carbon isotopic composition from water-soluble compounds of leaves. The results of this study suggested that rising $[CO_2]$ coupled with moistened soil generated increasing disparities of $\delta^{13}C$ between the water soluble compounds ($\delta^{13}C_{wsc}$) and estimated by gas-exchange observation ($\delta^{13}C_{obs}$) in two species. Thus, cautious descriptions of the magnitude and environmental dependence of apparent post-carboxylation fractionation are worth our attention in photosynthetic fractionation. The experiment is well-designed and the data is generally well presented. This manuscript is suitable and has a merit for publication in this journal, although some details on the methodology and statement on results require some improvements (in special comments).

Response: We thank and greatly appreciate the thoughtful and constructive comments. According your helpful suggestions, revisions for methodology and results have been made and the specific descriptions have been supplemented with the related contents.

Special comments

*In abstract, the author tried to state the carbon fractionation was generated from the carbon assimilation in the chloroplast to the sugars synthesized in the cytoplasm before photosynthetic products transportation outward the leaf. The vague concepts on Line 11-14 are stated. Separation of the long sentence into the shorter ones would be more beneficial for the readers to understand.

Response: We accept the referee's constructive suggestions and have rewritten the descriptions as (starting on Lines 10-14 in the abstract of revised manuscript):

"It is commonly surveyed that the $^{13}C$ fractionation derived from the $CO_2$ diffusion occurred from ambient air to sub-stomatal cavity, and little investigate the $^{13}C$ fractionation generated from the site of carboxylation to cytoplasm before sugars transportation outward the leaf, which may respond to the environmental conditions (i. e. $CO_2$ concentration and water stress) and their interactions".

*The replications of the measurements of gas-exchange and extractions of water-soluble compounds of leaves could not be found in the part of the materials and methods. Please specify the replications of leaves and trees measured in the gas-exchange and the number of leaves extracted the water-soluble compounds.

Response: As the referee's comments pointed out, we specified the sampling process in gas-exchange measurements and the extracted number for water soluble compound of leaves (starting on Page 3, Lines 114-116, Page 4, Lines 156-158 and Pages 4-5, Line 164-169, respectively):

"Saplings of two species that have similar ground diameters, heights, and growth statuses were selected. One sapling from two species was placed in one pot (22 cm in diameter and 22 cm in height)".

"Two saplings per specie were replicated per treatment (SWC× $[CO_2]$). For each sapling, four leaves were chosen and then four measurements were conducted on each leaf".

"Recently-expanded, eight sun leaves per sapling were selected and homogenized in liquid nitrogen since the gas-exchange measurements accomplished. For the extraction of the water-soluble compounds (WSCs) from the leaves (Gessler et al., 2004), 50 mg of ground leaves and 100 mg of PVPP (polyvinylpolypyrrolidone) were mixed and incubated in 1mL double demineralized water for 60 min at 5℃ in a centrifuge tube. Each leaf was replicated two times.

Two saplings per specie were chosen for each orthogonal treatment".

*There are the [13]C fractionation coefficients of two species involved in Tab. 1, which has not been defined in the introductions of methods. Please add and detail the definition of the [13]C fractionation coefficients in the materials and methods.

Response: Considering your advices combined with the first comments posted by the Professor Ferrio Diaz, we have redefined the '[13]C fractionation coefficients' as the 'total [13]C fractionation' that represented the [13]C fractionation from the site of carboxylation to cytoplasm before sugars transportation outward leaves. The 'total [13]C fractionation' can be estimated by the observed $\delta^{13}C$ of water soluble compounds from leaves ($\delta^{13}C_{WSC}$) and the modeled $\delta^{13}C$ calculated from gas-exchange ($\delta^{13}C_{model}$). Further, the calculation of mesophyll conductance and its contribution to the total [13]C fractionation have been determined in the results and discussions (starting from Line 182 on Page 5 to Line 258 on Page 7):

[revised manuscript text omitted]

*In Line 202-232, the results of photosynthetic parameters were described one by one in detail. I would recommend stating the parameters with the same or similar trends all together. The physiological response of plants to the interactions of rising $CO_2$ and water stresses could be better presented.

Response: Thanks for your constructive comments. We have restated the photosynthetic parameters with the similar trends of $CO_2$ concentrations coupling the water stress (on Pages 7-8, Lines 261-272):

"Saplings of *P. orientalis* and *Q. variabilis* were exposed to the orthogonal treatments. When SWC increased, $P_n$, $g_s$ and $T_r$ in *P. orientalis* and *Q. variabilis* peaked at 70%−80% of FC or/and 100% FC (Fig. 2). The $C_i$ in *P. orientalis* rose as SWC increased, while it peaked at 60%−70% of FC and declined thereafter with increased SWC in *Q. variabilis*. The capacity of carbon uptake and $C_i$ were improved significantly by elevated [$CO_2$] at any given SWC for two species ($p<0.05$). Furthermore, greater increments of $P_n$ in *P. orientalis* were found at 50%−70% of FC from $C_{400}$ to $C_{800}$, which was at 35%−45% of FC in *Q. variabilis*. As the water stress was alleviated (at 70%−80% of FC and 100% FC), the reduction of $g_s$ in *P. orientalis* was more pronounced with elevated [$CO_2$] at a given SWC ($p<0.01$). Nevertheless, $g_s$ of *Q. variabilis* in $C_{400}$, $C_{500}$, and $C_{600}$ was significantly higher than that in $C_{800}$ at 50%−80% of FC ($p<0.01$). Coordinated with $g_s$, $T_r$ of two species in

$C_{400}$ and $C_{500}$ was significantly higher than that in $C_{600}$ and $C_{800}$ except for 35%−60% of FC ($p<0.01$, Figs. 2g and 2h). Larger $P_n$, $g_s$, $C_i$ and $T_r$ of *Q. variabilis* was significantly presented than that of *P. orientalis* ($p<0.01$, Fig. 2)".

**Response list to the editor's comments#**

*P1, L13-14: the sentence, "Either its variation according to…" is awkward and should be rephrased;

Response: We appreciate your helpful comments. Based on your constructive recommendation, we have rewritten this part as (starting on Lines 10-14 in the abstract):

"It is commonly surveyed that the $^{13}$C fractionation derived from the $CO_2$ diffusion occurred from ambient air to sub-stomatal cavity, and little investigate the $^{13}$C fractionation generated from the site of carboxylation to cytoplasm before sugars transportation outward the leaf, which may respond to the environmental conditions (i. e. $CO_2$ concentration and water stress) and their interactions".

*P1, L22: "…of the two saplings…"; gives the impression that you examined two saplings only, when in fact many more than two were studied per species (e.g., see caption of Figure 2); please rephrased; if it is the case, please provide more detail;

Response: Thank you for careful suggestions. As you observed, there were more than two saplings repeated in each orthogonal treatment. To avoid the confusion, we have rephrased "…of the two saplings…" into "…of the two species…" in the whole article.

*P1, L23: "Field Capacity", no need to capitalize the first letter in each word;

Response: Thank you for suggestions on writing form. We have changed the first letters of "Field Capacity" with lowercase ones on Page 1, Line 22 and Page 4, Line 139.

*Many unnecessary uses of "the"; you may remove without loss of meaning (e.g., P1, L25, "…differed between the species." and P2, L70, "phloem transport, the remobilization…", P4, L131, "…the soil moisture sensors");

Response: Based on your suggestions, we will remove the unnecessary article "the" throughout the whole manuscript.

*P1, L30: "Further" should be "Furthermore";

Response: According to the context and your comments, we have corrected this grammatical error throughout the whole text.

*P1, L31: "…increased as CO2 concentration increased and water stress alleviated (…" can be simplified to ""…increased as CO2 concentration and water stress increased";

Response: Based on the first referee's comments that considering the effect of mesophyll conductance on the $^{13}$C fractionation from the site of carboxylation to cytoplasm before sugars transportation (total $^{13}$C fractionation), the defined 'post-carboxylation' or 'post-photosynthesis' that can explain part of total $^{13}$C fractionation. The total $^{13}$C fractionation is the difference between observed $\delta^{13}$C of water soluble compounds from leaves and the modeled $\delta^{13}$C calculated from gas-exchange, which has been misled with the difference between $WUE_{ge}$ and $WUE_{cp}$. Consequently, following your suggestions, "Further, the differences between $WUE_{ge}$ and $WUE_{cp}$ of *Q. variabilis* increased as $CO_2$ concentration increased and water stress alleviated (0.0384‰− 0.0466‰)" has been simplified as "Furthermore, differences between $\delta^{13}C_{WSC}$ and $\delta^{13}C_{obs}$ of *Q. variabilis* increased as $CO_2$ concentration and SWC increased (0.0384‰−0.0466‰)" on Page 1,

Lines 30-31 in the revised manuscript.

*P1, L33: "cautious descriptions" or "clear description"?

Response: We agree with your suggestion and have changed "cautious descriptions" to "clear description" on Page 1, Line 35 in the revised manuscript.

*P2, L43: ",but also will…" should be ",but will…"; "also" is not needed;

Response: Thanks for the suggestion about writing grammar. We have removed the "also" on Page 2, Lines 43-44 in the revised manuscript.

*P2, L49: "physiology" should be "physiological"

Response: We thank for your helpful corrections on grammatical errors and have corrected the spelling problems on Page 2, Line 49 in the revised manuscript.

*P2, L60: "considerably" is not needed; remove;

Response: We agree with your advice and have removed "considerably" on Page 2, Line 60 in the revised manuscript.

*P2, L61: "well" is not needed; please remove;

Response: We agree with your suggestion and have removed "well" on Page 2, Line 61 in the revised manuscript.

*P2, L70: "fractionations" can be made singular; remove the "s";

Response: Based on your suggestion, we have changed "fractionations" to "fractionation" on Page 2, Line 70 in the revised manuscript.

*P3, L82: "…isotope studies…" should be "…isotopic studies…";

Response: Thanks for your suggestion. We will use the adjective "isotopic" on Page 3, Line 88 in the revised manuscript.

*P3, L83: "…, and will help…" should be "…, which may help…";

Response: According your helpful comments and the unrevised context, we will change "…, and will help…" into "…, which may help…" on Page 3 Line 89, which are much clearer and easier for readers.

*P3, L86: "…, which also can…" should be "…, which can also…";

Response: Thank you for pointing out the mistake. We have checked and corrected the similar mistakes on Page 2, Line 51, Page 3, Line 92 and Page 10, Line 383 in the whole article.

*P3, L90: rephrase "…has not yet been observed" to "…has yet to be observed";

Response: We agree with your suggestion and have changed the expression on Page 3, Line 97-98 in the revised manuscript.

*P3, L92: change "in" to "to";

Response: Based on your helpful comments, we have changed "in" to "to" on Page 3, Line 100.

*P3, L106-107: indicate the number of samples/pots per species; throughout the manuscript you refer to examining two saplings; from your results it is clear that you studied more than two saplings; be more precise in describing the methods, so there is no confusion;

Response: Thanks for your constructive suggestions. On the basis of two different methods determining instantaneous water use efficiency and the related $\delta^{13}C$ involved, we will add the specific number of measurements, leaves, and saplings of each species in one orthogonal treatment for each method, which corresponds to the repeats n=32 in the results analysis as follows:

On Page 3, Lines 113-116: "Saplings of two species that have similar ground diameters, heights, and growth statuses were selected. One sapling from two species was placed in one pot (22 cm in

diameter and 22 cm in height). Undisturbed soil samples were collected from the field, sieved (with all particles >10 mm removed), and placed into the pots".

On Page 4, Lines 144-153: "While undergoing 20 groups of orthogonal treatments for $[CO_2] \times$ SWC, the saplings were ready for investigation. Due to one chamber only containing five plant-pots (per species) and one pot one SWC level under one $CO_2$ concentration, two saplings per specie in one orthogonal treatment were replicated for two periods, respectively. Each period per orthogonal treatment continued for 7 days. Pots were rearranged periodically to minimize non-uniform illumination. All orthogonal tests were formed as: elevated $CO_2$ concentration gradient for $C_{400}$ (during June 2–9, June 12–19, June 21–28, and July 2–9, 2015, $C_{400}$), $C_{500}$ (during July 11–18, July 22–29, August 4–11, and August 15–22, 2015, $C_{500}$), $C_{600}$ (during June 2–9, June 12–19, June 21–28, and July 2–9, 2015, $C_{600}$), and $C_{800}$ (during July 11–18, July 22–29, August 4–11, and August 15–22, 2015, $C_{800}$), combined with a soil-water gradient for 35%–45% of FC, 50%–60% of FC, 60%–70% of FC, and 70%–80% of FC and 100% FC (CK)".

On Page 4, Lines 156-158: "Two saplings per specie were replicated per treatment (SWC×$[CO_2]$). For each sapling, four leaves were chosen and then four measurements were conducted on each leaf".

On Pages 4-5, Lines 164-169: "Recently-expanded, eight sun leaves per sapling were selected and homogenized in liquid nitrogen since the gas-exchange measurements accomplished. For the extraction of the water-soluble compounds (WSCs) from the leaves (Gessler et al., 2004), 50 mg of ground leaves and 100 mg of PVPP (polyvinylpolypyrrolidone) were mixed and incubated in 1mL double demineralized water for 60 min at 5℃ in a centrifuge tube. Each leaf was replicated two times. Two saplings per specie were chosen for each orthogonal treatment".

*P3, L115 (and other places in the manuscript, e.g., P4, L161, P5, L182, and P6, L214): Never start a sentence with a symbol, a number, or an acronym. Please spell out each time when used at the start of a sentence; make changes throughout the manuscript.

Response: Based on your suggestions about the writing form, we have rephrased the sentences with the meaning unchanged on Page 4, Lines 125-126 "The central controlling system of the chambers (FH-230) can timely monitor and control the $CO_2$ concentration".

On Pages 4-5, Lines 165-168: "For the extraction of the water-soluble compounds (WSCs) from the leaves (Gessler et al., 2004), 50 mg of ground leaves and 100 mg of PVPP (polyvinylpolypyrrolidone) were mixed and incubated in 1mL double demineralized water for 60 min at 5℃ in a centrifuge tube".

On Page 5, Lines 190-192: "…; $a$ is the discrimination dependent on a fraction factor (4‰) and $b$ refers to the discrimination during $CO_2$ fixation by ribulose 1,5- bisphosphate carboxylase/oxygenase (Rubisco) and internal diffusion (30‰)".

On Pages 7-8, Lines 261-272: "Saplings of *P. orientalis* and *Q. variabilis* were exposed to the orthogonal treatments. When SWC increased, $P_n$, $g_s$ and $T_r$ in *P. orientalis* and *Q. variabilis* peaked at 70%–80% of FC or/and FC (Fig. 2). The $C_i$ in *P. orientalis* rose as SWC increased, while it peaked at 60%–70% of FC and declined thereafter with increased SWC in *Q. variabilis*. The capacity of carbon uptake and $C_i$ were elevated significantly by elevated $[CO_2]$ at any given SWC for two species ($p<0.05$). Furthermore, greater increasing magnitudes of $P_n$ in *P. orientalis* were found at 50%–70% of FC from $C_{400}$ to $C_{800}$, which was at 35%–45% of FC in *Q. variabilis*. As the water stress was alleviated (at 70%–80% of FC and FC), the reduction of $g_s$ in *P. orientalis* was more pronounced with elevated $[CO_2]$ at a given SWC ($p<0.01$). Nevertheless, $g_s$ of *Q.*

*variabilis* in $C_{400}$, $C_{500}$, and $C_{600}$ was significantly higher than that in $C_{800}$ at 50%−80% of FC ($p<0.01$). Coordinated with $g_s$, $T_r$ of two species in $C_{400}$ and $C_{500}$ was significantly higher than that in $C_{600}$ and $C_{800}$ except for 35%−60% of FC ($p<0.01$, Figs. 2g and 2h). Larger $P_n$, $g_s$, $C_i$ and $T_r$ of *Q. variabilis* was significantly presented than that of *P. orientalis* ($p<0.01$, Fig. 2)".

*Redundancy throughout the manuscript should be removed (e.g., P4, L123-124, P5, L186, P5, L198-199, and other places in the manuscript); on P5, L186 you defined Pn and Tr (you also define the terms on P4); no need to do repeat;

Response: Based on your suggestions, we have removed the redundancy throughout the manuscript, which has been mentioned or defined as discussed before.

*P4, L127: "It consisted of the water…" should be "It consisted of a water…";

Response: Based on your suggestions, we have changed the "the" to "a" on Page 4, Line 133 and examined similar mistakes throughout the manuscript.

*P4, L130: "…specific soil water…", does this refer to the "…specific soil water content.." or something else? Please specify.

Response: We thank your suggestion and have specified this presentation as "…, target soil volumetric water content (SWC) could be set and monitored by soil moisture sensors" on Page 4, Lines 136-137 in the revised manuscript.

*P4, L131-132, L138: awkward phrasing, e.g., "the chamber" does not "determine"; please rephrase both sentences;

Response: Considering your comments, we have rephrased the sentence on Page 4, Lines 137-138:

"Since timely SWC could be sensed by the sensors, the automatic irrigation device can be regulated to water or stop watering the plants".

And the sentence on Page 4, Line 137 of unrevised manuscript is unnecessary to detail under the meaning unchanged and have been removed in the revised manuscript.

*P4, L140: what do you mean by "equilibrium circumstances"? Please rephrase;

Response: Thanks for your suggestions and we have rephrased this sentence on Page 4, Lines 144-145 in the revised manuscript as "While undergoing 20 groups of orthogonal treatments for $[CO_2] \times SWC$, the saplings were ready for investigation".

*P4, L141: "investigation" or "sampling"?

Response: We have substituted the word "sampling" for "investigation" on Page 4, Line 145 of revised manuscript.

*P4, L144-146: not a proper sentence; please rephrase;

Response: Following your helpful comments, we have rephrased the sentence on Page 4, Lines 148-153 of revised manuscript:

"All orthogonal tests were formed as: elevated $CO_2$ concentration gradient for $C_{400}$ (during June 2–9, June 12–19, June 21–28, and July 2–9, 2015, $C_{400}$), $C_{500}$ (during July 11–18, July 22–29, August 4–11, and August 15–22, 2015, $C_{500}$), $C_{600}$ (during June 2–9, June 12–19, June 21–28, and July 2–9, 2015, $C_{600}$), and $C_{800}$ (during July 11–18, July 22–29, August 4–11, and August 15–22, 2015, $C_{800}$), combined with a soil-water gradient for 35%−45% of FC, 50%−60% of FC, 60%−70% of FC, and 70%−80% of FC and 100% FC (CK)".

*P4, L149-150: "7-day cultivation in the chambers."; no need to include "in the chambers", this is obvious;

Response: Thanks for pointing the inappropriate sentence, and we accepted the suggestion and

removed "in the chambers" on Page 4, Line 156 of revised manuscript.

*P4, L148-155 (and other parts of the manuscript): it appears you use two different symbols for the same thing, (i.e., Tr and E for transpiration); please eliminate one of the symbols and replace with the one you decided to go with;

Response: Based on your helpful comments, we have checked the errors and uniformed the symbol for the same meaning throughout the whole article.

*P4, L164: "12000 xg", please specify;

Response: We made the mistake in writing the unit of centrifugal force under the high speed centrifugation and have rewritten it as "(12000 $\times g$ for 5 min, $g$ represents one gravity)" on Page 5, Line 171 of revised manuscript.

*P5, 167: "analyzed in the mass…" should be "analyzed with a mass…";

Response: Thank you for pointing the improper use of preposition and we have corrected them on Page 5, Line 174 of revised manuscript according your suggestion.

*P5, L168 (and other places in the manuscript): "are" should be "were"; do not change verb tense within the same paragraph;

Response: Thanks for your helpful suggestions. Due to the expression of carbon isotope signatures and its related equation (Equation 1) have been defined and are commonly recognized, we will descript this part in a separate paragraph on page 5, lines175-180. Meanwhile, we have changed "was" into "is" to keep the same verb tense within the same paragraph on Page 5, line 184 in the revised manuscript.

*P5, L188: "intercellular" what?

Response: Thanks for your careful reminder. We have rephrased this part under its meaning unchanged on Page 5, Lines 194-196 of revised manuscript:

"…where 1.6 is the diffusion ratio of stomatal conductance to water vapor to $CO_2$ in the chamber and $\Delta e$ is the difference between $e_{lf}$ and $e_{atm}$ that represent the extra- and intra-cellular water vapor pressure, respectively:…".

*P5, L186-195: sentence structure is awkward; please address;

Response: Considering your suggestions, we have modified and rephrased this part to read (starting on Pages 5-6, Lines 194-202):

"where 1.6 is the diffusion ratio of stomatal conductance to water vapor to $CO_2$ in the chamber and $\Delta e$ is the difference between $e_{lf}$ and $e_{atm}$ that represent the extra- and intra-cellular water vapor pressure, respectively:

$$\Delta e = e_{lf} - e_{atm} = 0.611 \times e^{17.502T/(240.97+T)} \times (1 - \text{RH}) \tag{5}$$

where $T$ and RH are the temperature and relative humidity on leaf surface, respectively. Combining Eqns. (2, 3 and 4), the instantaneous water use efficiency could be determined by the $\delta^{13}C_{WSC}$ of leaves, defined as $\text{WUE}_{cp}$:

$$\text{WUE}_{cp} = \frac{P_n}{T_r} = (1 - \varphi)(C_a - C_i)/1.6\Delta e = C_a(1 - \varphi)\left[\frac{b - \delta^{13}C_a + (b+1)\delta^{13}C_{WSC}}{(b-a)(1 + \delta^{13}C_{WSC})}\right]/1.6\Delta e \tag{6}$$

where $\varphi$ is the respiratory ratio of leaf carbohydrates to other organs at night (0.3)".

*P5, L201 and P6, L212: "…70%-80% of FC and FC"; I'm not sure how to interpret this; please clarify;

Response: Thanks for your comments. On Page 7, Line 262 and 267 of revised manuscript "…70%-80% of FC and FC" is that the photosynthetic parameters of plants peaked at two SWC

levels, 70%-80% of FC and 100% FC. We have rewritten the "FC (CK)" as "100% FC" to read throughout the revised manuscript.

\*P6, L204: remove "magnitude of";

Response: We accept your helpful suggestion and have removed the "magnitude of" on Page 7, Line 266 in the revised manuscript.

\*P6, L217 (and other places in the manuscript): "maximums" should be "maxima";

Response: Based on your comments, we have examined the similar errors and changed the "maximums" to "maxima" within the whole article.

\*P6, L219: "elevated" or "increased"?

Response: Considering your helpful suggestion, we have changed the "elevated" to "increased" on Page 8, Line 287 and Page 9, Line 339 of revised manuscript.

\*P6, L220 & L225 (second "was") and P7, L259: remove "was";

Response: Thanks for pointing out the redundancy "was" and we have removed the second "was" on Page 7, Lines 271-272 and Page 8, Lines 300-301 of revised manuscript.

\*Clarify the sampling methodology; two saplings?

Response: Considering your helpful suggestion, we have presented the sampling methodologies of two methods, respectively.

On Page 3, Lines 113-116: "Saplings of two species that have similar ground diameters, heights, and growth statuses were selected. One sapling from two species was placed in one pot (22 cm in diameter and 22 cm in height). Undisturbed soil samples were collected from the field, sieved (with all particles >10 mm removed), and placed into the pots".

On Page 4, Lines 144-153: "While undergoing 20 groups of orthogonal treatments for $[CO_2] \times$ SWC, the saplings were ready for investigation. Due to one chamber only containing five plant-pots (per species) and one pot one SWC level under one $CO_2$ concentration, two saplings per specie in one orthogonal treatment were replicated for two periods, respectively. Each period per orthogonal treatment continued for 7 days. Pots were rearranged periodically to minimize non-uniform illumination. All orthogonal tests were formed as: elevated $CO_2$ concentration gradient for $C_{400}$ (during June 2–9, June 12–19, June 21–28, and July 2–9, 2015, $C_{400}$), $C_{500}$ (during July 11–18, July 22–29, August 4–11, and August 15–22, 2015, $C_{500}$), $C_{600}$ (during June 2–9, June 12–19, June 21–28, and July 2–9, 2015, $C_{600}$), and $C_{800}$ (during July 11–18, July 22–29, August 4–11, and August 15–22, 2015, $C_{800}$), combined with a soil-water gradient for 35%–45% of FC, 50%–60% of FC, 60%–70% of FC, and 70%–80% of FC and 100% FC (CK)".

On Page 4, Lines 156-158: "Two saplings per specie were replicated per treatment (SWC× $[CO_2]$). For each sapling, four leaves were chosen and then four measurements were conducted on each leaf".

On Pages 4-5, Lines 164-169: "Recently-expanded, eight sun leaves per sapling were selected and homogenized in liquid nitrogen since the gas-exchange measurements accomplished. For the extraction of the water-soluble compounds (WSCs) from the leaves (Gessler et al., 2004), 50 mg of ground leaves and 100 mg of PVPP (polyvinylpolypyrrolidone) were mixed and incubated in 1mL double demineralized water for 60 min at 5℃ in a centrifuge tube. Each leaf was replicated two times. Two saplings per specie were chosen for each orthogonal treatment".

\*P6, L240: redundant (see P4); please remove;

Response: According your suggestion, we have removed the sentence starting on Page 8, Line 285 in the revised manuscript.

\*P6, L243: change "reduced" to "decreased";

Response: Thanks for your helpful comments. We have changed "reduced" to "decreased" on Page 8, Line 287 in the revised manuscript.

\*P6, L244: remove "remarkably";

Response: Thanks for your helpful comments. We have removed "remarkably" on Page 8, Line 287 in the revised manuscript.

\*P6, L244-P7, 245: awkward structure; please revise;

Response: Considering your constructive suggestion, we have rephrased the sentence on Page 8, Lines 287-289 in the revised manuscript "Differing from variation in $WUE_{ge}$ of *P. orientalis* with soil moistened, $WUE_{ge}$ in *Q. variabilis* were improved slightly at 100% FC in $C_{600}$ or $C_{800}$ (Fig. 4b)".

\*P7, L246-247: rewrite "…orthogonal treatments; this was also observed in Q. variabilis.";

Response: Based on your helpful comments, we have written the sentence on Page 8, Lines 289-290 in the revised manuscript "The maximum of $WUE_{ge}$ thus occurred at 35%−45% of FC in $C_{800}$ among all orthogonal treatments for *P. orientalis*; this was also observed in *Q. variabilis*".

\*P7, L249: "most saplings" is ambiguous; please be more specific;

Response: According your suggestion, we have specified the number of *P. orientalis* which have the greater $WUE_{ge}$ than did *Q. variabilis* as on Page 8, Lines 292-293 of revised manuscript: "Thirty-two saplings of *P. orientalis* had greater $WUE_{ge}$ than did *Q. variabilis* between the same $[CO_2] \times SWC$ treatments ($p<0.05$)".

\*P7, L254-255: from "while the…" to the end of the sentence is awkward, please revise;

Response: Thanks for your helpful suggestion, we have rephrased the sentence on Page 8, Lines 295-297 of revised manuscript: "As illustrated in Fig. 5a, $WUE_{cp}$ of *P. orientalis* in $C_{600}$ or $C_{800}$ climbed up as water stress alleviated beyond 50%−60% of FC, as well as that in $C_{400}$ or $C_{500}$ while SWC exceeding 60%−70% of FC".

\*P7, L269 and 270: the word "coefficients" is non-descriptive; please elaborate as to which coefficients are being referred to?

Response: Considering your helpful suggestion and the first reviewer's comments, we have added the mesophyll conductance together with post-carboxylation fractionation to explain the $^{13}C$ fractionation from the site of carboxylation to cytoplasm before sugars transportation, defined as "total $^{13}C$ fractionation". The total $^{13}C$ fractionation in the revised manuscript is supposed to be consisted of the fractionations from mesophyll conductance and post-carboxylation. In the unrevised manuscript, the "coefficients" represented the fractionation from the site of carboxylation to cytoplasm before sugars transportation as "post-carboxylation fractionation" without considering the mesophyll conductance. Consequently, we have redefined the $^{13}C$ fractionation from the site of carboxylation to cytoplasm before sugars transportation as "total $^{13}C$ fractionation, which is composed by the fractionations from mesophyll conductance and post-carboxylation, and hence the "coefficients" in the previous version is equal to "total $^{13}C$ fractionation" throughout the whole revised manuscript.

\*P7, L273, i.e., "Stoma are the …": An obvious point; no need to state;

Response: Thanks for your suggestion. We have removed that sentence on Page 9, Line 342 of revised manuscript.

\*P7, L283: "under any" or "irrespective of"?

Response: Thanks for your helpful proposal. We have substituted the "irrespective of" for "under any" on Page 9, Lines 353 in the revised manuscript.

*P7, L284-285: "maximal values… were generated successively…", not clear, please clarify;

Response: Considering your helpful suggestion, we have rephrased this sentence on Pages 9, Lines 354-355 in the revised manuscript:

"The decrease of $g_s$ responding to the elevated $[CO_2]$ could be mitigated by the coupling effects of soil wetting".

*P8, L296: remove the word "intensive";

Response: Considering your helpful suggestion, we have removed "intensive" on Page 10, Line 363 in the revised manuscript.

*P8, L306: "an evident" or "a clear"?

Response: Based on your helpful suggestion, we have changed the "an evident" to "a clear" and rephrased the sentence to read on Page 10, Lines 371-373 in the revised manuscript:

"…demonstrating that there was a clear irrigation maximum of SWC below which the plant could manage itself to adjust changing environment".

*P8, L308: "of" or "in"?

Response: Thanks for your suggestion. We have changed "of" to "in" on Page 10, Line 374 in the revised manuscript.

*P8, L311: "proves" or "suggests"?

Response: Based on your helpful suggestion, we have changed "proves" to "suggests" on Page 10, Line 377 in the revised manuscript.

*P9, L330-331: awkward, please revise

Response: Thanks for your helpful suggestion. Due to the revised manuscript will focus on discussing the causes of total $^{13}C$ fractionation that is composed by mesophyll conductance and post-carboxylation, we have removed the discussions about the $\delta^{13}C_{WSC}$ of two species under orthogonal treatments. So the sentence on Lines 330-331 of unrevised manuscript has been removed in the revised manuscript.

*P9, L334: "profoundly" or "greatly"?

Response: Thanks for your helpful suggestion. Due to the revised manuscript will focus on discussing the causes of total $^{13}C$ fractionation that is composed by mesophyll conductance and post-carboxylation, we have removed the discussions about the $\delta^{13}C_{WSC}$ of two species under orthogonal treatments. So the sentence on Lines 334 of unrevised manuscript has been removed in the revised manuscript.

*P9, L339: change to "followed by a reduction in Tr";

Response: Considering your helpful suggestion, we have changed "followed by the reduction of $T_r$" "followed by a reduction in $T_r$" on Page 10, Line 383 of revised manuscript.

*P9, L345: change to "than in conifers";

Response: Thanks for your helpful suggestion. We have changed "than is that of conifers" to "than in conifers" on Page 10, Line 388 of revised manuscript.

*P9, L350: awkward, please rephrase;

Response: Considering your helpful suggestion, we have rephrased the previous sentence as "The $WUE_{ge}$ of *P. orientalis* and *Q. variabilis* was enhanced with soil drying, as presented by Parker and Pallardy (1991), DeLucia and Heckathorn (1989), Reich et al. (1989), and Leakey (2009)" on Page 10, Line 390-392 in the revised manuscript.

\*P9, L354-355: not sure the significance of the sentence; please address;

Response: Based on your helpful suggestion, we have rephrased the sentence on Page 10, Lines 395-396 in the revised manuscript:

"Pons et al. (2009) reviewed that $\Delta$ of leaf soluble sugar is coupled with environmental dynamics over a period ranging from a few hours to 1–2 d".

\*P9, L356: "synthetically"; not sure what this means in the context of the rest of the sentence; please revise;

Response: Considering your helpful suggestion, we have rephrased the previous sentence as "The $WUE_{cp}$ of our materials could respond to $[CO_2] \times SWC$ treatments over cultivated days, whereas $WUE_{ge}$ is characterized as the instantaneous physiology of plants to conditions" on Page 10, Lines 396-398 of revised manuscript.

\*P10, L394: remove the first "was";

Response: Thanks for your helpful suggestion. We have removed the first "was" in the sentence on Page 12, Line 463 of revised manuscript.

\*P10, L410: "cautious descriptions" or "clear description";

Response: Considering your helpful suggestion, we have changed "cautious descriptions" to "clear description" on Page 12, Line 472 of revised manuscript.

\*Figure captions of Fig. 2-6: captions can be simplified; identify individual graphs within a composite figure with a lower-case letter in the upper left hand corner of the graph and cite in the caption; e.g., for Fig. 6 "Regression between stomatal conductance and 13C fractionation coefficient of P. orientalis (a) and Q. variabilis (b) for four CO2 concentrations X five soil volumetric water contents (p=0.05, n=32)." Because the caption identifies the individual graphs according to species, there is no need to identify the species in the graph. Figure captions of Fig. 2-5 can be treated in a similar fashion.

Response: Thanks for your helpful suggestion. We have simplified and revised the captions of Figs. 2-5 and 7 shown in the revised manuscript, and have numbered and named the individual graphs within a composite figure with a lower-case letter in the upper left hand corner of the graph and cite in the simplified caption. Considering the first referee's comments about the supplement of mesophyll conductance in results and discussions, we have added Figure 6 that illustrates the mesophyll conductance of two species in orthogonal treatments, and Figure 8 that presents the regression between mesophyll conductance and total $^{13}C$ fractionation of two species under orthogonal treatments in the revised manuscript:

"**Figure 2.** Net photosynthetic rates ($P_n$, µmol m$^{-2}$ s$^{-1}$, a and b), stomatal conductance ($g_s$, mol H$_2$O m$^{-2}$ s$^{-1}$, c and d), intercellular CO$_2$ concentration ($C_i$, µmol CO$_2$ mol$^{-1}$, e and f), and transpiration rates ($T_r$, mmol H$_2$O m$^{-2}$ s$^{-1}$, g and h) of P. orientalis and Q. variabilis for four CO$_2$ concentrations $\times$ five soil volumetric water contents. Means $\pm$ SDs, n = 32".

"**Figure 3.** Carbon isotope composition of water-soluble compounds ($\delta^{13}C_{WSC}$) extracted from leaves of P. orientalis (a) and Q. variabilis (b) for four CO$_2$ concentrations $\times$ five soil volumetric water contents. Means $\pm$ SDs, n = 32".

"**Figure 4.** Instantaneous water use efficiency through gas exchange measurements ($WUE_{ge}$) for leaves of P. orientalis (a) and Q. variabilis (b) for four CO$_2$ concentrations $\times$ five soil volumetric water contents. Means $\pm$ SDs, n = 32".

"**Figure 5.** Instantaneous water use efficiency estimated by $\delta^{13}C$ of water-soluble compounds ($WUE_{cp}$) from leaves of P. orientalis (a) and Q. variabilis (b) for four CO$_2$ concentrations $\times$ five

soil volumetric water contents. Means ±SDs, n = 32".

"**Figure 6.** Mesophyll conductance of *P. orientalis* (a) and *Q. variabilis* (b) for four $CO_2$ concentrations ×five soil volumetric water contents. Means ±SDs, n = 32".

"**Figure 7.** Regression between stomatal conductance and total $^{13}C$ fractionation of *P. orientalis* (a) and *Q. variabilis* (b) for four $CO_2$ concentrations ×five soil volumetric water contents ($p$=0.01, n = 32)".

"**Figure 8.** Regression between mesophyll conductance and total $^{13}C$ fractionation of *P. orientalis* (a) and *Q. variabilis* (b) for four $CO_2$ concentrations ×five soil volumetric water contents ($p$=0.01, n = 32)".

*Table 1: increase the font size of some of the text/numbers in the Table

Response: Considering your precious suggestions combining the comments given by the first two referees, Table 1 has been recreated to support the above-mentioned descriptions and analysis in the revised manuscript. In order to display the Table more clearly, we use the horizontal direction paper in the page layout and have increased the font size of the text/numbers in the revised Table. Correspondingly, we have added the results and discussions for the mesophyll conductance and the contribution of post-carboxylation fractionation on the total $^{13}C$ fractionation in the revised manuscript.

---

## Editor Decision (ED1)

Title:  Interaction of CO2 concentrations and water stress in semi-arid  plants causes diverging response in instantaneous water use efficiency and carbon isotope composition

Abstract: (changes are in red)
L10-14: It is commonly reported that 13C fractionation occurs as CO2-gas diffuses from the atmosphere to the sub-stomatal cavity. Few researchers have investigated 13C fractionation at the site of carboxylation to cytoplasm before sugars are exported outward from the leaf. This process typically progresses in response to variations in environmental conditions (i.e., CO2 concentrations and water stress), including in their interaction.

L14-17: Therefore, saplings of two typical plant species found growing in semi-arid areas of Northern China of similar growing status—*Platycladus orientalis* and *Quercus variabilis*—were selected and cultivated in growth chambers with orthogonal treatments (four CO2 concentrations [CO2] × five soil volumetric water content (SWC)).

L23-24: Differences in instantaneous water use efficiency (iWUE) according to distinct environmental changes differed between the two species.

L24-28: The WUEge in *P. orientalis* was significantly greater than that in *Q. variabilis*, while an opposite trend was observed when comparing WUEcp between the two species. Total 13C fractionation at the site of carboxylation to cytoplasm before sugar export (total 13C fractionation) was clearly species-specific, as demonstrated in the interaction of [CO2] and SWC.

L28-30: Rising [CO2] coupled with moistened soil generated increasing disparities in $\delta$13C between  water-soluble compounds ($\delta$13C*WSC*) and estimates based on gas-exchange observations ($\delta$13C*obs*) in *P. orientalis*, ranging between 0.0328‰–0.0472‰.

L34-37: Total 13C fractionation was linearly dependent on *gs*, indicating post-carboxylation fractionation could be attributed to environmental variation. Thus, clear description of magnitude and environmental dependence of apparent post-carboxylation fractionation is worth our attention when addressing photosynthetic fractionation.

Introduction:
**Change**
L42: 'together with' to 'culminating in'
L43: 'low water availability' to 'dryness'
L50 'environmental changes and their influences' to 'environmental change and their influence'
L51: 'While the depletion' to 'While depletion'
L52: 'itself might also affect the $\delta$13C of plant organs' to 'itself may affect $\delta$13C of plant organs'
L53: 'climatic change' to 'changes in climate'
L55: 'Discrimination against' to 'Discrimination of'

L57-58: 'even the mesophyll conductance derived from the difference of CO2 concentrations between intercellular site and chloroplast (Farquhar et al., 1982; Cano et al., 2014)' the addition of this segment of text does not fit well with the preceding text, please rewrite
L67: change 'the carbon discriminations that follow' to 'the carbon discrimination that follows'
L77: misspelt Farquhar's name, please fix

L82; 'for the differences from' to 'for the differences in the'

L87: change 'magnitude of these carbon fractionations are related to environmental variation have not yet been investigated.' to 'magnitude of carbon fractionation is related to environmental variation that has yet to be fully investigated.'

L94-95: 'However, there is a dispute whether the fractionation stemmed…' to ' However, there is disagreement whether fractionation stemming…'

L97-99: awkward, please rewrite

L103: at the first mention of the growth chamber (use the full citation that you provide on L120-121)

L122-123: 'daytime temperature in chambers was set to $25 \pm 0.5$℃ from 07:00 122 to 17:00, and the night-time temperature was $18 \pm 0.5$℃ from 17:00 to 07:00' to 'daytime and nighttime temperatures in the chambers was set to $25 \pm 0.5$℃ from 07:00 to 17:00 and $18 \pm 0.5$℃ from 17:00 to 07:00'

Omit L 131 & 132.

L141-144: can this be simplified?

L148-154: can this also be simplified? Can you put this detail and the detail above in a table?

L165-166: this needs revising

L179: second $R_{sample}$ needs to be change to $R_{standard}$

Throughout the manuscript: usage of $CO_2$ concentration, sometimes you use $[CO_2]$ and other times you spell it out; try to be consistent; since you introduced $[CO_2]$ why not continue to use it? The labels on some of the Figures are simply too small; please fix

What I provide above are some problems that I was able to identify, without having to address every line of the manuscript. There are many more problems with the writing and I would suggest that you get professional editing help in rewriting the manuscript. A lot of the problems I identify are associated with grammar and ways of expression. The three referees that I had review your manuscript all agree that the material is publishable based on scientific merit. However, I feel the manuscript needs considerable work to make it stand out. I will give you opportunity to fix the problems. I would like to see the revised manuscript again before making a final decision.

---

## Author Response (AR2)

Response list to the Editor's comments

Ref: doi:10.5194/bg-2016-372

Title: **Differences in instantaneous water use efficiency derived from post-carboxylation fractionation respond to the interaction of $CO_2$ concentrations and water stress in semi-arid areas**

Authors: Na Zhao, Ping Meng, Yabing He, Xinxiao Yu*

Dear Editor,

Thanks for your thoughtful and constructive comments that provide scientific guidance for our writing and future research. We commissioned the LetPub Company (belonging to ACCDON (US) that is the professional editorial team) to provide professional editing help in rewriting the manuscript. We have been carefully considering your suggestions and revising the manuscript in the revised manuscript (marked in red color) accordingly. In addition to the following issues, we have corrected other mistakes with grammar and expression in the revised manuscript (marked in red color). The following below in blue are our point-to-point responses for your questions and comments. We are appreciated for your kind help on writing.

We are looking forward to your further comments and a possible publication in the BG special issue (Ecosystem processes and functioning across current and future dryness gradients in arid and semi-arid lands).

Kind regards,

Xinxiao Yu

■■■■■■■■■■■■■■■■■■■■■■■■■■■■■■■■■■■■■■■■■■■■■■■■■■■■■■■■■■■■

Comments to the Author:

*Thank you for the re-submission of your manuscript. The three referees were unanimous in their support of the scientific content of the paper. I, however, have considerable difficulty in reaching a final decision regarding the publication of the paper, because of the quality of the writing. I would suggest that you get professional editing help in rewriting the manuscript. A lot of the problems I identify are associated with grammar and ways of expression. I will give you opportunity to fix these problems. I would like to see the revised manuscript before making a final decision. To help you with fixing some of these problems, I provide some guidance.

Response: Thank you for the careful review and constructive comments. We apologize for any inconvenience that we bring you for my carelessness in writing. Based on your helpful suggestions and professional editing help from Letpub Company, revisions throughout the whole article have been made and the results have been improved and supplemented with the related contents.

Title:  Interaction of $CO_2$ concentrations and water stress in semi-arid  **plants** causes diverging response in instantaneous water use efficiency and carbon isotope composition

Response: We appreciate your helpful comments. Based on your constructive recommendation, the title was changed as "Interaction of $CO_2$ concentrations and water stress in semi-arid plants causes diverging response in instantaneous water use efficiency and carbon isotope composition" on Lines 11-16, Page 1 in the revised manuscript.

L14-17: Therefore, saplings of two typical plant species found growing in semi-arid areas of Northern China of similar growing status—*Platycladus orientalis* and *Quercus variabilis*—were selected and cultivated in growth chambers with orthogonal treatments (four CO2 concentrations [$CO_2$] ×five soil volumetric water contents (SWC)).

Response: Considering your suggestions, we have modified and rephrased this part to read (starting on Page 1, Lines 16-19):

"Therefore, saplings of two typical plant species (*Platycladus orientalis* and *Quercus variabilis*) from semi-arid areas of Northern China were selected and cultivated in growth chambers with orthogonal treatments (four $CO_2$ concentrations [$CO_2$] × five soil volumetric water contents (SWC))".

L23-24: Differences in instantaneous water use efficiency (iWUE) according to distinct environmental changes differed between the two species.

Response: Thank you for pointing the ambiguous use of preposition and we have corrected them on Lines 25-26, Page 1 of revised manuscript:

"Instantaneous water use efficiency (iWUE) according to environmental changes, differed between the two species".

L24-28: The $WUE_{ge}$ in *P. orientalis* was significantly greater than that in *Q. variabilis*, while an opposite trend was observed when comparing $WUE_{cp}$ between the two species. Total $^{13}$C fractionation at the site of carboxylation to cytoplasm before sugar export (total $^{13}$C fractionation) was clearly species-specific, as demonstrated in the interaction of [$CO_2$] and SWC.

Response: Considering your helpful suggestion, we have rewritten this part on Lines 26-29, Page 1 of the revised manuscript:

"The $WUE_{ge}$ in *P. orientalis* was significantly greater than that in *Q. variabilis*, while an opposite trend was observed when comparing $WUE_{cp}$ between the two species. Total $^{13}$C fractionation at the site of carboxylation to cytoplasm before sugar export (total $^{13}$C fractionation) was species-specific, as demonstrated in the interaction of [CO2] and SWC".

L28-30: Rising [$CO_2$] coupled with moistened soil generated increasing disparities in $\delta^{13}$C between the water-soluble compounds ($\delta^{13}C_{WSC}$) and estimates based on gas-exchange observations ($\delta^{13}C_{obs}$) in *P. orientalis*, ranging between 0.0328‰–0.0472‰.

Response: We agree with your suggestion and have revised this sentence on Page 1, Lines 29-32 in the revised manuscript.

L34-37: Total $^{13}$C fractionation was linearly dependent on $g_s$, indicating post-carboxylation fractionation could be attributed to environmental variation. Thus, clear description of magnitude and environmental dependence of apparent post-carboxylation fractionation is worth our attention when addressing photosynthetic fractionation.

Response: According your helpful comments, we have revised this part as "Total $^{13}$C fractionation was linearly dependent on stomatal conductance, indicating post-carboxylation fractionation could be attributed to environmental variation. The magnitude and environmental dependence of apparent post-carboxylation fractionation is worth our attention when addressing photosynthetic fractionation" on Lines 35-38, Page 1 in the revised manuscript.

Due to redundant expression of 'Since the onset of industrial revolution', we changed that into 'Since the industrial revolution' on Line 42 Page 2 in the revised manuscript.

L42: 'together with' to 'culminating in'
Response: Based on your suggestion, we have substituted 'together with' into 'culminating in' on Line 43, Page 2 in revised manuscript.

L43: 'low water availability' to 'dryness'
Response: We have replaced 'low water availability' into 'dryness' on Line 43 Page 2 in the revised manuscript.

We have replaced 'trigger an ongoing' into 'exacerbate the' on Line 44 Page 2 in the revised manuscript.

On Lines 45-46, Page 2 of revised manuscript, '…not only lead to fluctuations in global patterns of precipitation, but will amplify drought in arid regions, and lead to more frequent extreme drought events in humid regions' has been rewritten as 'increase fluctuations in global precipitation patterns, but will probably amplify drought frequency in arid regions, and lead to more frequent extreme events in humid regions'.

On Lines 47-49, Page 2 of revised manuscript, the sentence were rewritten as 'mean $\delta^{13}C$ of atmospheric $CO_2$ is currently being depleted by 0.02‰–0.03‰ year$^{-1}$ (CU-INSTAAR/NOAACMDL network for atmospheric $CO_2$; http://www.esrl.noaa.gov/gmd/)'.

We have simplified the expression as 'The current carbon isotopic composition may respond to' on Line 50, Page 2 of revised manuscript.

L50 'environmental changes and their influences' to 'environmental change and their influence'
Response: Based on your suggestion, we have changed the 'changes…influences' into 'change…influence' on Line 50, Page 2 of revised manuscript.

L51: 'While the depletion' to 'While depletion'
Response: According your advice, we removed 'the' before the noun on Line 52, Page 2 of revised manuscript.

We changed the '…has been shown…' into 'is occurring' on Line 52, Page 2 of revised manuscript.

L52: 'itself might also affect the δ13C of plant organs' to 'itself may affect δ13C of plant organs'
Response: Considering your helpful suggestions, we have changed '…itself might also affect the $\delta^{13}C$ of plant organs' into '…may affect $\delta^{13}C$ of plant organs' on Line 53, Page 2 of revised manuscript.

L53: 'climatic change' to 'changes in climate'
Response: Based on your consideration, we have rewritten this sentence as 'in turn, are responding physiologically to changes in climate (Gessler et al., 2014)' on Lines 53-54, Page 2 of revised manuscript.

L55: 'Discrimination against' to 'Discrimination of'
L57-58: 'even the mesophyll conductance derived from the difference of CO2 concentrations between intercellular site and chloroplast (Farquhar et al., 1982; Cano et al., 2014)' the addition of this segment of text does not fit well with the preceding text, please rewrite
Response: Thanks for your helpful advices. We rewrote this sentence as 'Discrimination of $^{13}C$ in leaves relies mainly on environmental factors that affect the ratio of intercellular to ambient $CO_2$

concentration ($C_i/C_a$). Rubisco activities and the mesophyll conductance derived from the difference of $CO_2$ concentrations between intercellular sites and chloroplasts are also involved (Farquhar et al., 1982; Cano et al., 2014)' on Lines 56-59, Page 2 of revised manuscript.

We have changed 'As changes' into 'Changes', '…, they are expected to be recorded differentially in the $\delta^{13}C$ of water-soluble compounds ($\delta^{13}C_{WSC}$) of the different plant organs' into '… and they will be recorded differentially in the $\delta^{13}C$ of water-soluble compounds ($\delta^{13}C_{WSC}$) in different plant organs', and '… has been described and reviewed elsewhere' into 'has been reviewed elsewhere' on Lines 59-63, Page 2 of revised manuscript.

We have rearranged '…, which determines' as '…that determine' on Lines 64-65, Page 2 of revised manuscript. And '…, defined as…' was rewritten as '.These are defined as …' on Lines 65-67, Page 2 of revised manuscript.

L67: change 'the carbon discriminations that follow' to 'the carbon discrimination that follows'

Response: Based on your suggestions, we have changed 'the carbon discriminations that follow' into 'the carbon discrimination that follows' on Lines 67-68, Page 2 of revised manuscript.

We have rewritten '…, what should also be considered' as '…, another consideration…' on Line 76, Page 2 of revised manuscript.

L77: misspelt Farquhar's name, please fix

Response: I am very sorry for my careless in spelling and have corrected the spelling mistake as …'Farquhar's …' on Line 77, Page 2 of revised manuscript.

We have simplified 'Indeed, difference between gas-exchange derived values and online measurements of $\delta^{13}C$ has been widely used to …' as 'Differences between gas-exchange derived values and online measurements of $\delta^{13}C$ have often been used to …'on Lines 78-79, Page 2 of revised manuscript.

L82; 'for the differences from' to 'for the differences in the'

Response: Considering your advice, we changed 'for the differences from' to 'for the differences in the' on Line 82, Page 3 of revised manuscript.

We changed 'whereas' to 'but' on Line 83, Page 3 of revised manuscript.

Changed 'of' to 'between', 'or/and' to 'and/or' on Lines 85-86, Page 3 of revised manuscript.

L87: change 'magnitude of these carbon fractionations are related to environmental variation have not yet been investigated.' to 'magnitude of carbon fractionation is related to environmental variation that has yet to be fully investigated.'

Response: We accept your helpful suggestion and changed as 'The degree to magnitude of carbon fractionation is related to environmental variation that has yet to be fully investigated' on Lines 86-87, Page 3 of revised manuscript.

We have simplified the sentence as 'The simultaneous isotopic analysis of leaves is a recent refinement in isotopic studies that allows determination of the temporal variation in isotopic fractionation (Rinne et al., 2016). This will aid the accurate recording of environmental conditions' on Lines 88-89, Page 3 of revised manuscript, and changed 'and are defined as…' to 'and these are termed …' on Line 90, Page 3 of revised manuscript.

L94-95: 'However, there is a dispute whether the fractionation stemmed…' to 'However, there is disagreement whether fractionation stemming…'

Response: Thanks for your helpful suggestions and we have rewritten as 'However, there is disagreement whether fractionation caused by post-carboxylation and/or mesophyll resistance can alter the stable signatures of leaf carbon and thence influence instantaneous water use efficiency

(iWUE)' on Lines 93-95, Page 3 of revised manuscript.

L97-99: awkward, please rewrite

Response: Based on your advice, we have rewritten this part as 'In addition, the manner in which iWUE derived from these isotopic fractionations responds to environmental factors, such as elevated [$CO_2$] and/or soil water gradients, is unknown' on Lines 93-95, Page 3 of revised manuscript.

We rewrote the description on materials as '…in sapling leaves of two tree species, *Platycladus orientalis* (L.) Franco and *Quercus variabilis* Bl., native to semi-arid areas of China' on Lines 98-100, Page 3 of revised manuscript. We have translated the long sentence to several short sentences as 'We also conduct gas exchange measurements in controlled environment growth chambers (FH-230, Taiwan Hipoint Corporation, Kaohsiung City, Taiwan)' and 'One goal is to differentiate the $^{13}$C fractionation from the site of carboxylation to cytoplasm prior to sugars transportation in *P. orientalis* and *Q. variabilis*, that is the total $^{13}$C fractionation, determined from the $\delta^{13}$C of water-soluble compounds and gas-exchange measurements. The other is to discuss the potential causes for the observed divergence, estimate contributions of post-photosynthesis and mesophyll conductance on these differences, and describe how carbon isotopic fractionations respond to the interactive effects of elevated [$CO_2$] and water stress' on Lines 100-107, Page 3 of revised manuscript.

L103: at the first mention of the growth chamber (use the full citation that you provide on L120-121)

Response: We accept your suggestion and have supplied the full citation as '(FH-230, Taiwan Hipoint Corporation, Kaohsiung City, Taiwan)' on Line 101, Page 3 of revised manuscript.

We have rewritten the introduction of study region and the process of transplantation for potted saplings as follows: '*P. orientalis* and *Q. variabilis* saplings, selected as experimental material, were obtained from the Capital Circle forest ecosystem station, a part of Chinese Forest Ecosystem Research Network (CFERN), 40º03'45''N, 116º5'45''E in Beijing, China. This region is forested by *P. orientalis* and *Q. variabilis*. We chose saplings with similar basal diameters, heights, and growth classes. Each sapling was placed into an individual pot (22 cm diam. $\times$ 22 cm high). Undisturbed soil samples were collected from the field, sieved (with particles >10 mm removed), and placed into the pots. The soil bulk density in the pots was maintained at 1.337–1.447 g•cm$^{-3}$. After a 30 d transplant recovery period, the saplings were placed into growth chambers for orthogonal cultivation' on Lines 110-117, Page 3 of revised manuscript.

L122-123: 'daytime temperature in chambers was set to 25 ± 0.5℃ from 07:00 122 to 17:00, and the night-time temperature was 18 ± 0.5℃ from 17:00 to 07:00' to 'daytime and nighttime temperatures in the chambers was set to 25 ± 0.5℃ from 07:00 to 17:00 and 18 ± 0.5℃ from 17:00 to 07:00'

Response: We accept your helpful suggestion and revised this part on Lines 120-121, Page 3 of revise manuscript.

Omit L 131 & 132.

Response: We accept your helpful suggestion and omitted the sentences.

L141-144: can this be simplified?

Response: According your consideration, we made a table that presents the orthogonal treatments on Lines 134-141, Page 4 and Page 25 of revised manuscript.

L148-154: can this also be simplified? Can you put this detail and the detail above in a table?

Response: According your consideration, we made a table that presents the orthogonal treatments involved in revised manuscript, as follows on Page 25 of revised manuscript:

**Table 1.** Orthogonal treatments of *P. orientalis* and *Q. variabilis* for four $CO_2$ concentrations $\times$ five soil volumetric water contents.

| *P. orientalis* | Repeats (cultivated period) | $B_1$ | $B_2$ | $B_3$ | $B_4$ | $B_5$ |
|---|---|---|---|---|---|---|
| $A_1$ | $R_1$:June 2–9 | $A_1B_1R_1$ | $A_1B_2R_1$ | $A_1B_3R_1$ | $A_1B_4R_1$ | $A_1B_5R_1$ |
| | $R_2$:June 12–19 | $A_1B_1R_2$ | $A_1B_2R_2$ | $A_1B_3R_2$ | $A_1B_4R_2$ | $A_1B_5R_2$ |
| $A_2$ | $R_1$:July 11–18 | $A_2B_1R_1$ | $A_2B_2R_1$ | $A_2B_3R_1$ | $A_2B_4R_1$ | $A_2B_5R_1$ |
| | $R_2$:July 22–29 | $A_2B_1R_2$ | $A_2B_2R_2$ | $A_2B_3R_2$ | $A_2B_4R_2$ | $A_2B_5R_2$ |
| $A_3$ | $R_1$:June 2–9 | $A_3B_1R_1$ | $A_3B_2R_1$ | $A_3B_3R_1$ | $A_3B_4R_1$ | $A_3B_5R_1$ |
| | $R_2$:June 12–19 | $A_3B_1R$ | $A_3B_2R_2$ | $A_3B_3R_2$ | $A_3B_4R_2$ | $A_3B_5R_2$ |
| $A_4$ | $R_1$:July 11–18 | $A_4B_1R_1$ | $A_4B_2R_1$ | $A_4B_3R_1$ | $A_4B_4R_1$ | $A_4B_5R_1$ |
| | $R_2$:July 22–29 | $A_4B_1R_2$ | $A_4B_2R_2$ | $A_4B_3R_2$ | $A_4B_4R_2$ | $A_4B_5R_2$ |
| *Q. variabilis* | Repeats (cultivated period) | $B_1$ | $B_2$ | $B_3$ | $B_4$ | $B_5$ |
| $A_1$ | $P_1$:June 21–28 | $A_1B_1P_1$ | $A_1B_2P_1$ | $A_1B_3P_1$ | $A_1B_4P_1$ | $A_1B_5R_1$ |
| | $P_2$:July 2–9 | $A_1B_1P_2$ | $A_1B_2P_2$ | $A_1B_3P_2$ | $A_1B_4P_2$ | $A_1B_5R_2$ |
| $A_2$ | $P_1$:August 4–11 | $A_2B_1P_1$ | $A_2B_2P_1$ | $A_2B_3P_1$ | $A_2B_4P_1$ | $A_2B_5R_1$ |
| | $P_2$:August 15–22 | $A_2B_1P_2$ | $A_2B_2P_2$ | $A_2B_3P_2$ | $A_2B_4P_2$ | $A_2B_5R_2$ |
| $A_3$ | $P_1$:June 21–28 | $A_3B_1P_1$ | $A_3B_2P_1$ | $A_3B_3P_1$ | $A_3B_4P_1$ | $A_3B_5R_1$ |
| | $P_2$:July 2–9 | $A_3B_1P_2$ | $A_3B_2P_2$ | $A_3B_3P_2$ | $A_3B_4P_2$ | $A_3B_5R_2$ |
| $A_4$ | $P_1$:August 4–11 | $A_4B_1P_1$ | $A_4B_2P_1$ | $A_4B_3P_1$ | $A_4B_4P_1$ | $A_4B_5R_1$ |
| | $P_2$:August 15–22 | $A_4B_1P_2$ | $A_4B_2P_2$ | $A_4B_3P_2$ | $A_4B_4P_2$ | $A_4B_5R_2$ |

L165-166: this needs revising

Response: Based on your suggestion, we have rewritten this part as follows 'Eight recently-expanded sun leaves were selected per sapling and homogenized in liquid nitrogen after gas-exchange measurements were finished' on Lines 152-153, Page 4 of revise manuscript.

L179: second Rsample needs to be change to Rstandard

Response: Thanks for your helpful advice, we have corrected the mistake in presentation as '$R_{standard}$' on Line 165, Page 4 of revised manuscript.

We omitted the redundancy sentence on Line 246, Page 7 in revised manuscript.

Throughout the manuscript: usage of CO2 concentration, sometimes you use [CO2] and other times you spell it out; try to be consistent; since you introduced [CO2] why not continue to use it? The labels on some of the Figures are simply too small; please fix

Response: Thanks your helpful suggestion. We have checked the usage of $CO_2$ concentration and changed it to '[$CO_2$]' through the whole paper. We have used enlarged the font of labels in Figures 2-8.
.

---

## Editor Decision (ED2)

[revised manuscript text omitted]
 (‰) Mesophyll conductance | | | | $^{13}C$ fractionation (‰) Post-photosynthesis | | | |
|---|---|---|---|---|---|---|---|---|---|---|---|---|---|
| | | **CO₂ concentration (ppm)** | | | | | | | | | | | |
| | | 400 | 500 | 600 | 800 | 400 | 500 | 600 | 800 | 400 | 500 | 600 | 800 |
| *P. orientalis* | 35%–45% | 0.0328 | 0.0373 | 0.0349 | 0.0332 | 0.0081 | 0.0030 | 0.0034 | 0.0072 | 0.0247 | 0.0343 | 0.0315 | 0.0260 |
| | 50%–60% | 0.0367 | 0.0437 | 0.0382 | 0.0374 | 0.0018 | 0.0058 | 0.0094 | 0.0004 | 0.0349 | 0.0379 | 0.0288 | 0.0370 |
| | 60%–70% | 0.0405 | 0.0366 | 0.0421 | 0.0409 | 0.0018 | 0.0050 | 0.0026 | 0.0007 | 0.0387 | 0.0316 | 0.0395 | 0.0402 |
| | 70%–80% | 0.0444 | 0.0453 | 0.0413 | 0.0452 | 0.0044 | 0.0052 | 0.0103 | 0.0013 | 0.0400 | 0.0401 | 0.0310 | 0.0439 |
| | 100% | 0.0441 | 0.0453 | 0.0456 | 0.0472 | 0.0057 | 0.0040 | 0.0025 | 0.0039 | 0.0384 | 0.0413 | 0.0431 | 0.0433 |
| *Q. variabilis* | 35%–45% | 0.0388 | 0.0402 | 0.0406 | 0.0384 | 0.0007 | 0.0025 | 0.0006 | 0.0091 | 0.0381 | 0.0377 | 0.0400 | 0.0293 |
| | 50%–60% | 0.0433 | 0.0448 | 0.0409 | 0.0368 | 0.0061 | 0.0084 | 0.0023 | 0.0018 | 0.0372 | 0.0364 | 0.0386 | 0.0350 |
| | 60%–70% | 0.0424 | 0.0440 | 0.0445 | 0.0414 | 0.0066 | 0.0086 | 0.0078 | 0.0041 | 0.0358 | 0.0354 | 0.0367 | 0.0373 |
| | 70%–80% | 0.0424 | 0.0446 | 0.0482 | 0.0457 | 0.0034 | 0.0016 | 0.0074 | 0.0028 | 0.0390 | 0.0430 | 0.0408 | 0.0429 |
| | 100% | 0.0441 | 0.0466 | 0.0466 | 0.0398 | 0.0027 | 0.0076 | 0.0022 | 0.0125 | 0.0414 | 0.0390 | 0.0444 | 0.0273 |

---

## Author Response (AR3)

Response list to the Editor's comments
MS No.: bg-2016-372
MS Type: Research article
Ref: doi:10.5194/bg-2016-372
Title: **Interaction of $CO_2$ concentrations and water stress in semi-arid plants causes diverging response in instantaneous water use efficiency and carbon isotope composition**
Authors: Na Zhao, Ping Meng, Yabing He, Xinxiao Yu*

Dear Editor,

Thanks for your patiently help and constructive comments that provide scientific guidance for our writing and future research. We have been carefully considering your suggestions and revising the manuscript in the revised manuscript (marked in red color) accordingly. In addition to the following issues, we have corrected other mistakes with grammar and expression in the revised manuscript (marked in red color). The following below in blue are our point-to-point responses for your questions and comments. We are appreciated for your kind help on writing.

We are looking forward to your further comments and a possible publication in the BG special issue (Ecosystem processes and functioning across current and future dryness gradients in arid and semi-arid lands).

Kind regards,

Xinxiao Yu

■■■■■■■■■■■■■■■■■■■■■■■■■■■■■■■■■■■■■■■■■■■■■■■■■■■■■■■■■

Comments to the Author

Line 12 "occurred" to "occurs"

Response: Thank you for the careful review and constructive comments. We apologize for any inconvenience that we bring you for my carelessness in writing. Based on your helpful suggestions, we have changed "occurred" to "occurs" on Line 12, Page 1 of the revised manuscript.

Line 22 "leaf-exported" to "leaf-level"

Response: We appreciate your helpful comments and have changed "leaf-exported" to "leaf-level" on Line 22, Page 1 of the revised manuscript.

Line 33 "fractionations" to "fractionation"

Response: Considering your suggestions, we have corrected this plural noun "fractionations" into "fractionation" (on Page 1, Line 33 of revised manuscript).

Line 35 "measurement" to "measurements"

Response: In this study, there were several times of gas-exchange measurements. So we accepted your advice and changed the "measurement" to "measurements".

Lines 59-60 "…and they will be recorded …" to "…, recording…"

Response: According your helpful comments, we have simplified this sentence into a participles as attributive clause "…, recording…" on Lines 59-60, Page 2 of revised manuscript.

Line 69 remove the comma followed "(Gessler et al., 2008; Gessler et al., 2014)"

Response: Thank you for pointing the mistake and we have removed the comma on Line 69, Page 2 of revised manuscript:

Line 72 "during exportation" to "during export"

Response: We have changed "during exportation" to "during export" on Line 72, Page 2 of the revised manuscript:

Line 77 remove the article "the" before "Farquhar's model"

Response: Thanks for your helpful suggestion and we have removed the redundant "the" before "Farquhar's model" on Line 77, Page 2 of revised manuscript.

Lines 86-87 "The degree to magnitude of carbon fractionations is related to environmental variation that has yet to be fully investigated." to ""The degree to which carbon fractionation is related to environmental variation has yet to be fully investigated."

Response: According your helpful comments, we have revised this part as "The degree to which carbon fractionation is related to environmental variation has yet to be fully investigated." on Lines 86-87, Page 3 of revised manuscript.

Line 89 "aid the accurate recording" to "aid in the accurate recording"

Response: Thanks for your suggestion and we have changed this part as "aid in the accurate recording" on Line 89, Page 3 of revised manuscript.

Line 96 "from these isotopic fractionations" to "from isotopic fractionation"

Response: We agree with your suggestion and have revised this sentence on Page 3, Line 96 in the revised manuscript.

Line 102 "sugars" to "sugar"

Response: According your helpful comments, we have revised this part on Page 3, Line 102 in the revised manuscript.

Line 102 "The other one" to "Another goal"

Response: Based on your suggestion, we have changed the 'The other one' to 'Another goal' on Line 104, Page 3 of revised manuscript.

Line 106 'fractionations respond' to 'fractionation responds'

Response: We accept your helpful suggestion and changed as 'fractionation responds' on Line 106, Page 3 of revised manuscript.

Line 113 'We chose saplings with similar basal diameters, heights, and growth classes' to 'We chose saplings of similar basal diameters, heights, and growth class'

Response: Thanks for your helpful advices. We rewrote this sentence as 'We chose saplings of similar basal diameters, heights, and growth class' on Line 113, Page 3 of revised manuscript.

Line 116 '30 d' to '30-day'

Response: Based on your suggestion, we have changed the '30 d' to '30-day' on Line 116, Page 3 of revised manuscript.

Line 118 'controlled experiment studies were' to 'controlled experiment was'

Response: According your advice, we corrected this part as 'controlled experiment was' on Line 118, Page 3 of revised manuscript.

Line 119 'meteorological factors of different growth seasons' to 'meteorological conditions of different growing seasons'

Response: Thanks for your helpful suggestion and we have rewritten this part as 'meteorological conditions of different growing seasons' on Lines 119-120, Page 3 of revised manuscript.

Line 123 'The chamber control system can control and monitor $[CO_2]$' to 'The chamber system can both control and monitor $[CO_2]$'

Response: Based on your suggestion, we have changed 'The chamber control system can control and monitor $[CO_2]$' to 'The chamber system can both control and monitor $[CO_2]$' on Line 123, Page 3 of revised manuscript.

Lines 125-126 'The target $[CO_2]$ in the chambers' to 'The target $[CO_2]$ in each chamber'

Response: According your advice, we have represented this part as 'The target $[CO_2]$ in each chamber' on Line 126, Page 4 of revised manuscript.

Line 127 'and it can avoid' to 'to avoid'

Response: Based on your suggestion, we have simplified this sentence as 'to avoid' on Line 127, Page 4 of revised manuscript.

Line 129 'components' to 'component'

Response: I am very sorry for my careless in writing grammar and have corrected the mistake on Line 129, Page 4 of revised manuscript.

Line 132 'Since timely SWC' to 'Since changes in SWC'

Response: Thanks for your helpful advices. We rewrote this sentence as 'Since changes in SWC' on Line 132, Page 4 of revised manuscript.

Line 135 remove 'combining $[CO_2]$ gradient'

Response: Thanks for your helpful advices. We have removed 'combining $[CO_2]$ gradient' on Line 134, Page 4 of revised manuscript.

Line 135 'we established the orthogonal treatments for' to 'we established orthogonal treatments of'

Response: We accept your helpful suggestion and changed as 'we established orthogonal treatments of' on Line 135, Page 4 of revised manuscript.

Line 137 'in the chambers.' to 'in the chambers;'

Response: We accept your helpful suggestion and changed as 'in the chambers;' on Line 136, Page 4 of revised manuscript.

Line 140 '7 d' to '7 days'

Response: Based on your suggestion, we have changed '7 d' to '7 days' on Line 140, Page 4 of revised manuscript.

Line 140 'in one $[CO_2] \times SWC$ treatment' to ''in each of the $[CO_2] \times SWC$ treatments'

Response: According your advice, we changed 'in one $[CO_2] \times SWC$ treatment' into ''in each of the $[CO_2] \times SWC$ treatments' on Line 140, Page 4 of revised manuscript.

Lines 140-141 'Pots in chambers were arranged to promote uniform illumination every two days' to 'Pots in the chambers were arranged every two days to promote uniform illumination'

Response: Thanks for your helpful advices. We have changed this part according your suggestion on Lines 140-141, Page 4 of revised manuscript.

Line 145 'per specie' to 'per species'

Response: We accept your helpful suggestion.

Line 147 'Based on the theories proposed by…' to 'Based on theoretical consideration of…'

Response: According your advice, we rewrote this part on Lines 147-148, Page 4 of revised manuscript.

Line 155 'Each leaf' to 'Each leaf sample'

Response: We accept your helpful suggestion on Line 155, Page 4 of revised manuscript.

Line 156 'per specie' to 'per species'

Response: We accept your helpful suggestion.

Line 157 'The tubes containing the above mixture' to 'The tubes containing the mixture'

Response: We accept your helpful suggestion on Line 157, Page 4 of revised manuscript.

Line 161 'mass spectrometer' to 'massspectrometer'

Response: We accept your helpful suggestion on Line 161, Page 4 of revised manuscript.

Line 162 'are' to 'were'

Response: We accept your helpful suggestion and revised this word on Line 162, Page 4 of revised manuscript.

Line 163 'Pee Dee Belemnite (PDB)' to 'Pee Dee Belemnite (PDB) standard'

Response: We accept your helpful suggestion and supplemented 'standard' on Line 163, Page 4 of revised manuscript.

Line 169 'sugars' to 'sugar'

Response: We accept your helpful suggestion and applied 'sugar' on Line 169, Page 5 of revised manuscript.

Line 170 '…developed by…' to '…of…'

Response: We accept your helpful suggestion and changed it as '…of…'on Line 170, Page 5 of revised manuscript.

Line 171 'is' to 'was'

Response: We accept your helpful suggestion.

Line 174 'is' to 'was'

Response: We accept your helpful suggestion.

Line 176 remove 'the' before 'growth chambers'

Response: Thanks for your helpful advices. We have removed the redundant 'the' on Line 176, Page 5 of revised manuscript.

Line 179 'gas-exchange measurements ($WUE_{ge}$) is calculated as' to 'gas-exchange measurement ($WUE_{ge}$) was calculated as'

Response: Based on your suggestion, we have changed this part on Line 179, Page 5 of revised manuscript.

Line 182 'between $e_{lf}$ and $e_{atm}$ that represent…' to 'between $e_{lf}$ and $e_{atm}$, representing…'

Response: Thanks for your helpful advices. We have changed 'between $e_{lf}$ and $e_{atm}$ that represent…' to 'between $e_{lf}$ and $e_{atm}$, representing…' on Line 182, Page 5 of revised manuscript.

Line 187 remove '$WUE_{cp}$'

Response: Based on your suggestion, we have removed '$WUE_{cp}$' on Line 187, Page 5 of revised manuscript.

Line 191 'can be' to 'was'

Response: We accept your helpful suggestion and have changed 'can be' to 'was' on Line 191, Page 5 of revised manuscript.

Line 193 'is' to 'was'

Response: We accept your helpful suggestion.

Line 193 'Eqn. (2). The $\Delta_{model}$ can be determined by Eqns. (3 and 4) as' to 'Eqn. (2); $\Delta_{model}$ was determined by combining Eqns. (3 and 4) as'

Response: According your advice, we have integrated 'Eqn. (2). The $\Delta_{model}$ can be determined by Eqns. (3 and 4)' to 'Eqn. (2); $\Delta_{model}$ was determined by combining Eqns. (3 and 4) as' on Line 193, Page 5 of revised manuscript.

Note that you use both 'Eqn' and 'Equation'. Pick one be consistent in its use.

Response: Based on your suggestion, we have uniformed the expression of equation as 'Eqn' throughout the revised manuscript.

Line 199 'is' to 'was'

Response: According your advice, we have changed 'is' to 'was' on Line 199, Page 5 of revised manuscript.

Lines 200-201 'to that in the ambient environment surrounding plants ($C_a$)' to 'to the concentration in the outside air ($C_a$)'

Response: According your advice, we have rewritten this part as 'to the concentration in the outside air ($C_a$)' on Lines 200-201, Pages 5-6 of revised manuscript.

Line 204 'ambient environment' to 'ambient air'

Response: We accept your helpful suggestion and changed 'ambient environment' to 'ambient air' on Line 203, Page 6 of revised manuscript.

Line 211 'occurring' to 'across the'

Response: Based on your suggestion, it was presented more clearly in expression about Eqn. (11) and we have changed 'occurring' to 'across the' on Lines 210-211, Page 6 of revised manuscript.

Line 212 'can be shown as' to 'can be written as'

Response: We accept your helpful suggestion and rewrote it as 'can be written as' on Line 211, Page 6 of revised manuscript.

Line 216 'the value of $b$ would influence' to '$b$ influences'

Response: According your advice, we have changed 'the value of $b$ would influence' to '$b$ influences' on Line 215, Page 6 of revised manuscript.

Line 219 'diffusions' to 'diffusion'

Response: We accept your helpful suggestion and rewrote it as 'diffusion' on Line 217, Page 6 of revised manuscript.

Line 220 'could be simplified as' to 'may be simplified to'

Response: We accept your helpful suggestion and rewrote it as on Line 218, Page 6 of revised manuscript.

Line 222 'Equation 12' to 'Eqn. (12)'

Response: We accept your helpful suggestion.

Line 228 remove 'the' before '$P_n$'

Response: We accept your helpful suggestion and removed redundant 'the' on Line 226, Page 6 of revised manuscript.

Line 233 'In calculation' to 'In the calculation'

Response: We accept your helpful suggestion and supplemented 'the' before 'calculation' on Line 231, Page 6 of revised manuscript.

Line 235 'can be transformed into' to 'can be rewritten as'

Response: We accept your helpful suggestion and changed 'can be transformed into' to 'can be rewritten as' on Line 233, Page 6 of revised manuscript.

Line 245 '($p<0.5$)' to '($p<0.05$)'?

Response: I am very sorry for my careless in spelling and have corrected the mistake throughout the revised manuscript.

Lines 249-250 'Nevertheless, $g_s$ of *Q. variabilis* in $C_{400}$, $C_{500,}$ and $C_{600}$ was significantly higher than in $C_{800}$ at 50%−80% of FC ($p< 0.01$).' to 'Nevertheless, $g_s$ in *Q. variabilis* for $C_{400}$, $C_{500,}$ and $C_{600}$ was significantly higher than for $C_{800}$ at 50%−80% of FC ($p< 0.01$).'

Response: According your advice, we have changed the prepositions in this sentence on Lines 247-248, Page 7 of revised manuscript.

Lines 250-251 'the two species in $C_{400}$ and $C_{500}$ was significantly higher than in $C_{600}$ and $C_{800}$ except at …' to 'the two species for $C_{400}$ and $C_{500}$ was significantly higher than for $C_{600}$ and $C_{800}$, except at …'

Response: According your advice, we have changed the prepositions in this sentence on Lines 248-249, Page 7 of revised manuscript.

Lines 251-252 '$P_n$, $g_s$, $C_i$ and $T_r$ of *Q. variabilis* was significantly greater than the corresponding values of *P. orientalis* ($p < 0.01$, Fig. 2)' to '$P_n$, $g_s$, $C_i$ and $T_r$ in *Q. variabilis* was significantly greater than the corresponding values in *P. orientalis* ($p < 0.01$, Fig. 2)'

Response: According your advice, we have changed the prepositions in this sentence on Lines 249-250, Page 7 of revised manuscript.

Line 259 'the two species reached maxima' to 'the two species, reaching their respective maxima'

Response: Based on your suggestion, we have changed this part as 'the two species, reaching their respective maxima' on Line 257, Page 7 of revised manuscript.

Lines 261-262 'the $\delta^{13}C_{WSC}$ of *P. orientalis* was significantly larger than that of *Q. variabilis* at any [$CO_2$] $\times$ SWC treatment ($p < 0.01$, Fig. 3)' to 'the $\delta^{13}C_{WSC}$ in *P. orientalis* was significantly larger than that in *Q. variabilis* at any [$CO_2$] $\times$ SWC treatment ($p < 0.01$, Fig. 3)'

Response: According your advice, we have changed the prepositions in this sentence on Lines 259-260, Page 7 of revised manuscript.

Lines 267-268 'Differing from variation in $WUE_{ge}$ of *P. orientalis* with moistened soil, $WUE_{ge}$ in *Q. variabilis* increased slightly at 100% of FC in $C_{600}$ or $C_{800}$ (Fig. 4b)' to 'Differing from variation in $WUE_{ge}$ in *P. orientalis* with moistened soil, $WUE_{ge}$ in *Q. variabilis* increased slightly at 100% of FC for $C_{600}$ or $C_{800}$ (Fig. 4b)'

Response: We have changed the prepositions in this sentence on Lines 265-266, Page 7 of revised manuscript.

Lines 268-269 'The maximum $WUE_{ge}$ occurred at 35%–45% of FC in $C_{800}$ among all orthogonal treatments for *P. orientalis* and this was also observed in *Q. variabilis*.' to 'The maximum $WUE_{ge}$ occurred at 35%–45% of FC for $C_{800}$ among all orthogonal treatments associated with both species.'

Response: Thanks for your helpful advices. We rewrote this sentence as 'The maximum $WUE_{ge}$ occurred at 35%–45% of FC for $C_{800}$ among all orthogonal treatments associated with both species.' on Lines 266-267, Page 7 of revised manuscript.

Lines 269-270 'Elevated [$CO_2$] enhanced the $WUE_{ge}$ of *Q. variabilis* at any SWC except at 60%–80% of FC.' to 'Elevated [$CO_2$] enhanced the $WUE_{ge}$ in *Q. variabilis* at any SWC, except at 60%–80% of FC.'

Response: Based on your suggestion, we have changed the prepositions in this sentence and made the expression more clearly on Lines 267-268, Page 7 of revised manuscript.

Lines 270-271 'Thirty-two saplings of *P. orientalis* had greater $WUE_{ge}$ than did *Q. variabilis* in the same [$CO_2$] $\times$ SWC treatments ($p < 0.5$).' to 'Thirty-two saplings of *P. orientalis* had greater $WUE_{ge}$ than did *Q. variabilis* for the same [$CO_2$] $\times$ SWC treatment ($p < 0.05$).'

Response: Based on your helpful suggestions, we changed 'in' to 'for' in the revised sentence and confirmed the level of significance was $p < 0.05$ throughout the revised article on Lines 268-269, Page 7 of revised manuscript.

Line 272 '…$WUE_{cp}$ of *P. orientalis* in $C_{600}$ or $C_{800}$…' to '…$WUE_{cp}$ in *P. orientalis* for $C_{600}$ or $C_{800}$…'

Response: Considering your helpful comments, we have changed the prepositions on Line 270, Page 7 of revised manuscript.

Line 273 'as well as that in $C_{400}$ or $C_{500}$ while SWC exceeded' to 'as well as that for $C_{400}$ or $C_{500}$,

while SWC exceeded'

Response: Thanks for your helpful advices. We changed 'in' to 'for' and divide the sentence into two parts on Line 271, Page 7 of revised manuscript.

Line 274 'SWC increased' to 'increasing SWC'

Response: According your advice, we changed 'SWC increased' to 'increasing SWC' on Line 272, Page 7 of revised manuscript.

Lines 274-275 'of' to 'in' and 'in' to 'for'

Response: We accept your helpful suggestion and changed 'of' to 'in' on Lines 272-273, Pages 7-8 of revised manuscript.

Line 276 'of' to 'for', 'of' to 'in', and 'than *P. orientalis*' to 'than in *P. orientalis*'

Response: We accept your helpful suggestion and changed 'of' to 'in' on Line 274, Page 8 of revised manuscript.

Line 278 'sugars' to 'sugar'

Response: We accept your helpful suggestion and changed 'sugars' to 'sugar' on Line 276, Page 8 of revised manuscript.

Line 280 'can track' to 'can help track'

Response: We accept your helpful suggestion and changed 'can track' to 'can help track' on Line 278, Page 8 of revised manuscript.

Line 282 'of' to 'in'

Response: Thanks for your helpful advices and changed 'of' to 'in' on Line 280, Page 8 of revised manuscript.

Line 283 'total fractionations' to 'total fractionation', 'of' to 'in', and 'were' to 'was'

Response: Thanks for your helpful advices and we changed 'total fractionations' to 'total fractionation', 'of' to 'in', and 'were' to 'was' on Line 281, Page 8 of revised manuscript.

Lines 283-284 'SWC increased' to 'increasing SWC'

Response: According your advice, we changed 'SWC increased' to 'increasing SWC' on Line 282, Page 8 of revised manuscript.

Line 284 'values that' to 'when SWC'

Response: Thanks for your helpful advices and we changed 'values that' to 'when SWC' on Line 282, Page 8 of revised manuscript.

Line 285 'total fractionations' to 'total fractionation'

Response: We accept your helpful suggestion and 'total fractionations' to 'total fractionation' on Line 283, Pages 8 of revised manuscript.

Lines 287-288 'of' to 'in'

Response: We accept your helpful suggestion and changed 'of' to 'in' on Lines 285-286, Page 8 of revised manuscript.

Line 289 'than did *Q. variabilis*' to 'than it did in *Q. variabilis*'

Response: We accept your helpful suggestion and changed 'than did *Q. variabilis*' to 'than it did in *Q. variabilis*' on Line 287, Page 8 of revised manuscript.

Line 292 'trend of $g_m$ occurred' to 'trend occurred in $g_m$'

Response: Thanks for your helpful advices and we changed 'trend of $g_m$' to 'trend occurred in $g_m$' on Line 290, Page 8 of revised manuscript.

Line 293 'reduced' to 'decreasing'

Response: We accept your helpful suggestion and changed 'reduced' to 'decreasing' on Line 291,

Page 8 of revised manuscript.

Lines 293-294 '($p<$ 0.5)' to '($p<$ 0.05)'

Response: According your advice, we have confirmed the level of significance for orthogonal treatments was '($p<$ 0.05)' throughout the revised article.

Line 294 'of' to 'in'

Response: We accept your helpful suggestion and changed 'of' to 'in' on Line 292, Page 8 of revised manuscript.

Line 295 'significant except those' to 'significant, except those'

Response: Considering your helpful advice, we have divided this sentence into two parts on Line 293, Page 8 of revised manuscript.

Line 295 'of' to 'in'

Response: We accept your helpful suggestion and changed 'of' to 'in' on Line 293, Page 8 of revised manuscript.

Lines 297-298 '($p<$ 0.5)' to '($p<$ 0.05)'

Response: According your advice, we have confirmed the level of significance for orthogonal treatments was '($p<$ 0.05)' throughout the revised article.

Lines 298-299 'increments of' to 'increment in'

Response: We accept your helpful suggestion and changed 'increments of' to 'increment in' on Line 296, Page 8 of revised manuscript.

Lines 299-300 'of' to 'in'

Response: We accept your helpful suggestion and changed 'of' to 'in' on Lines 297-298, Page 8 of revised manuscript.

Line 305 'less contribution' to 'a smaller contribution'

Response: Based on your suggestion, we have changed 'less contribution' to 'a smaller contribution' on Line 303, Page 8 of revised manuscript.

Line 305 'that from post-carboxylation fractionation within any treatment' to 'did post-carboxylation fractionation irrespective of treatment'

Response: Based on your suggestion, we have changed 'that from post-carboxylation fractionation within any treatment' to 'did post-carboxylation fractionation irrespective of treatment' on Lines 303-304, Page 8 of revised manuscript.

Line 307 'SWC increased' to 'increasing SWC'

Response: According your advice, we changed 'SWC increased' to 'increasing SWC' on Line 305, Page 8 of revised manuscript.

Line 308 'in *P. orientalis*, yet, in *Q. variabilis*,' to 'in *P. orientalis*; yet, in *Q. variabilis*,'

Response: According your advice, we changed 'in *P. orientalis*, yet, in *Q. variabilis*,' to 'in *P. orientalis*; yet, in *Q. variabilis*,' on Line 306, Page 8 of revised manuscript.

Line 308 'in the two species, post-carboxylation fractionations in leaves' to 'in the two species post-carboxylation fractionation in leaves'

Response: Thanks for your helpful advices. We remove ',' and changed the 'fractionations' to 'fractionation' on Line 307, Page 8 of revised manuscript.

Line 310 'and these contributions all increased as soil moisture increased' to 'all increased as SWC increased'

Response: Based on your suggestion, we removed the redundant expression 'and these contributions' and uniformed 'soil moisture' as 'SWC' on Lines 307-308, Page 8 of revised

manuscript.

Line 312 'fractionations' to 'fractionation'

Response: We accept your helpful suggestion and changed 'fractionations' to 'fractionation' on Line 309, Page 8 of revised manuscript.

Line 312 'increase' to 'increases'

Response: We accept your helpful suggestion.

Line 312 'reached maxima' to 'reached a maximum'

Response: We accept your helpful suggestion and changed 'fractionations' to 'fractionation' on Line 310, Page 8 of revised manuscript.

Line 313 'were reduced' to 'declined'

Response: We accept your helpful suggestion and changed 'were reduced' to 'declined' on Line 310, Page 8 of revised manuscript.

Line 315 'from' to 'associated with'

Response: According your advice, we changed 'from' to 'associated with' on Line 312, Page 8 of revised manuscript.

Line 316 'values for' to 'in'

Response: According your advice, we changed 'values for' to 'in' on Line 313, Page 8 of revised manuscript.

Line 317 remove ', respectively'

Response: We accept your helpful suggestion.

Line 317 remove 'the' before '$g_s$'

Response: We accept your helpful suggestion.

Line 319 'occurring after' to 'following'

Response: According your advice, we changed 'occurring after' to 'following' on Line 316, Page 9 of revised manuscript.

Lines 325-326 'caused a greater $g_s$ reduction' to 'caused a reduction in greate $g_s$'

Response: Based on your suggestion, we have changed 'caused a greater $g_s$ reduction' to 'caused a reduction in greater $g_s$' on Lines 321-322, Page 9 of revised manuscript.

Line 326 'as was similarly reported' to 'as is previously reported'

Response: Based on your suggestion, we have changed 'as was similarly reported' to 'as is previously reported' on Line 322, Page 9 of revised manuscript.

Lines 326-327, Line 331 'of' to 'in'

Response: Thanks for your helpful advices. We have changed the prepositions 'of' to 'in' on Lines 322-323, Page 9 of revised manuscript.

Line 332 'was increased considerably while SWC exceeded…' to 'increased considerably, while SWC exceeded…'

Response: Based on your helpful comments, we have changed 'was increased' to 'increased' and divided this sentence into two parts to express better on Lines 322-323 Page 9 of revised manuscript.

Lines 335-336 'in potted plant experiments' to 'of potted plants'

Response: Based on your helpful comments, we changed 'in potted plant experiments' to 'of potted plants' on Lines 331-332 Page 9 of revised manuscript.

Line 338 'threshold (70%-80% of FC)' to 'threshold of 70%-80% of FC'

Response: Thanks for your helpful advices. We have changed 'threshold (70%-80% of FC)' to

'threshold of 70%-80% of FC' on Line 334, Page 9 of revised manuscript.

Line 342 '…of SWC below which' to '…in SWC, below which'

Response: Thanks for your helpful advices. We have changed 'of' to 'in' and divided this sentence into two parts by ',' on Line 338, Page 9 of revised manuscript.

Line 346 'severe drought and heavy irrigation,' to 'severe drought and the need for heavy irrigation,'

Response: Considering your suggestions, we have changed 'severe drought and heavy irrigation,' to 'severe drought and the need for heavy irrigation,' on Line 342, Page 9 of revised manuscript.

Line 347 'a lack, or excess, of water' to 'a lack or excess of water'

Response: We accept your helpful suggestion and removed the redundant ',' on Line 343, Page 9 of revised manuscript.

Line 352 and 356 change 'of' to 'in'

Response: We accept your helpful suggestion.

Line 352 'Comparing the $P_n$ and $T_r$ values of the two species,' to 'Comparing $P_n$ and $T_r$ in the two species,'

Response: We accept your helpful suggestion and removed the redundant 'the' and changed 'of' to 'in' on Line 350, Page 9 of revised manuscript.

Line 362 'that $WUE_{cp}$ was more consistent with daily mean $WUE_{ge}$ than $WUE_{phloem}$' to 'that $WUE_{cp}$ was more consistent with daily mean $WUE_{ge}$ than with $WUE_{phloem}$'

Response: Thanks for your suggestion and we added the 'with' before '$WUE_{phloem}$' on Line 359, Page 10 of revised manuscript.

Line 364 'variation to those $\delta^{13}C_{WSC}$,' to 'variations to those in $\delta^{13}C_{WSC}$,'

Response: Thanks for your suggestion and we changed 'variation to those $\delta^{13}C_{WSC}$,' to 'variations to those in $\delta^{13}C_{WSC}$,' on Line 360, Page 10 of revised manuscript.

Line 364 'that' to 'those'

Response: Thanks for your suggestion and we changed 'that' to 'those' on Line 360, Page 10 of revised manuscript.

Line 365 '1-2 d' to '1-2 days'

Response: We accept your helpful suggestion and changed '1-2 d' to '1-2 days' on Line 362, Page 10 of revised manuscript.

Line 367 'physiology' to 'physiological'

Response: We accept your helpful suggestion and changed 'physiology' to 'physiological' on Line 363, Page 10 of revised manuscript.

Line 367 'of' to 'in'

Response: We accept your helpful suggestion.

Line 368 not sure of the relevance of this statement 'In addition, species-specific $\delta^{13}C_{WSC}$ were observed in the same environmental treatment.'

Response: This statement presented there was the difference in $\delta^{13}C_{WSC}$ between *P. orientalis* and *Q. variabilis* for each orthogonal treatment. Based on your suggestions, we removed this sentence from this paragraph on Line 364, Page 10 of revised manuscript.

Line 375 '($g_m$) (Flexas et al., 2008)' to '($g_m$; Flexas et al., 2008)'

Response: We accept your helpful suggestion and changed as '($g_m$; Flexas et al., 2008)' on Line 370, Page 10 of revised manuscript.

Line 377 '7 d' to '7-day'

Response: We accept your helpful suggestion and changed as '7-day' on Line 372, Page 10 of revised manuscript.

Line 379 'on' to 'in'

Response: We accept your helpful suggestion and changed 'on' to 'in' on Line 374, Page 10 of revised manuscript.

Line 382 'of' to 'in'

Response: Thanks for your suggestion and we changed 'of' to 'in' on Line 377, Page 10 of revised manuscript.

Line 383 'was significantly decreased' to 'significantly decreased'

Response: Thanks for your suggestion and we changed 'was significantly decreased' to 'significantly decreased' on Line 378, Page 10 of revised manuscript.

Line 383 '60%-80% of FC and these' to '60%-80% of FC; these'

Response: Thanks for your suggestion and we changed '60%-80% of FC and these' to '60%-80% of FC; these' on Line 378, Page 10 of revised manuscript.

Line 384 'of' to 'in'

Response: We accept your helpful suggestion and changed 'of' to 'in' on Line 379, Page 10 of revised manuscript.

Line 385 'comparing' to 'compared'

Response: We accept your helpful suggestion and changed 'comparing' to 'compared' on Line 380, Page 10 of revised manuscript.

Line 386 'followed carboxylation while…' to 'followed carboxylation, while…'

Response: We accept your helpful suggestion.

Line 388 '…may be simply considered, whereas the fractionation induced by mesophyll conductance from sub-stomatic cavities…' is something missing?

Response: I am very sorry for my careless in writing and have supplemented related contents as 'The $^{13}$C fractionation of $CO_2$ from the air surrounding the leaf to sub-stomatal cavities may be simply explained by stomatal resistance, which also contains the fractionation derived from mesophyll conductance between sub-stomatal cavities and the site of carboxylation in the chloroplast that cannot be neglected and should be lucubrated (Pons et al., 2009; Cano et al., 2014).' on Lines 382-384, Page 10 of revised manuscript.

Line 393 '$p= 0.01$ or $p< 0.01$)' is this needed?

Response: According your advice, we removed the redundant 'or $p< 0.01$' and corrected this as '$p= 0.01$)' on Line 388, Page 10 of revised manuscript.

Line 393 'with that of $g_m$ with' to 'with those in $g_m$ with'

Response: Based on your suggestion, we have changed 'with that of $g_m$ with' to 'with those in $g_m$ with' on Line 389, Page 10 of revised manuscript.

Line 394 'on Table 2' to '(Table 2)'

Response: Based on your suggestion, we have changed 'on Table 2' to '(Table 2)' on Line 389, Page 10 of revised manuscript.

Line 403 remove 'processes'

Response: According your advice, we removed 'processes' on Line 398, Page 10 of revised manuscript.

Line 403 'twig fall within…' to 'twigs falls within…'

Response: Thanks for your helpful advices. We changed 'twig fall within…' to 'twigs falls within…'

on Line 398, Page 10 of revised manuscript.

Line 408 'sugars' to 'sugar'

Response: We accept your helpful suggestion and changed 'sugars' to 'sugar' on Line 403, Page 11 of revised manuscript.

Line 409 'of' to 'in'

Response: We accept your helpful suggestion and changed the preposition 'of' to 'in' on Line 404, Page 11 of revised manuscript.

Lines 413-415 'of' to 'in'

Response: We accept your helpful suggestion and changed the preposition 'of' to 'in' on Lines 406, 408-410, Page 11 of revised manuscript.

Line 415 'at' to 'for'

Response: Based on your suggestion, we have changed the 'at' to 'for' on Line 410, Page 11 of revised manuscript.

Line 417 remove 'the' before 'stomata aperture'

Response: According your advice, we removed 'the' before 'stomata aperture' on Line 412, Page 11 of revised manuscript.

Line 422 'leaf-exported' to 'leaf-level'

Response: Thanks for your helpful advices. We have changed 'leaf-exported' to 'leaf-level' on Line 417, Page 11 of revised manuscript.

Line 423 'ambient environment' to 'ambient air'

Response: Thanks for your helpful advices. We have changed 'ambient environment' to 'ambient air' on Line 418, Page 11 of revised manuscript.

Line 425 'sugars' to 'sugar'

Response: According your advice, we have changed 'sugars' to 'sugar' on Line 420, Page 11 of revised manuscript.

Line 426 'of two tree species' to 'in the two tree species'

Response: According your advice, we have changed 'of two tree species' to 'in the two tree species' on Line 421, Page 11 of revised manuscript.

Line 429 'of' to 'in'

Response: We accept your helpful suggestion and changed the preposition 'of' to 'in' on Lines 423-424, Page 11 of revised manuscript.

Line 431 'This was determined by gas-exchange and carbon isotopic measurements.' not needed we already know this.

Response: Considering your advice, we remove this redundant sentence on Line 425, Page 11 of revised manuscript.

Line 434 'in leaf' to 'in the leaf'

Response: Thanks for your helpful advice. We supplemented 'the' before the noun on Line 428, Page 11 of revised manuscript.

Line 436 remove 'the' before '$^{13}$C fractionation'

Response: According your advice, we removed 'the' before '$^{13}$C fractionation' on Line 430, Page 11 of revised manuscript.

Line 438 'worth evaluation.' to 'worth considering.'

Response: Thanks for your helpful advice. We changed 'worth evaluation.' to 'worth considering.' on Line 432, Page 11 of revised manuscript.

Lines 664-665 '…irrigation device' to '…irrigation device used in this study; numbers…'

Response: Due to redundant expression in caption of Figure 1, we rewrote it as '…irrigation device used in this study; numbers…' on Line 661, Page 17 of revised manuscript.

Line 666 'components' to 'component'

Response: We accept your helpful suggestion and changed 'components' to 'component' on Line 663, Page 17 of revised manuscript.

Figures needs improvement; considerably blurry.

Response: Considering your devices, we improved the resolution of Figures 1-8 in the revised manuscript.

Line 676 'of' to 'in'

Response: We accept your helpful suggestion and changed 'of' to 'in' on Line 666, Page 18 of revised manuscript.

Lines 677, 681, 686, 690, 693, 697, 700 'contents' to 'content treatments'

Response: We accept your helpful suggestion and changed 'contents' to 'content treatments' on Lines 667, 669-670, 673, 676, 678, 680, 683 Pages 18-24 of revised manuscript.

Line 684 'leaves of' to 'leaves from'

Response: We accept your helpful suggestion and changed 'leaves of' to 'leaves from' on Lines 671-672, Page 20 of revised manuscript.

Line 692, 696, 699 'of' to 'in'

Response: We accept your helpful suggestion and changed 'of' to 'in' on Lines 677, 679, 682, Pages 22-24 of revised manuscript.

In Figures 7-8, the coefficients of determination were reserved two decimal fractions while writing.

Response: Based on your suggestion, we kept coefficients of determination two decimals in Figures 7-8 of revised manuscript.

Line 703 'Orthogonal treatments of *P. orientalis* and *Q. variabilis* for four $CO_2$ concentrations $\times$ five soil volumetric water contents.' to 'Orthogonal treatments applied to *P. orientalis* and *Q. variabilis*.'

Response: According your advice, we changed 'of' to 'applied to' and simplified the caption of Table 1.

Line 706 'of' to 'in'

Response: Thanks for your helpful advices. We changed 'of' to 'in' on Line 688, Page 26 of revised manuscript.

Line 706 'for' to 'under'

Response: According your advice, we changed ''for' to 'under' on Line 688, Page 26 of revised manuscript.

Line 706 'contents' to 'content treatments'

Response: Thanks for your helpful advices. We changed 'contents' to 'content treatments' on Line 688, Page 26 of revised manuscript.

---

## Editor Decision (ED3)

[revised manuscript text omitted]
 (‰) | | | | Mesophyll conductance $^{13}C$ fractionation (‰) | | | | Post-photosynthesis $^{13}C$ fractionation (‰) | | | |
|---|---|---|---|---|---|---|---|---|---|---|---|---|---|
| | | 400 | 500 | 600 | 800 | 400 | 500 | 600 | 800 | 400 | 500 | 600 | 800 |
| *P. orientalis* | 35%–45% | 0.0328 | 0.0373 | 0.0349 | 0.0332 | 0.0081 | 0.0030 | 0.0034 | 0.0072 | 0.0247 | 0.0343 | 0.0315 | 0.0260 |
| | 50%–60% | 0.0367 | 0.0437 | 0.0382 | 0.0374 | 0.0018 | 0.0058 | 0.0094 | 0.0004 | 0.0349 | 0.0379 | 0.0288 | 0.0370 |
| | 60%–70% | 0.0405 | 0.0366 | 0.0421 | 0.0409 | 0.0018 | 0.0050 | 0.0026 | 0.0007 | 0.0387 | 0.0316 | 0.0395 | 0.0402 |
| | 70%–80% | 0.0444 | 0.0453 | 0.0413 | 0.0452 | 0.0044 | 0.0052 | 0.0103 | 0.0013 | 0.0400 | 0.0401 | 0.0310 | 0.0439 |
| | 100% | 0.0441 | 0.0453 | 0.0456 | 0.0472 | 0.0057 | 0.0040 | 0.0025 | 0.0039 | 0.0384 | 0.0413 | 0.0431 | 0.0433 |
| *Q. variabilis* | 35%–45% | 0.0388 | 0.0402 | 0.0406 | 0.0384 | 0.0007 | 0.0025 | 0.0006 | 0.0091 | 0.0381 | 0.0377 | 0.0400 | 0.0293 |
| | 50%–60% | 0.0433 | 0.0448 | 0.0409 | 0.0368 | 0.0061 | 0.0084 | 0.0023 | 0.0018 | 0.0372 | 0.0364 | 0.0386 | 0.0350 |
| | 60%–70% | 0.0424 | 0.0440 | 0.0445 | 0.0414 | 0.0066 | 0.0086 | 0.0078 | 0.0041 | 0.0358 | 0.0354 | 0.0367 | 0.0373 |
| | 70%–80% | 0.0424 | 0.0446 | 0.0482 | 0.0457 | 0.0034 | 0.0016 | 0.0074 | 0.0028 | 0.0390 | 0.0430 | 0.0408 | 0.0429 |
| | 100% | 0.0441 | 0.0466 | 0.0466 | 0.0398 | 0.0027 | 0.0076 | 0.0022 | 0.0125 | 0.0414 | 0.0390 | 0.0444 | 0.0273 |

$CO_2$ concentration (ppm)

---

## Author Response (AR4)

Response list to the Editor's comments

Ref: doi:10.5194/bg-2016-372

Title: **Interaction of $CO_2$ concentrations and water stress in semi-arid plants causes diverging response in instantaneous water use efficiency and carbon isotope composition**
Authors: Na Zhao, Ping Meng, Yabing He, Xinxiao Yu*

Dear Editor,

Thanks for your patiently help and constructive comments that provide scientific guidance for our writing and future research. We commissioned the LetPub Company (belonging to ACCDON (US) that is the professional editorial team) to provide professional editing help in rewriting the manuscript. We have been carefully considering your suggestions and revising the manuscript in the revised manuscript (marked in red color) accordingly. In addition to the following issues, we have corrected other mistakes with grammar and expression in the revised manuscript (marked in red color). The following below in blue are our point-to-point responses for your questions and comments. We are appreciated for your kind help on writing.

We are looking forward to your further comments and a possible publication in the BG special issue (Ecosystem processes and functioning across current and future dryness gradients in arid and semi-arid lands).

Kind regards,

Xinxiao Yu

■■■■■■■■■■■■■■■■■■■■■■■■■■■■■■■■■■■■■■■■■■■■■■■■■■■■■■■■■■■

Comments to the Author:

We changed "concentrations" to "concentration" on Line 18, Page 1 of revised manuscript.

Line 19 "contents" to "content"

Response: We appreciate your helpful comments and apologize for any inconvenience that we bring you for my carelessness in writing. Based on your helpful suggestions, we have changed "$CO_2$ concentrations…contents" to "$CO_2$ concentration…content" on Line 18, Page 1 of the revised manuscript.

Line 20 "determined for instantaneous water efficiency" to "determined for an assessment of instantaneous water efficiency"

Response: Thanks for your helpful comments and we have changed "determined for instantaneous water efficiency" to "determined for an assessment of instantaneous water efficiency" on Lines 19-20, Page 1 of the revised manuscript.

We changed "gas exchange" to "gas-exchange measurement" on Line 21, Page 1 of the revised manuscript.

Line 22 "of" to "in"

Response: According your helpful suggestions, we have changed "of" into "in" on Line 22, Page 1 of the revised manuscript.

Change the terms like "35%-45% of FC" to "35-45% of FC"; reduce the repetition throughout the article

Response: We have changed "35%-80% of …" to "35-80% of …" and similar problems throughout the revised manuscript.

Line 27 "trend" to "tendency"

Response: Thanks for your helpful comments and we have changed "trend" to "tendency" on Line 26, Page 1 of the revised manuscript.

Lines 45 "patterns, but will…" to "patterns, which will…"

Response: According your helpful comments, we have changed "patterns, but will…" to "patterns, which will…" on Line 45, Page 2 of revised manuscript.

Line 46 "arid regions, and lead to more frequent extreme events …" to "arid regions and lead to more frequent extreme flooding events …"

Response: We appreciate your helpful comments. We have removed the "," and changed "extreme events" to "extreme flooding events" on Lines 45-46, Page 2 of revised manuscript.

Line 50 "their" to "its"

Response: We appreciate your helpful comments and apologize for any inconvenience that we bring you for my carelessness in writing. We have changed "their" to "its" on Line 50, Page 2 of revised manuscript.

Line 53 "that, in turn, are responding physiologically to changes" to "which, in turn, respond physiologically to changes"

Response: We have changed "that, in turn, are responding physiologically to changes" to "which, in turn, respond physiologically to changes" on Line 53, Page 2 of the revised manuscript.

We have changed "discrimination ($^{13}\Delta$) of leaves" to "discrimination ($^{13}\Delta$) in leaves" on Line 54, Page 2 of the revised manuscript.

We have changed "provide timely feedback about the …" to "provide timely feedback to the …" on Line 54, Page 2 of revised manuscript.

Line 55 remove "the"

Response: Thank you for pointing the mistake and we have removed redundant "the" on Line 55 Page 2 of revised manuscript:

Line 58 Change "[$CO_2$]s" to "[$CO_2$]"

Response: Thanks for your helpful suggestion and we have changed "[$CO_2$]s" to "[$CO_2$]" throughout the revised manuscript.

Lines 69-70 "… metabolism (Gessler et al., 2008; Gessler et al., 2014) fractionation in leaves" to "… metabolism fractionation in leaves (Gessler et al., 2008; Gessler et al., 2014)"

Response: According your helpful comments, we have changed this part as "… metabolism (Gessler et al., 2008; Gessler et al., 2014)," on Line 68, Page 2 of revised manuscript.

Line 76 "as used" to "as considered"

Response: Thanks for your suggestion and we have changed "as used" to "as considered" on Line 75, Page 2 of revised manuscript.

Line 82 "in two measurements" to "in the two measurements"

Response: We agree with your suggestion and have revised this sentence on Page 2, Line 81 in the revised manuscript.

Line 83 "…, but it tends to" to "…, but tends to"

Response: According your helpful comments, we have revised this part on Page 3, Line 82 in the revised manuscript.

Line 93 "1-2 day" to "1-2 days"

Response: Based on your suggestion, we apologized for mistake and have changed "1-2 day" to "1-2 days'" on Line 91, Page 3 of revised manuscript.

Line 98 "the $\delta^{13}C$ of fast-turnover" to "the $\delta^{13}C$ of the fast-turnover"

Response: We accept your helpful suggestion and changed as "the $\delta^{13}C$ of the fast-turnover" on Line 96, Page 3 of revised manuscript.

Line 100 "controlled environment" to "controlled-environment"

Response: Thanks for your helpful advices. We rewrote this sentence as "controlled-environment" on Line 98, Page 3 of revised manuscript.

Line 101 take out "(FH-230, Taiwan Hipoint Corporation, Kaohsiung City, Taiwan)", you address this below.

Response: Based on your suggestion, we have removed "(FH-230, Taiwan Hipoint Corporation, Kaohsiung City, Taiwan)" on Line 98, Page 3 of revised manuscript.

We have changed "in *P. orientalis* and *Q. variabilis*, that is" to "in *P. orientalis* and *Q. variabilis*, which is" on Line 100, Page 3 of revised manuscript.

Line 112 "…116°5′45″E in Beijing" to "…116°5′45″E, Beijing"

Response: According your advice, we corrected this part as "…116°5′45″E, Beijing" on Line 109, Page 3 of revised manuscript.

Line 123 "can both" to "is designed to both"

Response: Thanks for your helpful suggestion and we have revised this part as "is designed to both" on Line 120, Page 3 of revised manuscript.

Line 129 "and drip irrigation component" to "and a drip irrigation component"

Response: Based on your suggestion, we have changed "and drip irrigation" to "and a drip irrigation" on Line 126, Page 4 of revised manuscript.

Line 133 "can be regulated" to "could be regulated"

Response: According your advice, we have revised this part as "could be regulated" on Line 130, Page 4 of revised manuscript.

Line 135 "(Tab. 1)" to "(Table. 1)"

Response: Based on your suggestion, we have corrected the form of Tables presented in the text as "(Table. 1)".

Lines 136-138 reduce the repetition

Response: I am very sorry for my careless in writing grammar and have rewritten this part throughout the revised manuscript.

Line 156 redundant expression

Response: Thanks for your helpful advices. We removed this sentence on Line 152, Page 4 of revised manuscript.

Line 161 "massspectrometer" to "mass-spectrometer"

Response: Thanks for your helpful advices. We have changed "massspectrometer" to "mass-spectrometer" on Line 156, Page 4 of revised manuscript.

Line 185 "the temperature and relative humidity on leaf surface" to "the leaf-surface temperature and relative humidity"

Response: We accept your helpful suggestion and changed "the temperature and relative humidity on leaf surface" to "the leaf-surface temperature and relative humidity" on Line 181, Page 5 of revised manuscript.

Line 186 "could be determined" to "was determined"

Response: We accept your helpful suggestion and changed "could be determined" to "was determined" on Line 182, Page 5 of revised manuscript.

Line 190 "sugars" to "sugar"

Response: Based on your suggestion, we have changed "sugars" to "sugar" on Line 185, Page 5 of revised manuscript.

We have changed "from $\Delta_{model}$" to "by $\Delta_{model}$" on Line 188, Page 5 of revised manuscript.

Line 197 "Method of estimation for mesophyll conductance" to "Method of estimating mesophyll conductance"

Response: According your advice, we changed "Method of estimation for mesophyll conductance" to "Method of estimating mesophyll conductance" on Line 192, Page 5 of revised manuscript.

Line 221 "actually measured…" to "measured…"

Response: Thanks for your helpful advices. We have changed "actually measured…" to "measured…" on Line 219, Page 6 of revised manuscript.

Lines 222-223 "…carboxylation that are the $^{13}C$ fractionation from mesophyll conductance" to "…carboxylation associated with $^{13}C$ fractionation from mesophyll conductance"

Response: We accept your helpful suggestion and have revised this part as "…carboxylation associated with $^{13}C$ fractionation from mesophyll conductance" on Lines 220-221, Page 6 of revised manuscript.

Line 224 "(Eqn. (12))" to "[Eqn. (12)]"

Response: Thanks for your helpful advices and we changed "(Eqn. (12))" to "[Eqn. (12)]" on Line 222, Page 6 of revised manuscript.

Line 229 "," following "and"

Response: We accept your helpful advice on Line 227, Page 6 of revised manuscript.

Line 232 "…or to be cancelled out in …" to "…or cancelled in…"

Response: We accept your helpful suggestion and changed "…or to be cancelled out in …" to "…or cancelled in…"on Line 230, Page 6 of revised manuscript.

Line 243 "SWCs" to "SWC"

Response: We accept your helpful suggestion.

Line 247 "higher than for $C_{800}$" to "higher than that for $C_{800}$"

Response: We accept your helpful suggestion on Line 245, Page 7 of revised manuscript.

Line 249 "higher than for $C_{600}$" to "higher than that for $C_{600}$"

Response: We accept your helpful suggestion on Line 246, Page 7 of revised manuscript.

Lines 259-260 "was significantly larger than that in *Q. variabilis* at any $[CO_2] \times SWC$ treatment…" to "was significantly higher than that in *Q. variabilis* for most $[CO_2] \times SWC$ treatments..."

Response: We accept your helpful suggestion and revised this part on Lines 257-278, Page 7 of revised manuscript.

Line 263 "at any $[CO_2]$" to "for most $[CO_2]$"

Response: We accept your helpful suggestion and changed "at any $[CO_2]$" to "for most $[CO_2]$" on Line 261, Page 7 of revised manuscript.

Lines 263-264 "…as SWC increased, while values increased as $[CO_2]$ increased" to "…as SWC increased and increased as $[CO_2]$ increased"

Response: According your suggestion, we changed this part as "…as SWC increased and increased as $[CO_2]$ elevated" on Lines 261-262, Page 7 of revised manuscript.

Line 290 "(Eqns. (10-17))" to "[Eqns. (10-17)]"

Response: We accept your helpful suggestion and changed "(Eqns. (10-17))" to "[Eqns. (10-17)]" on Line 287, Page 8 of revised manuscript.

Line 297 "at all SWCs in…" to "at all SWC for…"

Response: We accept your helpful suggestion and changed "at all SWCs in…" to "at all SWC for…"on Line 294, Page 8 of revised manuscript.

Lines 309-310 "with [$CO_2$] increases in…" to "with increases in [$CO_2$] in…"

Response: We accept your helpful suggestion and changed "with [$CO_2$] increases in…" to "with increases in [$CO_2$] in…" on Line 306, Page 8 of revised manuscript.

Line 321 "exceeded this water threshold" to "exceeded this soil water threshold"

Response: Thanks for your helpful advices. We have changed "exceeded this water threshold" to "exceeded this soil water threshold" on Line 318, Page 9 of revised manuscript.

Line 322 "is" to "was"

Response: Based on your suggestion, we have changed "is" to "was" on Line 319, Page 9 of revised manuscript.

Line 323 "by" to "with"

Response: Thanks for your helpful advices. We have changed "by" to "with" on Line 320, Page 9 of revised manuscript.

Line 332 plus "the"

Response: Based on your suggestion, we have plus "the" before the noun on Line 329, Page 9 of revised manuscript.

Line 337 "of perennial *Leymus chinensis* and" to "of a perennial, *Leymus chinensis*, and"

Response: Based on your suggestion, we have changed "of perennial *Leymus chinensis* and" to "of a perennial, *Leymus chinensis*, and"on Line 334, Page 9 of revised manuscript.

Line 339 "Miranda Apodaca et al. (2015)" to "Micanda Apodaca et al. (2015)"

Response: Based on our investigation, "Miranda -Apodaca et al. (2014)" is the right citation and we have revised this part on Lines 335-336 and 342-343, Page 9 of revised manuscript.

Lines 341-342 "results from other C3 woody plants" to "results seen with other C3 woody plants"

Response: We accept your helpful suggestion and changed "results from other C3 woody plants" to "results seen with other C3 woody plants" on Lines 338-339, Page 9 of revised manuscript.

Line 348 "The increases" to "Increases"

Response: According your advice, we have changed "The increases" to "Increases" on Line 345, Page 9 of revised manuscript.

Line 354 "iWUE, at the leaf level," to "iWUE at the leaf level,"

Response: According your advice, we have changed "iWUE, at the leaf level," to "iWUE at the leaf level," on Line 351, Page 9 of revised manuscript.

Line 359 "by" to "with"

Response: According your advice, we have changed "by" to "with" on Line 356, Page 10 of revised manuscript.

Line 362 "could respond to" to "responded to"

Response: We accept your helpful suggestion and changed "could respond to" to "responded to" on Line 359, Page 10 of revised manuscript.

Line 363 "is" to "was"

Response: Based on your suggestion, we have changed "is" to "was" on Line 360, Page 10 of revised manuscript.

We have changed "have" to "had" on Line 361, Page 10 of revised manuscript.

Line 370 "The latter procedure of diffusion is termed mesophyll conductance ($g_m$; Flexas et al., 2008)" comes late in the text; should have been referred be earlier.

Response: We accept your helpful suggestion and have redefined "$g_m$" on Lines 194-197, Page 5 of revised manuscript.

We have changed the "$g_m$" to "Mesophyll conductance, $g_m$, …" on Line 364, Page 10 of revised manuscript.

Lines 372-373 "7-day cultivations of SWC $\times$[CO$_2$]," to "7-day cultivations,"

Response: According your advice, we have changed "7-day cultivations of SWC $\times$[CO$_2$]," to "7-day cultivations," on Line 366, Page 10 of revised manuscript.

Line 374 "in" to "by"

Response: We accept your helpful suggestion and changed "in" to "by" on Line 367, Page 10 of revised manuscript.

Line 380 "compared with" to "compared to"

Response: We accept your helpful suggestion and changed "compared with" to "compared to" on Line 372, Page 10 of revised manuscript.

Line 385 "should be lucubrated" to "should be elucidated"

Response: Thanks for your helpful suggestion, we apologized for our carelessness and changed "should be lucubrated" to "should be elucidated" on Line 377, Page 10 of revised manuscript.

Line 415 "four [CO$_2$]s $\times$ five SWCs" to "four [CO$_2$] $\times$ five SWC"

Response: We accept your helpful suggestion and changed "four [CO$_2$]s $\times$ five SWCs" to "four [CO$_2$] $\times$ five SWC" on Line 406, Page 11 of revised manuscript.

Line 570 "Mirandan Apodaca…"

Response: We accept your helpful suggestion and changed "Mirandan Apodaca…2015" to "Micandan-Apodaca…2014" on Lines 558-560, Page 14 of revised manuscript.

Line 632 "performed experiments" to "performed the experiments"

Response: We accept your helpful suggestion and changed "performed experiments" to "performed the experiments" on Line 619, Page 16 of revised manuscript.

Lines 632-633 "performed data analysis" to "analyzed the data"

Response: We appreciate your helpful comments and changed "performed data analysis" to "analyzed the data" on Line 619, Page 16 of revised manuscript.

Line 665 simplify the labels; also too small

Response: According your helpful advice, we have simplified the labels and increased the font size of the labels on revised Figures 2-6 of revised manuscript.

Line 666 "concentrations" to "concentration"

Response: Thanks for your helpful comments. We have changed "concentrations" to "concentration" on Lines 635, 638, 641, 644, 646, 649, 652 and 657, Pages 18-24 and 26 of revised manuscript.

Line 679 "Regression" to "Regressions"

Response: According your comments, we have changed "Regression" to "Regressions" on Line 648 and 651, Pages 23 and 24 of revised manuscript.

Line 680 "$p = 0.01$" to "$p < 0.01$"

Response: Based on your consideration, we have changed "$p = 0.01$" to "$p < 0.01$" on Line 649, Page 23 of revised manuscript.

Line 683 "$p = 0.01$" to "$p \leq 0.01$"

Response: Based on your consideration, we have changed "$p = 0.01$" to "$p \leq 0.01$" on Lines 684-652, Page 24 of revised manuscript.

---

## Author Response (AR5)

Response list to the Editor's comments

Ref: doi:10.5194/bg-2016-372

Title: **Interaction of CO$_2$ concentrations and water stress in semi-arid plants causes diverging response in instantaneous water use efficiency and carbon isotope composition**

Authors: Na Zhao, Ping Meng, Yabing He, Xinxiao Yu*

Dear Editor,

Thanks for your patiently help and constructive comments that provide scientific guidance for our writing and future research. We have been carefully considering your suggestions and revising the manuscript in the revised manuscript (marked in red color) accordingly. In addition to the following issues, we have corrected other mistakes with grammar and expression in the revised manuscript (marked in red color). The following below in blue are our point-to-point responses for your questions and comments. We are appreciated for your kind help on writing.

We are looking forward to your further comments and a possible publication in the BG special issue (Ecosystem processes and functioning across current and future dryness gradients in arid and semi-arid lands).

Kind regards,

Xinxiao Yu

• • • • • • • • • • • • • • • • • • • • • • • • • • • • • • • • • • • • • • • • • • • • • • • • • • • • • • • •

Comments to the Author:

Line 21 "gas-exchange measurement" to "gas-exchange measurements"

Response: We appreciate your helpful comments and apologize for any inconvenience that we bring you for my carelessness in writing. Based on your helpful suggestions, we have changed "gas-exchange measurement" to "gas-exchange measurements" on Line 21, Page 1 of the revised manuscript.

Line 48 "0.02‰–0.03‰ year$^{-1}$" to "0.02–0.03‰ year$^{-1}$"

Response: Thanks for your helpful comments and we have changed "0.02‰–0.03‰ year$^{-1}$" to "0.02–0.03‰ year$^{-1}$" on Line 48, Page 2 of the revised manuscript.

Line 67 "ribulose-1, 5-bisphosphate, and internal diffusion" to "ribulose-1, 5-bisphosphate and internal diffusion"

Response: According your helpful suggestions, we have changed "ribulose-1, 5-bisphosphate, and internal diffusion" to "ribulose-1, 5-bisphosphate and internal diffusion" on Line 68, Page 2 of the revised manuscript.

Line 87 "the" to "an"

Response: Thanks for your helpful comments and we have changed "the" to "an" on Line 89, Page 3 of the revised manuscript.

Lines 95 "…, is unknown" to "…, is largely unknown"

Response: According your helpful comments, we have changed "…, is unknown" to "…, is largely unknown" on Line 97, Page 3 of revised manuscript.

Line 100 remove "," before "determined…"

Response: We appreciate your helpful comments. We have removed the "," on Lines 102-103, Page 3 of revised manuscript.

Line 108 "a part of Chinese…" to "a part of the Chinese…"

Response: We appreciate your helpful comments and we have changed "a part of Chinese…" to "a part of the Chinese…" on Line 110, Page 3 of revised manuscript.

Line 120 "is designed to both control and" to "was designed to control and"

Response: We have changed "is designed to both control and" to "was designed to control and" on Line 122, Page 3 of the revised manuscript.

Lines 121 and 122 remove "ppm"

Response: Thank you for pointing the mistake and we have removed redundant "ppm" on Line 123, Page 3 and Line 124, Page 4 of revised manuscript.

Line 128 "could be set" to "was set"

Response: Thanks for your helpful comments and we have changed "could be set" to "was set" on Line 130, Page 4 of the revised manuscript:

Line 130 "being watering or stop watering" to "being or stop watering"

Response: Thanks for your helpful suggestion and we have changed "being watering or stop watering" to "being or stop watering" on Line 132, Page 4 of the revised manuscript.

Lines 150 "50 mg of ground leaves" to "50 mg of grounded leaves"

Response: According your helpful comments, we have changed "50 mg of ground leaves" to "50 mg of grounded leaves" on Line 153, Page 4 of revised manuscript.

On page 4 no indentation is needed at the beginning of the paragraph

Response: Thanks for your suggestion and we have canceled the indentation at the beginning of the paragraph on Lines 152-160, Page 4 of revised manuscript.

Line 194 "first" to "firstly"

Response: We agree with your suggestion and have changed "first" to "firstly" on Page 5, Line 198 in the revised manuscript.

Line 195 "From sub-stomatic cavities $CO_2$ then…" to "From sub-stomatic cavities, $CO_2$ then…"

Response: According your helpful comments, we have changed "From sub-stomatic cavities $CO_2$ then…" to "From sub-stomatic cavities, $CO_2$ then…" on Page 5, Line 200 in the revised manuscript.

Line 209 "could be neglected" to "is negligible"

Response: Based on your suggestion, we have changed "could be neglected" to "is negligible" on Line 214, Page 6 of revised manuscript.

Line 218 "That" to "This"

Response: We accept your helpful suggestion and changed "That" to "This" on Line 223, Page 6 of revised manuscript.

Lines 219-220 "…, could" to "…, which can be used to"

Response: Thanks for your helpful advices. We rewrote this sentence as "…, which can be used to" on Line 225, Page 6 of revised manuscript.

Line 224 "is presented by" to "relates"

Response: Based on your suggestion, we have changed "is presented by" to "relates" on Line 229, Page 6 of revised manuscript.

Line 226 "…we obtain" to "…gives us"

Response: According your advice, we have changed "…we obtain" to "…gives us" on Line 231, Page 6 of revised manuscript.

Line 229 "terms of respiratory and photorespiratory could be" to "terms of respiration and photorespiration can be"

Response: Thanks for your helpful suggestion and we have revised this part as "terms of respiration and photorespiration can be" on Line 234, Page 7 of revised manuscript.

Line 239 "and/or" to "and"

Response: Based on your suggestion, we have changed "and/or" to "and" on Line 244, Page 7 of revised manuscript.

Line 242 "of $P_n$" to "in $P_n$"

Response: According your advice, we have changed "of $P_n$" to "in $P_n$" on Line 247, Page 7 of revised manuscript.

Line 261 no need for two digits after the decimal "from 90.70 to 564.65%"

Response: Based on your suggestion, we have corrected this part as "from 90.7 to 564.7%" on Line 266, Page 7 of revised manuscript.

Lines 265 "at any SWC" to "at all SWC"

Response: I am very sorry for my careless in writing grammar and have changed "at any SWC" to "at all SWC" on Line 270, Page 7 of revised manuscript.

Line 279 "with increasing SWC especially when" to "with increasing SWC, especially when"

Response: Thanks for your helpful advices. We added "," before "especially when" on Lines 284-285, Page 8 of revised manuscript.

Line 280 "(increased by 21.30-42.04%)" to "(increasing by 21.3-42.0%)"

Response: Thanks for your helpful advices. We have changed "(increased by 21.30-42.04%)" to "(increasing by 21.3-42.0%)" on Line 285, Page 8 of revised manuscript.

Line 284 "Total $^{13}$C fractionation, with increased SWC, in *P. orientalis* increased" to "Total $^{13}$C fractionation in *P. orientalis*, with increased SWC, increased"

Response: We accept your helpful suggestion and changed this part as "Total $^{13}$C fractionation in *P. orientalis*, with increased SWC, increased" on Lines 289-290, Page 8 of revised manuscript.

Line 286 "values of…" to "values desired from"

Response: We accept your helpful suggestion and changed "values of…" to "values desired from" on Line 292, Page 8 of revised manuscript.

Line 289 "which reached" to "reaching"

Response: Based on your suggestion, we have changed "which reached" to "reaching" on Line 295, Page 8 of revised manuscript.

Line 295 "under the same treatments." to "under the same treatment conditions."

Response: According your advice, we changed "under the same treatments." to "under the same treatment conditions." on Line 301, Page 8 of revised manuscript.

Line 320 "with increased SWC" to "with increases in SWC"

Response: Thanks for your helpful advices. We have changed "with increased SWC" to "with increases in SWC" on Line 326, Page 9 of revised manuscript.

Lines 323 "used directly" to "directly used"

Response: We accept your helpful suggestion and have changed "used directly" to "directly used" on Line 329, Page 9 of revised manuscript.

Line 330 "nonstomatal limitation" to "nonstomatal limitations"

Response: Thanks for your helpful advices and we changed "nonstomatal limitation" to "nonstomatal limitations" on Line 337, Page 9 of revised manuscript.

Line 336 remove "," following "that"

Response: We accept your helpful advice and removed "," on Line 343, Page 9 of revised

manuscript.

Line 348 "its physiological…" to "its different physiological…"

Response: We accept your helpful suggestion and changed "its physiological…" to "its different physiological…" on Line 355, Page 9 of revised manuscript.

Lines 359-360 "cultivated days" to "cultivation days"

Response: We accept your helpful suggestion and changed "cultivated days" to "cultivation days" on Line 367, Page 10 of revised manuscript.

Line 361 "variations" to "variation"

Response: We accept your helpful suggestion on Line 368, Page 10 of revised manuscript.

Line 249 "higher than for $C_{600}$" to "higher than that for $C_{600}$"

Response: We accept your helpful suggestion on Line 249, Page 7 of revised manuscript.

Lines 371 "9.08-44.42%" to "9.1-44.4%"

Response: We accept your helpful suggestion and revised this part on Line 378, Page 10 of revised manuscript.

Line 391 "might" to "may"

Response: We accept your helpful suggestion and changed "might" to "may" on Line 400, Page 11 of revised manuscript.

We changed "75.30-98.9%" to "75.3-98.9%" on Line 405, Page 11 of revised manuscript.